# From Kakeya to Kernels: A Multi-Scale Geometric Framework for Robust Representation Learning

## Abstract

This paper addresses the gap between the empirical efficacy of deep learning and the theoretical understanding of its robustness by introducing a novel geometric framework for representation learning, inspired by multi-scale analysis techniques used to resolve the Kakeya set conjecture. The concept of a representation field is formalized, modeling feature activations as geometric entities, and the notion of "stickiness" is defined as the stability of the geometric structure across network layers. The multi-scale Wolff axioms quantify this stability as a formal measure of representation quality. The principal contribution is the Sticky Representation Theorem, which establishes a provable relationship between a network's geometric stickiness and its functional robustness to input perturbations and resilience to missing modalities in multimodal settings. To operationalize this theoretical framework, the Katz-Tao Convex Wolff (KT-CW) Regularizer is derived as an architecture-agnostic loss term that can potentially incentivize the learning of provably robust, sticky representations. This work presents a new, unified approach for analyzing, understanding, and constructing more reliable AI systems within both single- and multi-modal contexts.

## 1 Introduction

Deep learning has come to define the current era of artificial intelligence (AI), with large language models (LLMs) exhibiting advanced reasoning capabilities and diffusion models generating photo-realistic images (Bansal et al., 2024; Sun et al., 2025; Wang et al., 2025). However, the rapid empirical advancements in these domains significantly outpace the theoretical understanding of their underlying mechanisms. A critical gap persists between the observed capabilities of models and the provable guarantees related to robustness and generalization (Dziugaite et al., 2020; Freiesleben & Grote, 2023).

Despite their superhuman performance on benchmarks, state-of-the-art (SOTA) models are fragile. They show vulnerabilities to adversarial perturbations (Moosavi-Dezfooli et al., 2017; Zhang et al., 2021) and frequently underperform when faced with distribution shifts (i.e., out-of-distribution) (Chakraborty et al., 2021; Liu et al., 2024c). This fragility undermines trust in safety-critical applications, such as autonomous driving and medical diagnosis. Various initiatives, including adversarial training (Shafahi et al., 2019), data augmentation (Shorten & Khoshgoftaar, 2019; Shorten et al., 2021), and probabilistic robustness (Kishan, 2021; Weng et al., 2019), offer partial solutions but fall short of establishing a principled framework that provides formal worst-case guarantees.

Central to the success of deep learning is the method of hierarchical representation learning, wherein neural networks such as Convolutional Neural Networks (CNNs) and Vision Transformers (ViTs) progressively transform raw inputs into increasingly abstract features (Raghu et al., 2021; Khan et al., 2022). While this process has proven effective, it lacks an architecture-agnostic framework. Key questions remain unanswered: What geometric structures and representations generalize effectively? Does the stability of learned representations vary as data progresses through the layers of a neural network?

This study aims to contribute to the development of such a theoretical foundation by utilizing tools from harmonic analysis and geometric measure theory. The research builds upon recent advance-

ments related to the Kakeya conjecture[1] (Wang & Zahl, 2022; 2025), specifically the resolution of the three-dimensional case through a multi-scale analysis framework (the connection between Kakeya "stickiness" and learning stability is detailed in Appendix B). The technique of "induction on scales" provides a rigorous framework for characterizing the geometry of high-dimensional sets across varying scales. This framework demonstrates that it can provide the multi-scale, quantitative foundation necessary to advance understanding and ultimately ensure the robustness of hierarchical deep representations.

By translating the power from harmonic analysis into the language of machine learning, this paper develops a new geometric framework for understanding and designing robust models. **Key contributions** are:

**A unified geometric theory for single- and multi-modal learning:** This paper presents a novel theoretical framework based on the multi-scale analysis of the Kakeya conjecture for quantifying the geometric structure of learned representations in deep networks. A representation field is defined to associate abstract feature vectors with concrete geometric objects. The multi-scale Wolff axioms for representations are introduced, providing rigorous, quantitative measures of feature collapse, sparsity, and a cross-modal alignment constant for evaluating the quality of geometric fusion in multimodal cases. This framework establishes a universal language for describing the emergent geometry of both single- and multi-modal representations.

**A provable theory of representation robustness:** The Sticky Representation Theorem is introduced and proven as a fundamental result that establishes a link between a measurable geometric property of hidden layers, termed "stickiness," and the functional robustness of models. This theorem shows that representations that maintain their geometric structure across layers exhibit provable stability under input perturbations. In multimodal settings, these stability measures characterize a model's robustness to missing modalities. This work introduces a novel class of theoretical guarantees within deep learning, correlating internal geometric properties with external robustness guarantees.

**A geometric reinterpretation of core machine learning concepts:** The proposed framework offers a novel geometrically grounded perspective on key machine learning concepts. It establishes a formal link between "grains decomposition" from harmonic analysis and hierarchical feature clustering, presenting a non-parametric, data-dependent alternative to attention mechanisms. Additionally, it reformulates the information bottleneck principle in geometric stability terms, introducing the feature collapse constant as a measurable proxy for information complexity, thereby addressing significant ambiguities in applying the principle to deep learning contexts.

**A new design principle for robust models:** The KT-CW regularizer is derived from the proposed theoretical framework, introducing an architecture-agnostic loss term that optimizes geometric stability during training. By penalizing feature collapse and cross-modal misalignment, this regularizer converts theoretical insights into a potential training tool for deep networks.

## 1.1 RELATED WORK

Previous research has elaborated various representations through three main approaches: (i) Geometric deep learning (GDL) (Bronstein et al., 2017; Cao et al., 2020; Ye, 2022), which enforces symmetries to achieve equivariance; (ii) wavelet scattering networks (WSNs) (Bruna & Mallat, 2013; Gauthier et al., 2022; Shi et al., 2021), which demonstrates deformation stability through the use of fixed multi-scale filters; and (iii) the information bottleneck (IB) (Tishby et al., 2000; Tishby & Zaslavsky, 2015; Geiger & Kubin, 2020), which conceptualizes learning as a process of compression characterized by mutual information (MI) $I(X; Z)$ vs. $I(Y; Z)$. In addition, techniques for assessing parallel robustness delineate sensitivity bounds using Lipschitz analysis (often yielding loose constraints in deep networks) (Virmaux & Scaman, 2018; Fazlyab et al., 2019; Gouk et al., 2021) or employ adversarial certifiers tailored to specific $l_p$ threats (Raghunathan et al., 2018; Ghiasi et al., 2020; Valentin, 2024; Anisetti et al., 2023; Liu et al., 2024a). However, a significant gap remains in these approaches as they either impose geometric structures a priori, depend on non-learned filters, or rely on unstable estimates of mutual information or weight-level bounds. Consequently, they provide limited, model-agnostic diagnostics of the emergent geometry that develops within standard

---

[1]Definition: The standard Kakeya set conjecture posits that a set of points in $\mathbb{R}^n$ containing a unit line segment in every direction must have Hausdorff dimension $n$.

learned networks. To address this gap, this work proposes a Kakeya-based framework to measure the emergent geometry using Wolff-style axioms that focus on collapse and density.

In multimodal systems, which encompass representation (Guo et al., 2019; Liang et al., 2022), alignment (Baltrušaitis et al., 2018; Liang et al., 2024), and fusion (Zhang et al., 2020; Zhao et al., 2024), joint embeddings (Balaneshin-Kordan & Kotov, 2018; Suzuki et al., 2016), such as those found in image-text models like CLIP (Radford et al., 2021), are commonly employed to co-locate semantically related items. Numerous heuristics exist for loss functions and fusion methodologies, which, despite yielding strong empirical results, remain theoretically underexplored with respect to the geometric properties that underpin alignment quality, resilience to missing or noisy modalities, and robustness to out-of-distribution inputs. To date, the existing literature lacks principled, quantitative assessments of intra-modal structures and cross-modal interactions within the shared embedding space, as well as modality-aware assurances of robustness beyond empirical evaluations. To extend the Kakeya framework into the joint embedding space, this work introduces concepts of intra-modal stickiness and a cross-modal alignment axiom. More literature review is presented in Appendix C.

## 2 PRELIMINARIES AND PROBLEM FORMULATION

### 2.1 FOUNDATIONS FROM MULTI-SCALE GEOMETRIC ANALYSIS

The primary mathematical tools used in this study are derived from recent advances by Wang & Zahl (2025) on the Kakeya conjecture[2]. Their work provides a framework that is conducive to multi-scale analysis of geometric objects.

**Definition 1 (Scales and $\delta$-tubes).** A scale is a small positive number $\delta \in (0, 1]$. A $\delta$-tube $T$ in $\mathbb{R}^n$ is the $\delta$-neighborhood of a unit line segment. A collection of such tubes is denoted by a set $\mathcal{T}$. The analysis is fundamentally multi-scale, relating the properties of objects at a fine scale $\delta$ to those at a coarser scale $\rho > \delta$.

**Definition 2 (The Wolff Non-Clutering Axioms).** These axioms provide a rigorous language for what it means for a set of objects to be "spread out" or "non-clustered." They are important for preventing degenerate configurations where all tubes are trivially packed into a small volume.

(1) **Katz-Tao Convex Wolff ($C_{KT-CW}$) Axiom:** The constant $C_{KT-CW}(\mathcal{T})$ is the infimum of all $C > 0$ such that for any convex set $W \subset \mathbb{R}^n$, the number of tubes from $\mathcal{T}$ contained in W satisfies:

$$\#\{T \in \mathcal{T} : T \subset W\} \leq C \cdot |W| \cdot (\#\mathcal{T}). \tag{1}$$

A small $C_{KT-CW}(\mathcal{T})$ implies that the tubes are sparse; they cannot concentrate in high numbers within any convex region relative to its volume. This axiom penalizes feature redundancy or collapse.

(2) **Frostman Slab Wolff ($C_{F-SW}$) Axiom:** The constant $C_{F-SW}(\mathcal{T})$ is the infimum of all $C > 0$ such that for any slab $W \subset \mathbb{R}^n$ (the region between two parallel hyperplanes), the number of tubes from $\mathcal{T}$ contained in $W$ satisfies:

$$\#\{T \in \mathcal{T} : T \subset W\} \leq C \cdot |W| \cdot (\#\mathcal{T}). \tag{2}$$

A small $C_{F-SW}(\mathcal{T})$ is a measure of how well-distributed the tubes are with respect to planar regions.

**Definition 3 (Inductive Volume Estimates and "Stickiness").** The core of the latest findings in the Kakeya conjecture lies in two formulated assertions about the volume of the union of tubes, designed to be amenable to an inductive, self-improving argument across scales.

(1) **Assertion $D(\sigma, \omega)$:** States that if a set of tubes $\mathcal{T}$ is well-behaved (i.e., its $C_{KT-CW}$ and $C_{F-SW}$ constants are small), then the volume of its union has a lower bound of the form $|\bigcup T| \geq \kappa \delta^{\omega+\epsilon}(\#\mathcal{T})|T| \left((\#\mathcal{T})|T|^{1/2}\right)^{-\sigma}$.

(2) **Assertion $E(\sigma, \omega)$:** A more general statement that holds for any set of tubes, where the volume bound is explicitly penalized by the Wolff axiom constants.

---

[2]While the general conjecture allows segments to be placed anywhere (translation invariance), this work adapts the geometry to a "radial" setting where the segments (tubes) representing feature vectors are anchored at the origin to explicitly model feature magnitude and direction.

The parameters $\sigma$ and $\omega$ quantify the "wastefulness" of tube packing, while the Kakeya conjecture is equivalent to demonstrating the truth of Assertions $D(0,0)$ and $E(0,0)$. A set of features shows a structural property known as stickiness if its geometric arrangement, as defined by the Wolff axioms, remains stable (i.e., the rank of the set does not degrade or collapse as the scale $\delta \to 0$) and well-behaved (i.e., the set satisfies the Wolff non-clustering axioms, meaning it does not contain dense clusters or planar concentrations) across various scales. This property ensures that the set is structurally complex and non-degenerate at all levels of resolution. In contrast, a "non-sticky" set is characterized by a geometric structure that collapses at certain scales, thereby forcing the configuration into a lower-dimensional subspace and restricting its volume to a minimal extent.

## 2.2 Formalism for Hierarchical Representation Learning

This section now presents the corresponding formal definitions from machine learning, covering both single- and multi-modal scenarios.

**Definition 4 (Hierarchical Representation: Single-Modal).** A deep neural network is a parameterized function $f_\Theta : \mathcal{X} \to \mathcal{Y}$ that is a composition of $L$ layers, $f_\Theta = f_L \circ \cdots \circ f_1$. Each layer $f_l : \mathcal{Z}_{l-1} \to \mathcal{Z}_l$ is a function mapping from one representation space to another, where $\mathcal{X} = \mathcal{Z}_0$ is the input space. For a given input $x \in \mathcal{X}$, the activation or feature vector at layer $l$ is $z_l = (f_l \circ \cdots \circ f_1)(x)$. The set of all such activations for a dataset $\{x_i\}_{i=1}^N$ forms the representation at layer $l$, denoted $\mathbf{Z}_l = \{z_{l,i}\}_{i=1}^N \subset \mathcal{Z}_l$.

**Definition 5 (Hierarchical Representation: Multimodal).** A multimodal learning system operates on input data from $M \geq 2$ distinct modalities. An input instance is a tuple $x = \left(x^{(1)}, \ldots, x^{(M)}\right)$, where each $x^{(m)} \in \mathcal{X}^{(m)}$ belongs to the space of the $m$-th modality. The system typically comprises:

(1) A set of $M$ modality-specific encoders, $\left\{ g_\Theta^{(m)} : \mathcal{X}^{(m)} \to \mathcal{Z}^{(m)} \right\}_{m=1}^M$, that map each modality's raw input into a feature representation.

(2) A fusion mechanism, $h_\Phi$, that combines the individual representations. Joint embedding maps all unimodal representations into a common, shared Joint Embedding Space $\mathcal{Z}_{\text{joint}}$. In this case, each encoder is a map $g_\Theta^{(m)} : \chi^{(m)} \to \mathcal{Z}_{\text{joint}}$. The set of all embeddings for a dataset is $\mathbf{Z}_{\text{joint}} = \bigcup_{m=1}^M \mathbf{Z}^{(m)}$, where $\mathbf{Z}^{(m)} = \left\{ g_\Theta^{(m)}\left(x_i^{(m)}\right) \right\}_i$.

**Definition 6 (Lipschitz Stability and Functional Robustness).** A key desired property of a learned representation is stability with respect to small, irrelevant variations in the input. Let $\mathcal{V}$ be a set of deformation operators $\nu : \mathcal{X} \to \mathcal{X}$. The stability of the function $f$ can be quantified by its Lipschitz constant with respect to these deformations:

$$\mathcal{L}_\mathcal{V}(f) = \sup_{x \in \mathcal{X}, \nu \in \mathcal{V}, \nu \neq \text{id}} \frac{\|f(x) - f(\nu(x))\|\dagger}{\|\nu\|}, \tag{3}$$

where $\| \cdot \|\dagger$ is a metric on the output space and $\|\nu\|$ is a measure of the deformation's magnitude. A small Lipschitz constant implies that small deformations of the input result in small changes to the output, a hallmark of a robust model. Obtaining provable guarantees on such properties is a major goal of theoretical deep learning. A notation summary of this research is provided in Appendix D.

## 3 Main Results: Single-Modal Representation

### 3.1 The Representation Field

To effectively apply the tools of geometric measure theory, it is important to transform the abstract set of feature activations from a specific network layer into a tangible collection of geometric objects.

**Definition 7 (The Representation Field).** Let $\mathbf{Z}_l \subset \mathbb{R}^{d_l}$ be the set of feature activations at layer $l$ for a given dataset. For a chosen scale parameter $\delta > 0$, the representation field $\mathcal{T}_l(D)$ is a set of $\delta$-tubes in $\mathbb{R}^{d_l}$ defined as:

$$\mathcal{T}_l(D) = \left\{ T_z \subset \mathbb{R}^{d_l} \mid z \in \mathbf{Z}_l \right\}, \tag{4}$$

where each feature tube $T_z$ is the $\delta$-neighborhood of the line segment connecting the origin to the feature vector $z$. The direction of the tube is given by the normalized feature vector $\frac{z}{\|z\|}$, and its length is $\|z\|$.

This definition provides the crucial link: it transforms a discrete set of abstract vectors into a tangible geometric arrangement that can be analyzed using the multi-scale machinery of harmonic analysis. The scale parameter $\delta$ serves as a metric for the "granularity" at which the feature space is analyzed, analogous to the radius of the tubes in the Kakeya problem.

## 3.2 GEOMETRIC MEASURES OF REPRESENTATION QUALITY

Following Definition 7, it becomes possible to incorporate the Wolff axioms from harmonic analysis and reinterpret them as quantitative metrics for assessing the structural quality of a learned representation. These axioms establish a formal framework to articulate the geometry of a learned feature manifold, thereby contributing to a deeper understanding of its structural characteristics. Satisfying these axioms ensures that Definition 3 holds, which forms the basis of the Lipschitz bound in Theorem 1.

**Axiom 1 (Feature Collapse Constant).** This constant $C_{KT-CW}$ of the representation field $\mathcal{T}_l$ measures the degree of feature clustering or redundancy. A large $C_{KT-CW}(\mathcal{T}_l)$ indicates that many different inputs are mapped to feature tubes that are geometrically close or contained within the same small convex regions of the representation space. This signifies a collapse in representational diversity, where the network fails to learn discriminative features and instead maps distinct inputs to similar locations in the feature space. A small constant, conversely, indicates a geometrically sparse and non-redundant representation. While our axioms constrain volume density, their primary purpose in this framework is to serve as a geometric proxy for maintaining effective dimensionality and preventing rank collapse.

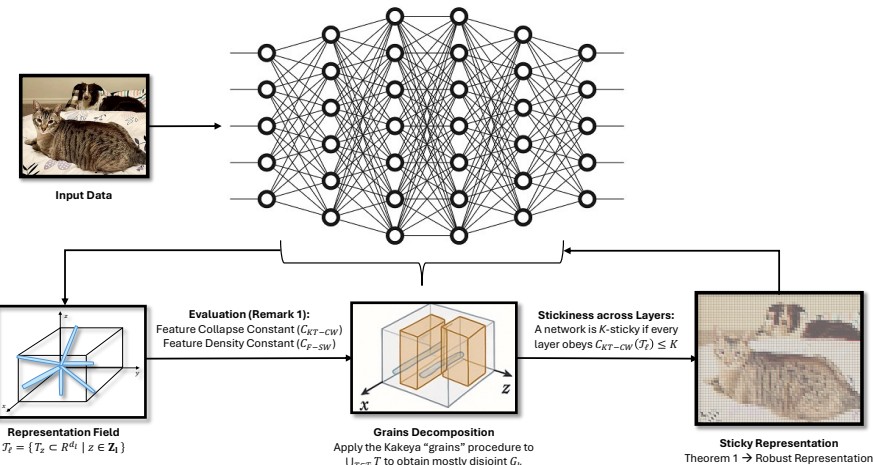

Figure 1: Single-modal geometric framework.

**Axiom 2 (Feature Density Constant).** This constant $C_{F-SW}$ of $\mathcal{T}_l$ measures how densely the feature tubes fill the representation space with respect to hyperplane-like regions (slabs). It quantifies the distribution of features across the entire space.

**Remark 1 (Geometric Sparsity as a Proxy for Information Complexity).** These geometric axioms provide a powerful new lens through which to view the IB principle. The IB framework seeks a representation Z that optimally balances complexity, measured by the mutual information $I(X; Z)$, and predictive power, measured by $I(Y; Z)$. The proposed geometric framework provides a more direct and stable method for quantifying the complexity term $I(X; Z)$. A representation field $\mathcal{T}_l$ with a low $C_{KT-CW}(\mathcal{T}_l)$ is, by definition, geometrically sparse and non-redundant. This geometric sparsity is a tangible signature of an information-theoretically simple, or compressed, representation. The objective of the latest Kakeya research, to find a large volume estimate for a set of tubes subject to the geometric constraints of the Wolff axioms, is thus deeply analogous to the IB objective of maximizing predictive information subject to a complexity constraint. By grounding the notion of complexity in a stable geometric measure rather than a volatile statistical estimate, our framework avoids the central controversies that have hindered the application of IB to deep learning.

### 3.3 Grains Decomposition as Data-Dependent Feature Clustering

In Kakeya, "grains decomposition[3]" organizes a collection of tubes into a more structured arrangement of predominantly disjoint rectangular prisms, referred to as "grains." When applied to a representation field, it gains a novel and insightful interpretation within the context of machine learning. This approach facilitates a deeper understanding of the underlying structures and relationships in the data, thereby enhancing the utility of "grains decomposition" in various applications.

**Lemma 1 (Grains as Meta-Features).** Applying the grains decomposition algorithm to a representation field $\mathcal{T}_l$ is equivalent to a data-dependent, unsupervised clustering of the feature tubes. Each resulting grain $G \subset \mathcal{Z}_l$ corresponds to a cluster of feature activations that are geometrically proximate and co-linear. These grains can be interpreted as learned "metafeatures," representing common combinations or motifs of the features from layer $l$.

This result formalizes the long-held intuition that deep networks construct hierarchies by grouping simple features to form more complex representations. This geometric clustering provides a theoretical framework for understanding the functions of mechanisms (e.g., attention) in Transformers. The attention mechanism learns a data-dependent pooling operation to aggregate and weigh features (tokens). In contrast, the grains decomposition also functions as a data-dependent grouping mechanism; however, its construction is based on the geometry of the representation field rather than relying on learned parameters.

This observation indicates that the grains decomposition can be characterized as a principled, non-parametric analog of attention, thereby paving the way for the development of new "geometric attention" layers that demonstrate more provable stability properties. The full proof for this lemma is provided in Appendix F.1.

Figure 2 presents a comparison of sticky and non-sticky representation fields with the grains decomposition. In a sticky configuration, feature activations at layer $l$ form a representation field $\mathcal{T}_l = \{T_z\}$ with mostly disjoint rectangular grains $G_k$, leading to low feature-collapse and density constants that promote a $K$-sticky hierarchy, ensuring well-distributed $\delta$-tubes and geometry preservation. In a non-sticky configuration, tubes converge into a convex "feature-collapse" region $W$, with overlapping grains resulting in high feature collapse and density constants, which create a geometric bottleneck and degrade generalization.

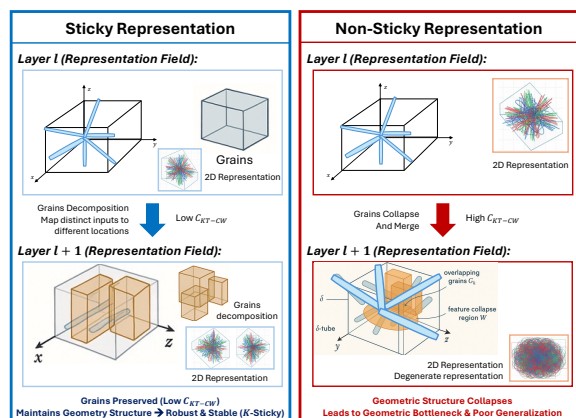

Figure 2: Sticky vs. non-sticky representation fields with grains decomposition.

### 3.4 The Sticky Representation Theorem (Single-Modal Case)

The central result for the single-modal case establishes a provable connection between the geometric property of "stickiness" and the functional attribute of robustness.

**Definition 8 ($K$-Sticky Representation).** A hierarchical representation $f = f_L \circ \cdots \circ f_1$ is $K$-sticky if the sequence of its representation fields $\{\mathcal{T}_l\}_{l=1}^{L}$ satisfies the KT-CW axiom at every layer with a uniformly bounded constant:

$$C_{KT-CW}\left(\mathcal{T}_l\right) \leq K \quad \text{for all } l = 1, \ldots, L. \tag{5}$$

This definition is a direct translation of the "stickiness" property, which is the crucial ingredient for a Kakeya set to have maximal dimension. A sticky representation is one where features remain well-structured and do not collapse into redundant, low-dimensional configurations as they are processed through the network. It provides a layer-wise diagnostic for model quality; a layer $l$ with a large $C_{KT-CW}\left(\mathcal{T}_l\right)$ can be identified as a "geometric bottleneck" where representational diversity is lost.

---

[3]Definition: Rectangular prisms of longitudinal length 1 and cross-sectional diameter $\rho$ with specific coherence and incidence properties. More details are in Appendix F.1.

**Theorem 1 (The Sticky Representation Theorem).** Let $f$ be a $K$-sticky hierarchical representation. Then $f$ is Lipschitz-stable with respect to a class of input deformations $\mathcal{V}$, with a Lipschitz constant $\mathcal{L}_\mathcal{V}(f)$ that is a monotonically increasing function of the stickiness parameter $K$. That is,

$$\mathcal{L}_\mathcal{V}(f) \leq g(K), \tag{6}$$

for some monotonically increasing function $g$.

This theorem, to our knowledge, provides the first provable link between a multi-scale geometric property of a network's hidden layers and its functional robustness to input perturbations. Figure 1 illustrates the proposed framework in a single-modal scenario. The full proof for this theorem is provided in Appendix F.2.

## 4 EXTENSION: MULTIMODAL REPRESENTATION

### 4.1 GEOMETRIC FORMULATION OF MULTIMODAL LEARNING

The primary challenge in multimodal learning involves bridging the "heterogeneity gap" by integrating information from diverse sources into a cohesive framework. This process requires formalizing the geometric space in which such integration takes place.

**Definition 9 (The Joint Embedding Metric Space).** Let $\left\{ g_\Theta^{(m)} : \chi^{(m)} \to \mathcal{Z}_{\text{joint}} \right\}_{m=1}^{M}$ be a set of $M$ encoders that map inputs from $M$ different modalities into a common, $\mathcal{Z}_{\text{joint}} \subset \mathbb{R}^d$. $\mathcal{Z}_{\text{joint}}$ is equipped with a metric $d_\mathcal{Z}$, which is typically learned (e.g., a Mahalanobis distance) or is the standard Euclidean distance. The objective of multimodal representation learning is to learn the encoder parameters $\Theta$ such that the geometry of this space reflects semantic relationships across modalities. That is, $d_\mathcal{Z}\left( g_\Theta^{(i)}\left(x^{(i)}\right), g_\Theta^{(j)}\left(x^{(j)}\right) \right)$ should be small if the underlying concepts of $x^{(i)}$ and $x^{(j)}$ are related, and large otherwise.

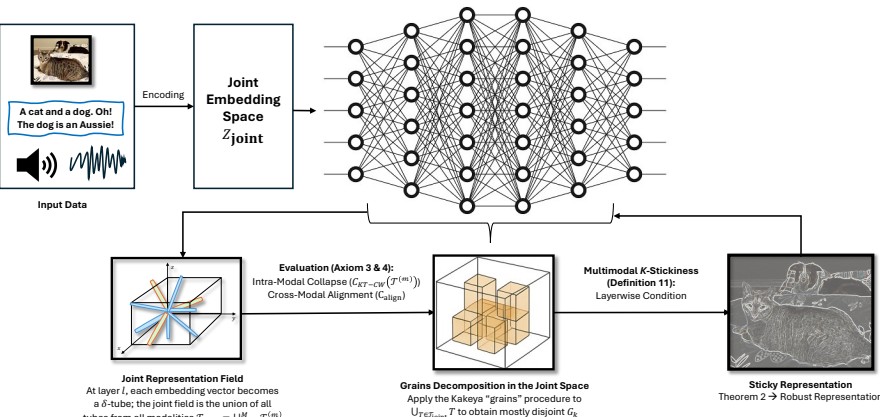

Figure 3: Multimodal geometric framework.

**Definition 10 (The Joint Representation Field).** For a given multimodal dataset, the set of all embeddings in the joint space forms the joint representation $\mathbf{Z}_{\text{joint}} = \bigcup_{m=1}^{M} \mathbf{Z}^{(m)}$. The joint representation field $\mathcal{T}_{\text{joint}}$ is the set of $\delta$-tubes in $\mathbb{R}^d$ corresponding to the vectors in $\mathbf{Z}_{\text{joint}}$. This is the central geometric object for our multimodal analysis. It is composed of subsets of tubes corresponding to each modality, $\mathcal{T}_{\text{joint}} = \bigcup_{m=1}^{M} \mathcal{T}^{(m)}$, where $\mathcal{T}^{(m)}$ is the representation field for the $m$-th modality.

### 4.2 MULTIMODAL GEOMETRIC STRUCTURE

The Wolff axioms are hereby extended to the joint space, involving the definition of new measures that not only capture the intrinsic structure within each modality's representation but also, importantly, explain the geometric relationships between the modalities.

**Axiom 3 (Intra-Modal Collapse, $C_{KT-CW}(\mathcal{T}^{(m)})$).** For each modality $m \in \{1, \ldots, M\}$, the standard Feature Collapse Constant can be computed on its corresponding subset of tubes $\mathcal{T}^{(m)} \subset \mathcal{T}_{\text{joint}}$.

This axiom measures the representational diversity and geometric sparsity within each modality. A low value for all $m$ is a necessary precondition for a good multimodal representation, as it ensures that each encoder produces a high-quality, non-redundant unimodal representation.

**Axiom 4 (Cross-Modal Alignment Constant, $C_{\text{align}}$).** This is a novel axiom designed specifically to measure the quality of geometric alignment between modalities in the joint space. Let $W$ be any convex set in the joint space $\mathbf{Z}_{\text{joint}}$. $C_{\text{align}}(\mathcal{T}_{\text{joint}})$ is defined as the infimum of $C > 0$ such that the relative variance in the number of tubes from different modalities contained within $W$ is bounded:

$$\frac{\text{Var}_m\left(\#\left\{T \in \mathcal{T}^{(m)} : T \subset W\right\}\right)}{\text{E}_m\left(\#\left\{T \in \mathcal{T}^{(m)} : T \subset W\right\}\right)} \leq C. \tag{7}$$

A small $C_{\text{align}}$ implies that the different modalities are well-mixed throughout the joint space; no single modality's features dominate any local region. This provides a direct, geometric measure of successful fusion. A large $C_{\text{align}}$, in contrast, indicates poor alignment. This can manifest as "modality dominance," a known issue where a model overrelies on one modality. In the proposed framework, this phenomenon has a clear geometric signature: the tubes corresponding to the dominant modality cluster in certain regions of the joint space, leading to high variance in tube counts within those regions and, consequently, a large $C_{\text{align}}$. This axiom transforms an empirical observation into a mathematically tractable property.

**Definition 11 (Multimodal $K$-Stickiness).** A multimodal representation, including its encoders and fusion mechanism, is $K$-sticky if, across all layers of the fusion network, both the intra-modal collapse constants and the cross-modal alignment constant are uniformly bounded:

$$C_{KT-CW}\left(\mathcal{T}_l^{(m)}\right) \leq K \quad \text{and} \quad C_{\text{align}}(\mathcal{T}_{l,\text{joint}}) \leq K, \tag{8}$$

for all layers $l$ and modalities $m$. This definition formalizes the notion of a robust multimodal representation as one that preserves both the internal geometric structure of each modality and the geometric coherence between them throughout the processing hierarchy.

## 4.3 THE STICKY REPRESENTATION THEOREM (MULTIMODAL CASE)

The primary theoretical result for the multimodal setting is presented below. It extends the robustness guarantee to address the complexities of multimodal data, including missing or noisy modalities.

**Theorem 2 (The Sticky Representation Theorem).** Let a multimodal system with a joint embedding space be $K$-sticky. The system is then robust to perturbations and missing modalities. Specifically:

(1) A function of $K$ bounds its Lipschitz constant with respect to input deformations, $\mathcal{L}(f) \leq g_1(K)$.

(2) Its performance degradation (the bound on the distance between the full-modal embedding and the missing-modal embedding.) when a subset of modalities is dropped at inference time is also bounded by a function of $K$.

This extension of the proposed theory delineates a principled pathway for evolving a single-modal architecture into a robust multimodal framework. The single-modal theory can be regarded as a specific instance of the multimodal theory, characterized by the condition where $M = 1$. The fundamental geometric principles, namely the avoidance of feature collapse and the preservation of structural integrity across scales ("stickiness"), are universally applicable. In the multimodal context, the primary challenge stems from the introduction of additional geometric complexities due to interactions between modalities. To address this challenge, $C_{\text{align}}$ is introduced to quantify this complexity explicitly. The full proof for this theorem is provided in Appendix F.3.

This framework establishes clear design principles for multimodal systems. The objective extends beyond mere feature fusion; it involves co-designing modality-specific encoders and a fusion mechanism to learn a cohesive joint representation field that maintains joint stickiness. Consequently, the encoders are required to generate individually non-collapsed representations, indicated by low $C_{KT-CW}\left(\mathcal{T}^{(m)}\right)$ for each modality. Simultaneously, the fusion mechanism must facilitate the

alignment of these representations within the joint space without inducing new collapses or geometric misalignments, as characterized by low $C_{\text{align}}$. This approach reframes the design of multimodal architectures into a formal problem of constrained geometric optimization, thereby enhancing the rigor and effectiveness of the system's design process. Figure 3 illustrates the proposed geometric framework in a multimodal scenario.

# 5 A NEW DESIGN PRINCIPLE: THE KT-CW REGULARIZER

The theoretical insights outlined in the preceding sections underpin a practical methodology for training more robust neural networks. Suppose a "sticky" representation can be demonstrated to possess provable robustness. In that case, it becomes feasible to explicitly promote the learning of such representations by introducing a penalty for deviations from the desired geometric structure during training. This section translates the theoretical framework into a differentiable tool that deep learning practitioners can utilize. Hence, a novel, architecture-agnostic loss term is proposed to directly minimize the geometric constants that quantify feature collapse and misalignment. This approach enhances the robustness and effectiveness of the model by addressing inherent deficiencies in feature representation.

**Proposition 1 (The KT-CW Regularizer).** For a single-modal network, the KT-CW regularizer is defined as a weighted sum of the feature collapse constants across all layers:

$$\mathcal{L}_{KT-CW} = \sum_{l=1}^{L} \lambda_l C_{KT-CW}\left(\mathcal{T}_l\right), \tag{9}$$

where $\{\lambda_l\}$ are hyperparameters that weight the importance of stickiness at each layer $l$. For a multimodal network with a joint embedding space, the regularizer is extended to penalize both intra-modal collapse and cross-modal misalignment:

$$\mathcal{L}_{MM-KT-CW} = \sum_{l=1}^{L}\left(\lambda_l C_{\text{align}}\left(\mathcal{T}_{l,\,\text{joint}}\right) + \sum_{m=1}^{M} \mu_{l,m} C_{KT-CW}\left(\mathcal{T}_l^{(m)}\right)\right), \tag{10}$$

where $\{\lambda_l\}$ and $\{\mu_{l,m}\}$ are hyperparameters. Minimizing this combined loss during training directly penalizes feature collapse within each modality. It encourages the network to find weight configurations that yield a geometrically well-mixed and sparse joint representation.

The optimization process informed by this regularizer can be interpreted as a computational analog of the constructive moves used to resolve the Kakeya conjecture. In theory, these moves represent iterative geometric refinements applied to the grains decomposition, thereby enhancing structural properties such as elongation or broadening of the grains. Similarly, each iteration of gradient descent during network training with the $\mathcal{L}_{KT-CW}$ regularizer modifies the network's weights, thereby altering the geometry of its representation fields. The regularizer provides a gradient that explicitly directs this geometric transformation toward a "sticky" configuration, akin to the constructive refinement process in the Kakeya conjecture. The full proof for this proposition is provided in Appendix F.4, and more theoretical results are discussed in Appendix E.

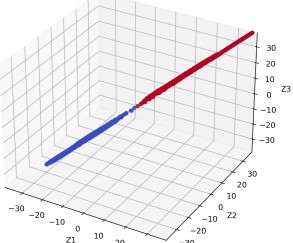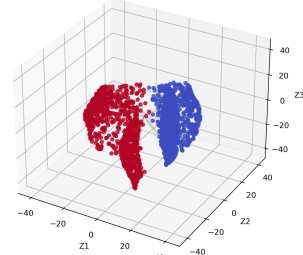

Figure 4: Sticky vs. non-sticky representations in 3D space (seed = 1022, $z_{dim}$ = 256, N = 5000). The grey lines represent the "tubes" defined in Definition 7. Non-sticky field: Tubes cluster into a narrow cone/subspace. Sticky field: Tubes splay out, covering the sphere (geometric sparsity).

# 6 SIMULATIONS AND EMPIRICAL VALIDATIONS

To validate the theoretical guarantees outlined in this paper, we performed a series of controlled simulations. These experiments aimed to isolate and examine the geometric properties of "Stickiness" and "Feature Collapse" within high-dimensional contexts. We contrasted standard training

methods with those guided by our proposed framework, enabling a comprehensive assessment of the regularization approach's effectiveness. Full experimental setup and additional results are detailed in Appendix G.

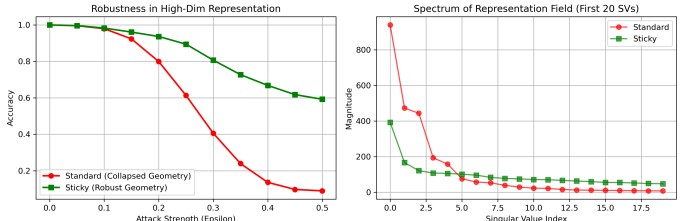

Figure 5: High-dimensional FGSM robustness and spectrum analysis for the make_moons dataset (seed = 1022, hidden dimension = 256, N = 5000).

Figure 4 provides the geometric intuition for the representation field ($\mathcal{T}_l(D)$) defined in Definition 7. It visualizes the latent activations of a neural network trained on a complex manifold (Swiss Roll), contrasting non-sticky vs. sticky geometries. In the left panel, the representation shows severe feature collapse, a phenomenon penalized by the Katz-Tao Convex Wolff Axiom ($C_{KT-CW}$). The features cluster densely into a lower-dimensional "pencil" shape. While linearly separable, this configuration is geometrically degenerate; any perturbation orthogonal to the pencil pushes data off the manifold, leading to high sensitivity. In the right panel, the KT-CW regularizer forces the representation field to satisfy the Wolff axioms, resulting in geometric sparsity. The feature vectors splay out to cover the angular space (sphere).

Figure 5 provides the direct empirical validation of Theorem 1, linking the geometric property of stickiness to the functional property of adversarial robustness. In the left panel, the standard model shows a catastrophic failure mode, while the sticky model demonstrates superior stability. To understand the geometric mechanism behind this robustness, we analyze the Singular Value Decomposition (SVD) of the representation field. The standard model exhibits sharp spectral decay, indicating severe feature collapse, and compresses the high-dimensional feature space into a low-rank subspace. The sticky model maintains a heavier tail in its spectrum, utilizing a higher effective rank. This confirms that Proposition 1 successfully prevents collapse, enforcing the full-rank geometric structure required for $K$-stickiness.

Figure 6 illustrates the results obtained from a representative configuration. The left panel shows the average embedding shift in the joint representation after excluding modality B. The Standard model demonstrates a substantial embedding shift, suggesting that the joint representation is geometrically unstable and overly dependent on specific modalities. In contrast, the Sticky model effectively minimizes this shift, thereby empirically validating the bound established in Theorem 2.

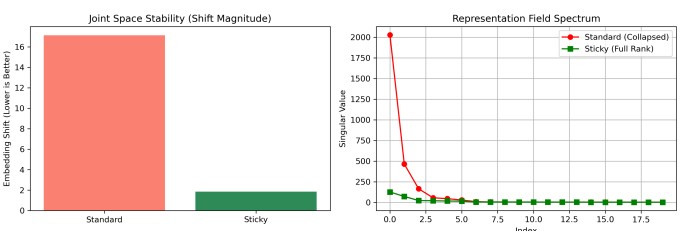

Figure 6: Manifold alignment stability and spectrum under missing-modality stress (seed = 1022, $z_{dim}$ = 256, N = 5000).

## 7 CONCLUSION

This research tackles a key challenge in AI: reconciling the empirical power of deep learning with theoretical insights into robustness and generalization (Appendix F.5). Instead of refining existing theories, it employs a novel mathematical framework from multi-scale geometric analysis used in the Kakeya set conjecture. The study introduces a unified geometric theory of representation for single- and multi-modal learning, formalizing concepts such as the representation field and the multi-scale Wolff axioms to quantify the geometric structure of learned features. The central finding, the Sticky Representation Theorem, links the geometric stability of features, or stickiness, to functional robustness. Additionally, it introduces the KT-CW Regularizer, a training objective that optimizes geometric stability, laying the groundwork for a geometric approach to deep learning and enhancing the reliability of AI systems.

## 8 ETHICS STATEMENT

This research is foundational and theoretical. It does not involve sensitive personal data or the training of models for applications that have direct societal consequences. The primary goal is to enhance the scientific understanding of AI robustness and reliability. The potential ethical implications are positive: by advancing more trustworthy AI, this work could help mitigate the risks of deploying fragile, unpredictable models in safety-critical applications.

## 9 REPRODUCIBILITY STATEMENT

All theoretical claims are presented with complete proofs.

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

# Appendices

## A  DECLARATION: USE OF LLMS

The use of LLMs was restricted to aiding or polishing writing, while human authors conceived all novel theoretical claims.

## B  LINKING "STICKINESS" TO STABILITY

This paper posits that the Kakeya conjecture framework emerging from (Wang & Zahl, 2025) extends beyond the realm of harmonic analysis. It offers a powerful perspective for viewing, analyzing, and ultimately designing hierarchical representation learning systems. A formal connection between these two fields is established by demonstrating that the principles governing the geometric structure of Kakeya sets closely resemble the principles that should guide the formation of robust feature hierarchies in deep neural networks.

The main idea of this work is that the geometric property of "stickiness," a term originating from the Kakeya literature that describes the structural integrity of a set of tubes across varying scales, serves as a formal, geometric analog to the desired attributes of robustness and stability in learned feature hierarchies. This analogy is further developed into a rigorous theoretical framework through the following conceptual bridge (Table 1).

Table 1: Conceptual bridge between the Kakeya conjecture in geometric measure theory and the proposed framework for representation learning.

| Kakeya Conjecture (Geometric Measure Theory) | Representation Learning (This Work) |
| --- | --- |
| Set of $\delta$-tubes | Set of feature activations (Representation Field) |
| Multi-scale analysis | Hierarchical feature extraction across layers |
| Grains decomposition | Data-dependent feature clustering (non-parametric attention) |
| Wolff Axioms (Non-clustering) | Geometric measures of feature sparsity & redundancy |
| "Stickiness" property | Stability of feature geometry across layers |
| Inductive Volume Estimate | Provable bound on generalization/robustness |

The translation of concepts, axioms, and theorems from the Kakeya conjecture into the language of machine learning facilitates the development of a novel geometric theory of representation. This framework allows for the quantification of the structure of learned features, the derivation of new theorems that establish connections between this structure and model robustness in both single- and multi-modal learning contexts, and the formulation of new potential principles for the design and training of deep neural networks.

# C    Related Work

## C.1    Theoretical Frameworks for Representation Learning

Several influential frameworks have been developed to formalize the characteristics of learned representations. Although these frameworks exhibit significant strengths, each is constrained by inherent limitations that underscore the necessity for a novel perspective.

### C.1.1    Geometric Deep Learning (GDL)

GDL (Bronstein et al., 2017) offers a methodology for integrating geometric priors, such as symmetries and invariances, directly into network architectures. By harnessing the mathematics of group theory, GDL enables the construction of equivariant networks in which feature representations transform predictably under various transformations of the input data (Gerken et al., 2023), such as rotation or translation. This approach has demonstrated significant effectiveness in domains where the underlying symmetries of the data are known and can be explicitly encoded.

However, the principal strength of GDL also constitutes its primary limitation: it is fundamentally prescriptive (Cao et al., 2020; Ye, 2022). It requires a priori knowledge of the geometric structure of the data to tailor the network architecture accordingly. This raises an important question regarding the numerous deep learning models, such as standard Transformers or ResNets, that are not explicitly designed with equivariance in mind. These models still show the capacity to learn powerful geometric biases implicitly from the data. In this context, GDL lacks the tools necessary for analyzing the emergent geometry present in such architectures.

In contrast, the proposed framework serves as a complementary and more generalized approach. Rather than mandating a network's geometry through its architecture, it provides a descriptive toolkit for analyzing the geometric properties of the representations that emerge from the learning process within any given network. This enables the characterization of the geometric structures that are learned implicitly, providing a more universal analytical framework for understanding deep learning representations.

### C.1.2    Wavelet Scattering Networks (WSNs)

WSNs (Bruna & Mallat, 2013; Gauthier et al., 2022; Shi et al., 2021) represent a significant advancement in the field of signal processing, demonstrating that multi-scale analysis, rooted in harmonic analysis, can yield architectures akin to those of deep learning models, endowed with provable stability guarantees. These networks are structured as a cascade of fixed wavelet transforms combined with non-linear modulus operators, leading to representations that are demonstrably stable against minor deformations and invariant to translations. This framework provides compelling evidence that a causal relationship exists between multi-scale geometric structure and robustness (Gao et al., 2021).

However, a notable limitation of WSNs is their lack of traditional learning capabilities, as they depend on a predetermined set of engineered wavelet filters (Liu et al., 2018). While this reliance on fixed filters affords strong theoretical assurances, it distances WSNs from the mainstream deep learning paradigm, which emphasizes the end-to-end learning of filters directly from data. Consequently, a critical question arises: can the stability properties achieved through engineering in WSNs also be replicated through optimization in learned networks?

The proposed Kakeya-based framework addresses this inquiry directly by offering analytical tools and a training objective, the KT-CW regularizer, designed for networks with fully learned filters. This framework facilitates the understanding, measurement, and promotion of the emergence of provable stability within the context of standard deep learning methodologies.

### C.1.3    The Information Bottleneck (IB)

The IB principle (Tishby et al., 2000; Tishby & Zaslavsky, 2015) provides a robust and abstract framework for representation learning, conceptualizing it as an optimal data compression process. The IB hypothesis asserts that an effective representation for a task should minimize the mutual information $I(X; Z)$ between the input variable $X$ and the representation $Z$ while maximizing the

mutual information $I(Y; Z)$ regarding the target variable $Y$ (Kawaguchi et al., 2023). This dual focus offers a coherent language for understanding generalization in machine learning.

The application of the IB principle to deep learning, however, remains contentious (Geiger & Kubin, 2020). Initial studies (Tishby et al., 2000; Tishby & Zaslavsky, 2015) indicated that deep networks experience a distinct "compression phase" during training, characterized by a reduction in $I(X; Z)$, and that this phase is causally related to improved generalization. Subsequent investigations have called these claims into question, suggesting that the observed compression may result from the specific neural nonlinearities employed (such as tanh vs. ReLU) or, more fundamentally, from the challenges associated with accurately estimating mutual information in high-dimensional spaces (Saxe et al., 2019). This ongoing debate underscores a significant limitation: while the IB principle offers theoretical appeal, it is predicated on quantities that are difficult to measure reliably, complicating the verification and extension of its claims.

To address this ambiguity, this research proposes a geometric framework that provides a more concrete and stable interpretation of the IB principle. It will be argued that the Wolff axioms, which quantify the geometric sparsity and non-redundancy of the representation space, function as a physical proxy for the abstract information-theoretic complexity term $I(X; Z)$. A representation field characterized by a low feature collapse constant is both geometrically sparse and non-redundant, serving as a direct, measurable indicator of a representation that is information-theoretically simple or compressed. This approach reframes the IB objective within a geometric framework, circumventing the contentious and unstable methods typically employed for mutual information estimation. Table 2 provides a comparison of the three frameworks alongside the proposed framework.

Table 2: Comparison of theoretical lenses for representation learning and robustness,

| Framework | Core Principle | Nature of Guarantees |
|---|---|---|
| GDL | Prescribe geometry (equivariance) via architecture. | Provable invariance/equivariance. |
| WSNs | Achieve stability via fixed multi-scale wavelet filters. | Provable stability to deformations. |
| IB | Learning as information-theoretic compression. | Theoretical generalization bounds. |
| **This Work** | **Analyze emergent geometry of learned features.** | **Provable robustness linked to geometric "stickiness."** |

## C.2 Provable Guarantees for Model Robustness

The pursuit of provable guarantees regarding model robustness constitutes a central theme in the field of theoretical deep learning (Kwiatkowska, 2020; Meng et al., 2022; Li et al., 2024). A significant line of investigation centers on Lipschitz analysis, which seeks to quantify the relationship between changes in a network's output and corresponding variations in its input (Virmaux & Scaman, 2018; Fazlyab et al., 2019; Gouk et al., 2021). A small Lipschitz constant indicates that minor perturbations in input will result in only modest changes in output, a characteristic associated with robust functions. However, deriving tight, scalable Lipschitz bounds for deep networks remains a challenge, and many existing bounds are too loose to be practically significant.

Alternative strategies include certification methods that deliver formal assurances against the existence of adversarial examples within a designated radius around a given input (Raghunathan et al., 2018; Ghiasi et al., 2020; Valentin, 2024). Although these methods possess considerable power, they are often computationally intensive and generally restricted to specific threat models (Anisetti et al., 2023), such as $L_p$-norm bounded perturbations (Liu et al., 2024a). The presented approach introduces a novel perspective through the Sticky Representation Theorem. This theorem provides a robustness guarantee that is not directly tied to the network's weights, architecture, or a specific threat model. Instead, it correlates the functional property of robustness with an intrinsic, measurable geometric property of the network's internal representations. This shift in focus from the function's parameters to the geometric structure of the learned spaces provides a more fundamental characterization of network robustness.

## C.3 Foundational Challenges in Multimodal Representation Learning

The second half of this paper extends the geometric theory to the multimodal setting, a domain of increasing practical relevance that poses its own unique foundational challenges. Research in this area is typically organized around three core problems: representation, alignment, and fusion. The representation problem focuses on learning effective representations for individual modalities (Guo et al.,

2019; Liang et al., 2022). The alignment problem seeks to identify and model the relationships and correspondences between elements from different modalities (Baltrušaitis et al., 2018; Liang et al., 2024). Finally, the fusion problem addresses how to combine information from multiple modalities to facilitate prediction or generation tasks Zhang et al. (2020); Zhao et al. (2024). A central challenge underlying these three problems is the heterogeneity gap (Lu, 2023), which recognizes that different modalities, such as images, text, and audio, exist in fundamentally distinct spaces with unique statistical properties.

A prevalent approach for bridging this gap is the implementation of joint embedding architectures (Balaneshin-Kordan & Kotov, 2018; Suzuki et al., 2016). These models are designed to map data from various modalities into a common, shared latent space where semantic similarity is expressed through proximity. Notable models, such as CLIP (Radford et al., 2021), have showcased the effectiveness of this approach, demonstrating impressive zero-shot capabilities and cross-modal retrieval functionality. Despite their empirical successes, the theoretical understanding of joint embedding spaces remains largely underdeveloped. The design of such systems is primarily informed by heuristics and empirical validation, with an absence of a foundational theory that characterizes the geometry of an effective joint space (Lu, 2023). Key questions remain, such as what geometric properties a robust multimodal representation should possess, how to formally measure the quality of alignment between modalities in the shared space, and whether provable guarantees can be provided regarding the robustness of models operating on these fused representations, especially when dealing with noisy or missing modalities.

This paper addresses this theoretical gap by extending the Kakeya-based framework to define geometric axioms for multimodal representations and proving a corresponding robustness theorem. This work contributes to the establishment of the formal geometric theory for this important domain, paving the way for further advancements in multimodal research.

# D  NOTATION SUMMARY

Table 3 provides a summary of key notation used throughout the paper. Symbols are grouped by origin in harmonic analysis, machine learning, or this work's proposed framework. Harmonic analysis terms capture multi-scale geometric properties, while machine learning terms describe inputs, network layers, and multimodal settings. The new terms formalize the geometric tools introduced in this framework.

Table 3: Notation summary

| Symbol | Field | Definition |
|---|---|---|
| $\delta, \rho$ | Harmonic Analysis | Scales, typically $0 < \delta \le \rho \le 1$. |
| $\mathcal{T}$ | Harmonic Analysis | A set of $\delta$-tubes in $\mathbb{R}^n$. |
| $C_{KT-CW}$ | Harmonic Analysis | Katz-Tao Convex Wolff constant, a measure of sparsity. |
| $C_{F-SW}$ | Harmonic Analysis | Frostman Slab Wolff constant, a measure of density. |
| $\sigma, \omega$ | Harmonic Analysis | Exponents in the Kakeya volume estimates (Assertions $D, E$). |
| Kernel | Harmonic Analysis | The integral kernels used to derive the volume estimates. |
| $f_l$ | Machine Learning | Function representing layer $l$ of a neural network. |
| $\mathcal{X}, \mathbf{Z}_l$ | Machine Learning | Input space and set of representations at layer $l$. |
| $M$ | Machine Learning | Number of modalities in a multimodal system. |
| $x^{(m)}$ | Machine Learning | Input data from modality $m$. |
| $g_{\Theta}^{(m)}$ | Machine Learning | Encoder network for modality $m$. |
| $Z_{\text{joint}}$ | Machine Learning | The shared Joint Embedding Space for multimodal data. |
| $\mathcal{T}_l$ | This Work | The Representation Field at layer $l$. |
| $\mathcal{T}_{\text{joint}}$ | This Work | The Joint Representation Field in a multimodal system. |
| $C_{\text{align}}$ | This Work | Cross-Modal Alignment Constant, a measure of geometric fusion quality. |
| $\mathcal{L}$ | Machine Learning | Lipschitz constant, a measure of stability. |
| $\mathcal{L}_{KT-CW}$ | This Work | The proposed KT-CW regularizer. |
| $\nu$ | This Work | Deformation operator. |

# E ADDITIONAL RESULTS

## E.1 A GEOMETRIC PERSPECTIVE ON MODE COLLAPSE IN GANS

Mode collapse in Generative Adversarial Networks (GANs) represents a significant training failure, where the generator restricts its output variety, thereby failing to capture the diversity inherent in the true data distribution adequately (Barsha & Eberle, 2025; Saxena & Cao, 2021; Jabbar et al., 2021). This work proposes a formal definition of mode collapse as a geometric pathology characterized by an excessively large Feature Collapse Constant, $C_{KT-CW}$, within the generator's intermediate representation fields.

In this context, a generator network ($G$) transforms latent vectors ($z$) into outputs ($y$) via a sequence of layers: $G = f_L \circ \ldots \circ f_1$. The manifestation of mode collapse in the output space indicates that a broad array of latent codes converges into a constricted, clustered region within one of the final feature spaces, specifically $Z_{L-1}$. This geometric clustering is quantitatively captured by a large $C_{KT-CW}$, which serves as a formal and measurable indication of this failure mode.

This insight suggests that applying the KT-CW regularizer to the layers of the generator may serve as a principled mechanism for countering mode collapse, thereby promoting geometric diversity within the learned feature manifold.

## E.2 THE LOTTERY TICKET HYPOTHESIS: A SEARCH FOR STICKY SUBNETWORKS

The Lottery Ticket Hypothesis (LTH) posits that dense, randomly initialized neural networks contain sparse subnetworks, termed "winning tickets," which can be trained to achieve performance levels comparable to those of the full network (Frankle & Carbin, 2018; Frankle et al., 2019). This phenomenon is attributed to what is referred to as "fortuitous initialization." The LTH framework offers a concrete geometric interpretation, suggesting that winning tickets are subnetworks that present inherent $K$-stickiness at the time of initialization.

A $K$-sticky subnetwork is characterized by its initial weights, which define a function with well-structured, non-collapsing representation fields, indicated by low $C_{KT-CW}$ values across layers. This property provides a stable geometric scaffold conducive to the learning process. The iterative magnitude pruning (IMP) technique (Malach et al., 2020) employed to identify these winning tickets can be reinterpreted as a search algorithm that implicitly optimizes for $K$-sticky subnetworks by eliminating weights that contribute to redundant features, which are geometrically collapsed. This connection between the empirical efficacy of LTH and the foundational principles of geometric robustness is further substantiated by research (Liu et al., 2024b) demonstrating that winning tickets possess robust graph-theoretic properties, such as being classified as good expanders.

## E.3 REMARK 2. NATURAL GRADIENTS AND THE GEOMETRY OF THE PARAMETER SPACE

The optimization of neural networks via gradient descent is conventionally conducted within a Euclidean parameter space. However, principles from information geometry indicate that the space of model parameters ($\Theta$) possesses a Riemannian structure characterized by the Fisher Information Matrix (FIM), $F(\theta)$. The FIM quantifies the sensitivity of the model's output distribution to variations in its parameters. Natural Gradient Descent (NGD) is an optimization algorithm that adheres to the steepest descent direction on this Riemannian manifold, with updates formulated as:

$$\theta_{t+1} = \theta_t - \eta F(\theta_t)^{-1} \nabla \mathcal{L}(\theta_t). \tag{11}$$

This approach often leads to faster convergence by taking steps of equal "information" size, rather than steps of equal size in the Euclidean parameter space.

This work introduces a complementary geometric perspective. While NGD emphasizes the intrinsic geometry of the parameter space $\Theta$, the proposed framework, particularly through the application of the KT-CW regularizer, concentrates on the extrinsic geometry of the representation space $\mathbf{Z}_l$. These two approaches, while related, are distinct: the parameters $\theta$ specify the function $f_\theta$, which subsequently generates the representation field $\mathcal{T}_l$. The optimization process utilizing the KT-CW regularizer strategically directs the search within the parameter space toward regions that yield geometrically stable representations. This suggests a powerful synthesis: one could employ NGD to

navigate the parameter manifold more effectively, while using the KT-CW regularizer to ensure the path leads to solutions that are not only optimal with respect to the loss function but also possess provably robust geometric properties in their representations.

### E.4 PROPOSITION 2. GEOMETRIC COMPLEXITY AND PAC-BAYES GENERALIZATION BOUNDS

The PAC-Bayes framework provides high-probability bounds on the generalization error of a randomized predictor (a posterior distribution $Q$ over hypotheses) in terms of its empirical risk and its Kullback-Leibler (KL) divergence from a data-independent prior distribution $P$. A typical bound takes the form:

$$R(Q) \leq \hat{R}(Q) + \sqrt{\frac{KL(Q||P) + ln(\frac{n}{\delta})}{2n}}, \tag{12}$$

where $R(Q)$ is the true risk and $\hat{R}(Q)$ is the empirical risk. The term $KL(Q||P)$ serves as a measure of complexity or "information cost" for learning the posterior $Q$ from the prior $P$. A central challenge is selecting a prior $P$ that minimizes this term while enabling $Q$ to achieve a low empirical risk.

The geometric complexity of a representation, as measured by the Feature Collapse Constant $C_{KT-CW}$, can be formally related to the complexity term in a PAC-Bayes bound. Specifically, a low $C_{KT-CW}$ of the representation fields generated by hypotheses sampled from a posterior $Q$ implies a "simpler" posterior that can be described with a smaller KL divergence from a suitable prior.

The intuition is that a $K$-sticky network (low $C_{KT-CW}$) produces non-redundant, geometrically sparse representations. This structural simplicity in the representation space translates to a lower effective complexity of the function class from which the posterior $Q$ is drawn. A prior $P$ can be constructed to favor functions that produce such geometrically simple representations. Consequently, a posterior $Q$ learned by optimizing for stickiness (e.g., using the KT-CW regularizer) will naturally remain close to this prior, resulting in a small $KL(Q||P)$ and a tighter generalization bound. This proposition formalizes the connection between the geometric simplicity of learned features and the information-theoretic simplicity required for guaranteed generalization. The full proof for this proposition is provided in Appendix F.5.

### E.5 REMARK 3. MANIFOLD STICKINESS AND ADVERSARIAL ROBUSTNESS

The manifold hypothesis asserts that high-dimensional data, such as images, reside on or near a low-dimensional ambient space. Adversarial vulnerability can be examined through this perspective: an adversarial example is generated by a small perturbation that displaces a data point off its manifold and across the decision boundary of a classifier. These perturbations can be dissected into two distinct components: tangential (in-manifold) and normal (off-manifold) to the data manifold. Research has demonstrated that perturbations normal to the manifold are often particularly effective in crafting adversarial attacks, as they tend to represent the most direct path to the decision boundary (Lin et al., 2020; Han et al., 2023).

This work offers a compelling explanation for robustness against adversarial attacks. A $K$-sticky representation refers to a learned data manifold that maintains structural stability and resists local collapse. A high Feature Collapse Constant ($C_{KT-CW}$) indicates the existence of "geometric bottlenecks," where the manifold is excessively compressed or folded, resulting in regions that can be considered fragile. In such areas, even minor perturbations normal to the manifold can easily navigate through collapsed regions and breach the decision boundary. In contrast, a $K$-sticky network, characterized by a uniformly low $C_{KT-CW}$, learns a "well-behaved" manifold that lacks these critical bottlenecks. This preservation of geometric structure implies that a perturbation of a specified magnitude in the normal direction leads to a correspondingly minor displacement on the manifold, thereby reducing the likelihood of crossing a decision boundary. Consequently, $K$-stickiness serves as a direct metric for evaluating the resilience of the learned manifold to the normal-direction perturbations, which are important for enhancing adversarial robustness.

# F  PROOFS

## F.1  PROOF FOR LEMMA 1

*Statement.* Fix a network layer $l$. Let $Z_l \subset \mathbb{R}^{d_l}$ be the set of activations and, for a scale $\delta \in (0, 1]$, define the representation field $T_l(\delta) = \{T_z : z \in Z_l\}$, where each $T_z$ is the $\delta$-tube given by the $\delta$-neighbourhood of the line segment $[0, z]$ with axis direction $\xi(z) := \frac{z}{\|z\|}$ and length $\|z\|$. Fix any intermediate scale $\rho$ with $\delta \ll \rho \leq 1$ and an angular threshold $\vartheta \in (0, 1)$.

Then there exists a finite family of grains $\mathcal{G} = \{(G_k, v_k)\}_{k=1}^K$ where each $G_k \subset \mathbb{R}^{d_l}$ is a rectangular prism of longitudinal length $\asymp 1$ and cross-sectional diameter $\asymp \rho$, and $v_k \in \mathbb{S}^{d_l - 1}$ is its axis direction, such that the following hold:

  (i) Converge and near-disjointness: The grains are essentially disjoint and cover a positive-measure portion of $\bigcup_{T \in T_l(\delta)} T$ at scale $\rho$ (the uncovered part is a negligible boundary layer at scale $\rho$).

  (ii) Coherence inside grains: If a tube $T_z \in T_l(\delta)$ meets $G_k$ in a longitudinal segment of length $\gtrsim \rho$, then $\angle(\xi(z), v_k) \leq \vartheta$.

  (iii) Bounded grain-incidence: Each tube $T_z$ intersects at most $C\rho^{-\beta}$ grains, for some dimension-dependent $C, \beta > 0$.

Define the grain cluster

$$S_k := \{z \in Z_l : T_z \cap G_k \text{ contains a segment of length } \geq \rho, \angle(\xi(z), v_k) \leq \vartheta\}. \tag{13}$$

Then $\{S_k\}_{k=1}^K$ is a data-dependent, unsupervised geometric clustering of the features at layer $l$: for any $z, z' \in S_k$,

$$\angle(\xi(z), \xi(z')) \leq 2\vartheta, \quad \mathrm{diam}(\{x \in T_z \cap G_k\} \cup \{x' \in T_{z'} \cap G_k\}) \lesssim \rho, \tag{14}$$

so members of a cluster are co-linear (directionally coherent) and spatially proximate at scale $\rho$. Moreover, each grain admits a canonical meta-feature summary (e.g., the principal axis $v_k$ together with a robust centroid of $\{z : z \in S_k\}$), yielding a non-parametric, data-dependent "meta-feature" for layer $l$.

*Proof.* For a unit vector $u$, write $\angle(u, u')$ for the geodesic angle on $\mathbb{S}^{d_l - 1}$. For a rectangular prism $G$ with axis $v$, we call longitudinal any direction within an angle $\leq \vartheta$ of $v$ and transverse the orthogonal directions. Constants implicit in $\lesssim, \asymp$ depend only on the ambient dimension and harmless absolute choices.

Recent advances in the Kakeya conjecture demonstrate that tube configurations can be decomposed at an intermediate scale $\rho$ into grains, which are rectangular prisms of transverse thickness $\sim \rho$ and unit-scale length, obeying coverage, near-disjointness, and incidence properties. Concretely, for a set of $\delta$-tubes obeying quantitative Wolff-type non-clustering hypotheses, one can construct a two-scale ("$\delta \ll \rho$") grains decomposition with properties (i)-(iii) above and further quantitative regularity, see Wang & Zahl (2025)'s structural decomposition and its exposition (the "maximal grains"/two-scale grain structure) and standard summaries of the grains toolkit. For didactic statements of the typical properties (near-jointness, almost-full coverage, and bounds on tube-grain incidences), see also Fisher (2018) on polynomial/grain decompositions. We then obtain $\mathcal{G} = \{(G_k, v_k)\}_{k=1}^K$ satisfying (i)-(iii).

Define a measurable assignment map $a : Z_l \to \{1, \ldots, K\} \cup \{\varnothing\}$ by:

  (i) $a(z) = k$ if $T_z$ intersects $G_k$ in a longitudinal segment of length $\geq c\rho$ and $\angle(\xi(z), v_k) \leq \vartheta$.

  (ii) If multiple grains satisfy this, choose any fixed tie-breaker (e.g., lexicographic on $k$ or the grain with maximal intersection length).

  (iii) If no grain qualifies, set $a(z) = \varnothing$.

Define clusters $S_k := a^{-1}(k)$. By construction, the clusters use no labels and depend only on $\{T_z\}$ and the geometric parameters $(\delta, \rho, \vartheta)$, i.e., an unsupervised, data-dependent rule.

If $z, z' \in S_k$, then $\angle (\xi(z), v_k) \leq \vartheta$ and $\angle (\xi(z'), v_k) \leq \vartheta$. By the triangle inequality on $\mathbb{S}^{d_l-1}$,

$$\angle (\xi(z), \xi(z')) \leq \angle (\xi(z), v_k) + \angle (v_k, \xi(z')) \leq 2\vartheta. \tag{15}$$

Thus the set $\{\xi(z) : z \in S_k\}$ lies in a spherical cap of radius $2\vartheta$. This formalizes co-linearity (directional coherence) claimed in the lemma.

Because $T_z \cap G_k$ contains a longitudinal segment of length $\gtrsim \rho$ and $G_k$ has transverse diameter $\asymp \rho$, the axial lines of $T_z$ and $T_{z'}$ both pass through a common $\rho$-ball inside $G_k$. Therefore, any points $x \in T_z \cap G_k$, $x' \in T_{z'} \cap G_k$ satisfy $\|x - x'\| \lesssim \rho$, establishing spatial proximity within each cluster.

Formally, write $G_k = \left\{ x : |\langle x - c_k, v_k \rangle| \leq L/2, \left\| P_{v_k^\perp} (x - c_k) \right\| \leq C\rho \right\}$. If $T_z \cap G_k$ contains a longitudinal segment of length $\geq c\rho$ with $\angle (\xi(z), v_k) \leq \vartheta$, then there exists a parameter $t$ with $|t| \leq C'\rho$ such that $c_k + tv_k \in T_z$ (up to $O(\delta)$ which is negligible since $\delta \ll \rho$ ). The same holds for $z'$. Hence, both tubes meet a common $\rho$-neighborhood of $c_k$, giving the desired bound.

By property (iii), each tube meets at most $C\rho^{-\beta}$ grains. Our tie-breaking ensures that $a$ is single-valued, except on a negligible boundary set (where intersection lengths are equal up to lower-order errors). Thus, almost every $z$ with $T_z$ entering the $\rho$-interior of $\bigcup_k G_k$ is assigned to exactly one cluster. Property (i) ensures that the unassigned activations (tubes only grazing grain boundaries) form a set whose contribution vanishes as the grain boundary thickness is shrunk at fixed $\rho$.

Consider the following threshold-based clustering objective at scale $\rho$ and angle $\vartheta$:

Find a partition $Z_l = \bigsqcup_k S_k$ and representatives:

$$(v_k, c_k) \quad \text{s.t.} \quad \begin{cases} \angle (\xi(z), v_k) \leq \vartheta & \forall z \in S_k, \\ \mathrm{dist} (T_z, \text{the line} c_k + \mathbb{R} v_k) \lesssim_\rho^{(\star)} & \forall z \in S_k. \end{cases} \tag{16}$$

The grains construction produces such a partition and representatives ( $v_k, c_k$ ) (the grain axis and center), hence it is a valid solution to ( $\star$ ). Conversely, any solution to ( $\star$ ) induces a cover by rectangular prisms of longitudinal length $\asymp 1$ and transverse diameter $\asymp \rho$ around the lines $c_k + \mathbb{R} v_k$, i.e., a grains-like cover at scale $\rho$. Therefore, grains decomposition $\iff$ geometric clustering at scale $\rho$ in the precise thresholded sense of ( $\star$ ). (This is the standard interpretation of grains as a structural partition capturing directional and spatial coherence; see Wang & Zahl (2025)).

For each grain cluster $S_k$, define a meta-feature as any measurable summary functional that depends only on the cluster's geometry; two canonical choices are:

(i) The principal direction $u_k \in \mathbb{S}^{d_i - 1}$ solving $u_k = \arg\min_{u \in \mathbb{S}_{l-1}^d} \sum_{z \in S_k} \angle (\xi(z), u)^2$ (first principal component on the sphere).

(ii) An aggregate endpoint $m_k := \frac{1}{|S_k|} \sum_{z \in S_k} z$ or a robust median/trimmed mean.

By previous steps, $\{\xi(z) : z \in S_k\}$ lies in a spherical cap of radius $2\vartheta$, so $u_k$ is well-defined and satisfies $\angle (u_k, v_k) \leq 2\vartheta$. By Step 4, the tube axes pass through a common $\rho$-neighborhood, so $m_k$ is stable at scale $\rho$. Thus, each grain admits a stable, data-dependent meta-feature that summarizes a coherent motif of layer-$l$ activations (direction and location/scale). This matches the qualitative role of grains as "structure packets" in the modern Kakeya theory (rectangular cells within which tubes are well-aligned and well-localized) now instantiated on $T_l(\delta)$. This completed the proof. ∎

## F.2 PROOF FOR THEOREM 1

*Statement.* Let $f = f_L \circ \cdots \circ f_1 : \mathcal{X} \to \mathbb{R}^{d_L}$ be a (piecewise $C^1$ ) hierarchical representation. For each layer $l$, let $\mathbf{Z}_l = \left\{ z_i^{(l)} \right\} \subset \mathbb{R}^{d_l}$ be the set of activations produced by a fixed dataset $\mathcal{D} \subset \mathcal{X}$, and for a scale $\delta \in (0, 1]$ define the representation field $\mathcal{T}_l(\delta) = \{T_z : z \in \mathbf{Z}_l\}$ where $T_z$ is the $\delta$-tube around the segment $[0, z]$ with axis $\xi(z) = \frac{z}{\|z\|}$. Assume there exists $K \geq 1$ such that each $\mathcal{T}_l(\delta)$ satisfies the Katz-Tao convex Wolff axiom with constant $C_{KT-CW} (\mathcal{T}_l(\delta)) \leq K$ (uniformly in $l$ ). Let $\mathcal{V}$ be a class of $C^1$ input deformations acting by flows $(\nu_s)_{s \in [0,1]}$ generated by bounded vector

fields $V$ (i.e., $\frac{d}{ds}\nu_s(x) = V(\nu_s(x))$, with $\|V\|_\infty \leq 1$ ), and define

$$\mathrm{dist}_\mathcal{V}(\nu) := \inf\left\{ \int_0^1 \|V\|_\infty ds : \nu_1 = \nu, \nu_0 = \mathrm{id}\right\}. \tag{17}$$

Then there exists a monotone increasing function $g : [1, \infty) \to (0, \infty)$ (depending only on fixed layerwise Lipschitz budgets and the choice of $\delta$ ) such that

$$\|f(\nu(x)) - f(x)\| \leq g(K)\,\mathrm{dist}_\mathcal{V}(\nu) \quad \text{for all } x \in \mathcal{D}, \nu \in \mathcal{V}. \tag{18}$$

Equivalently, the Lipschitz constant against $\mathcal{V}$ satisfies $\mathcal{L}_\mathcal{V}(f) \leq g(K)$.

*Proof*[4]. By the convex Wolff axioms with constant $K$, any collection of $\delta$-tubes obeys quantitative non-clustering constraints (at all intermediate scales $\rho$): not too many tubes can lie inside any convex set $W$, and not too many near any low-degree algebraic variety. From these axioms one obtains a two-scale grains decomposition: a cover by rectangular "grains" $G$ of longitudinal size $\asymp 1$ and transverse diameter $\asymp \rho$, which are essentially disjoint, count almost all tube mass, and are met by any single tube only $O\left(\rho^{-\beta}\right)$ times (for some $\beta > 0$). Moreover, tube families satisfying the axioms enjoy volume lower bounds for their unions at all scales (the Kakeya-type "Assertion D/E" estimates): unions cannot concentrate into sets much smaller than what the axioms allow. These facts are standard outcomes of the modern Kakeya machinery (Wolff axioms $\to$ grains $\to$ volume lower bounds). For recent expositions emphasizing the role of grains and the Katz-Tao convex Wolff axioms, see Katz & Tao (2000) and surveys following Wang & Zahl (2022). For the "sticky" regime and self-similar structure that underlies sharp volume bounds, see Wang & Zahl (2025) and follow-ups. We will only use two quantitative consequences at each layer $l$ from the Kakeya conjecture:

(i) Bounded tube occupancy in convex sets: For every convex $W \subset \mathbb{R}^{d_i}$,

$$\#\left\{T \in \mathcal{T}_l(\delta) : T \subset W\right\} \leq K \cdot \Phi_l(W, \delta), \tag{19}$$

where $\Phi_l(W, \delta)$ is the model-dependent scale factor comparable to $|W|\delta^{1-d_l}$ for standard tube families. Intuitively, no small convex region can contain too many tubes.

(ii) Grains: For each intermediate $\rho$ with $\delta \ll \rho \leq 1$, there is a grains cover $\{G\}$ with: (a) near-disjointness up to boundary; (b) almost full-converage of $\bigcup \mathcal{T}_l(\delta)$; (c) each tube meets $O(\rho^{-\beta})$ grains; and (d) inside a grain, intersecting tubes are directionally coherent, where the axes lie in a spherical cap of radius $O(1)$.

Fix a differentiable map $F : \mathbb{R}^d \to \mathbb{R}^{d'}$ and a $\delta$-tube $T$ with axis direction $u$. Let $J_F(x)$ denote the Jacobian. For $x$ in the core of $T$ and for small $h > 0$,

$$\frac{F(x + hu) - F(x)}{h} = J_F(x)u + o(1). \tag{20}$$

Thus, on each short axial segment of $T$, $F$ maps that segment to a segment of direction close to $\frac{J_F(x)u}{\|J_F(x)u\|}$, within a transverse error $\lesssim \|J_F\|_{\mathrm{Lip}}\, h^2$. Consequently, for $\delta$ sufficiently small and $F$ locally Lipschitz, the image $F(T)$ is contained in a $\delta'$-tube $T'$ whose axis is $u' := \frac{J_F(\bar{x})u}{\|J_F(\bar{x})u\|}$ for some $\bar{x}$ on the axis of $T$, with

$$\delta' \lesssim \|J_F\|_\infty\, \delta, \quad \mathrm{len}\,(T') \asymp \|J_F(\bar{x})u\|\,\mathrm{len}(T) \tag{21}$$

For piecewise $C^1$ networks, the above holds on each smooth patch. By Rademacher's theorem, almost every point is differentiable, so the conclusion holds for almost all tubes. We refer to this as the tube push-forward under $F$. Intuition is that the Jacobian pushes forward the axis direction; cross-section scales by $\|J_F\|_\infty$. This is the standard differential-geometric approximation used in all wave-packet/tube arguments underlying Kakeya and restriction theory. Apply this to $F = f_l$ acting on $\mathcal{T}_{l-1}(\delta)$ to obtain a family $\mathcal{T}_l^{\mathrm{im}}$ of $\delta'$-tubes contained in $\mathcal{T}_l(\delta')$ (modulo $O(\delta')$ boundary error).

---

[4]Throughout this proof, constants implicit in $\lesssim, \gtrsim, \asymp$ depend only on the ambient dimensions and on the fixed scale regime $\delta \ll \rho \leq 1$.

Fix a layer $l$. Let $x \in \mathcal{D}$ and let $u$ be a unit direction in input to this layer (i.e., a direction in $\mathbb{R}^{d_{l-1}}$). Consider a short axial segment of length $h$ in the pre-image representation (a micro-tube), and push it through $f_l$. By Equation (21), its image is contained in a $\delta'$-tube $T'$ whose axial length is $\asymp \|J_{f_i}(x)u\| h$.

Suppose, for contradiction, that for a positive-density set of $(x, u)$, one has a large directional gain:

$$\|J_{f_l}(x)u\| \geq A, \tag{22}$$

with $A \gg 1$. Form a finite family $\mathcal{U}$ of such input micro-tubes, arranged in parallel stacks indexed by $x$ and $u$, with disjoint interiors in the domain (standard Vitali selection). Pushing them forward yields a family $\mathcal{U}'$ of output $\delta'$-tubes whose axes are confined to small spherical caps because $u$ was fixed in each stack and $J_{f_l}$ varies slowly on the small input segments, and whose lengths are $\gtrsim Ah$.

Now place a convex capturing set $W$ in the output layer: let $W$ be a rectangular prism of longitudinal size $\asymp Ah$ aligned with the common output axis and of transverse diameter $C\delta'$. By construction, each tube $T' \in \mathcal{U}'$ from a given stack is contained in $W$ (up to negligible edge effects). Hence, for each such stack,

$$\#\{T' \in \mathcal{U}' : T' \subset W\} \asymp \#\{\text{ input micro-tubes in the stack }\}. \tag{23}$$

Choosing enough stacks (still within a region of controlled size) produces many output tubes contained in the same convex $W$. By property (i) (Equation (19)), this cannot exceed $K \cdot \Phi_l(W, \delta')$, i.e.,

$$\#\{T' \subset W\} \leq K \cdot \Phi_l(W, \delta') \asymp K \cdot \frac{|W|}{(\delta')^{d_l - 1}} \asymp K \cdot \frac{(Ah)(C\delta')^{d_l - 1}}{(\delta')^{d_l - 1}} \asymp K \cdot Ah. \tag{24}$$

But the left-hand side scales like the number of selected micro-tubes, which we can make $\gg K \cdot Ah$ by (a) taking a dense enough Vitali packing in input, (b) using the fact that $A$ is fixed while the number of disjoint micro-tubes in a fixed region can be taken arbitrarily large as $\delta \downarrow 0$. This contradiction shows that Equation (24) cannot hold with arbitrarily large $A$. Therefore, there exists a layerwise bound:

$$\|J_{f_l}(x)u\| \leq c_l(K) \quad \text{for a.e. } x \text{ and all unit } u, \tag{25}$$

where $c_l(K)$ is a monotone increasing function of $K$ that depends only on fixed scale budgets and local regularity.

**Remark 4 (Counting vs. Volume).** If directional amplification were too large on many micro-tubes, their images would create too many tubes inside one convex box, violating the convex Wolff occupancy bound. This is the same logic that produces quantitative volume lower bounds for unions of tubes from Wolff-type axioms and the grains structure. Directional coherence inside grains (ii) is used implicitly to align the output axes so that a single convex $W$ captures many tubes; the Vitali selection and two-scale control are standard in grains-based arguments. Hence, each layer $f_l$ is uniformly Lipschitz on the data manifold in all directions, with

$$\|J_{f_l}(x)\|_{\mathrm{op}} \leq c_l(K) \quad \text{a.e. } x. \tag{26}$$

Let $(\nu_s)_{s \in [0,1]} \subset \mathcal{V}$ be a deformation flow with generator $V$ ($\|V\|_\infty \leq 1$). Define the layer-$l$ trajectory

$$z_l(s) := f_l \circ f_{l-1} \circ \cdots \circ f_1(\nu_s(x)) \in \mathbb{R}^{d_l}. \tag{27}$$

By the chain rule and Rademacher's theorem (applied a.e. in $s$),

$$\frac{d}{ds} z_l(s) = J_{f_l}(z_{l-1}(s)) \cdots J_{f_1}(\nu_s(x)) V(\nu_s(x)). \tag{28}$$

Using Equation (26) layerwise,

$$\left\| \frac{d}{ds} z_L(s) \right\| \leq \prod_{l=1}^{L} c_l(K) \cdot \|V\|_\infty \leq \left( \prod_{l=1}^{L} c_l(K) \right) \tag{29}$$

Integrating from $s = 0$ to 1 yields

$$\|f(\nu(x)) - f(x)\| \leq \left(\prod_{l=1}^{L} c_l(K)\right) \cdot \int_0^1 \|V\|_\infty ds. \tag{30}$$

Taking the infimum over all generating flows of $\tau$ gives

$$\|f(\nu(x)) - f(x)\| \leq g(K) \operatorname{dist}_{\mathcal{V}}(\nu), \quad g(K) := \prod_{l=1}^{L} c_l(K), \tag{31}$$

and $g$ is monotone increasing in $K$ because each $c_l$ is. This completed the proof. ∎

### F.3 PROOF FOR THEOREM 2

*Statement.* Let there be $M$ modalities with inputs $x^{(m)} \in \mathcal{X}^{(m)}$ and encoders $g^{(m)} = g_{L_m}^{(m)} \circ \cdots \circ g_1^{(m)}$. Let a fusion network $h = h_L \circ \cdots \circ h_1$ map the concatenated (or otherwise fused) hidden states to the task output, and denote the overall representation by

$$f\left(x^{(1)}, \ldots, x^{(M)}\right) = (h_L \circ \cdots \circ h_1)\left(g^{(1)}\left(x^{(1)}\right), \ldots, g^{(M)}\left(x^{(M)}\right)\right). \tag{32}$$

At layer $\ell$, for each modality $m$ let $\mathbf{Z}_\ell^{(m)} \subset \mathbb{R}_\ell^{d^{(m)}}$ be the set of activations on a fixed dataset $\mathcal{D}$, and define the per-modality representation field

$$\mathcal{T}_\ell^{(m)}(\delta) = \left\{T_z : z \in \mathbf{Z}_\ell^{(m)}\right\}, \quad T_z = \text{ the } \delta\text{-tube around } [0, z], \text{ axis } \xi(z) = \frac{z}{\|z\|}. \tag{33}$$

Let $Z_{\ell, \text{joint}}$ denote the joint embedding at layer $\ell$ after fusion at that depth, and define the joint representation field

$$\mathcal{T}_{\ell, \text{joint}}(\delta) = \bigcup_{m=1}^{M} \iota_m\left(\mathcal{T}_\ell^{(m)}(\delta)\right), \tag{34}$$

where $L_m$ embeds modality-$m$ tubes into the joint space.

Assume multimodal $K$-stickiness: for every $\ell$,

$$C_{KT-CW}\left(\mathcal{T}_\ell^{(m)}(\delta)\right) \leq K \quad \forall m, \quad C_{\text{align}}\left(\mathcal{T}_{\ell, \text{joint}}(\delta)\right) \leq K. \tag{35}$$

Here, $C_{KT-CW}$ is a Katz-Tao convex Wolff-type non-clustering constant (occupancy bound in convex sets), and $C_{\text{align}}$ is a cross-modal alignment constant that controls the variance across modalities of tube occupancy inside any convex set in the joint space (well-mixed modalities).

Let $\mathcal{V}$ be a class of input deformations acting independently on each modality via flows $\nu_s^{(m)}$ generated by bounded vector fields $V^{(m)}$ (with $\left\|V^{(m)}\right\|_\infty \leq 1$), and equip $\mathcal{X}^{(1)} \times \cdots \times \mathcal{X}^{(M)}$ with the modal deformation distance

$$\operatorname{dist}_{\mathcal{V}}(\nu) := \sum_{m=1}^{M} \inf\left\{\int_0^1 \left\|V_s^{(m)}\right\|_\infty ds : \nu_1^{(m)} = \nu^{(m)}, \nu_0^{(m)} = \operatorname{id}\right\}. \tag{36}$$

*Claim.* There exist monotone increasing functions $g_1, g_2 : [1, \infty) \to (0, \infty)$ depending only on fixed layerwise regularity and the scale regime $\delta \ll \rho \leq 1$ such that:

(i) Lipschitz stability to deformations: For all $\nu \in \mathcal{V}$ and $x \in D$,

$$\|f(\nu(x)) - f(x)\| \leq g_1(K) \operatorname{dist}_{\mathcal{V}}(\nu). \tag{37}$$

(ii) Controlled degradation under modality drop: For any subset $S \subset \{1, \ldots, M\}$, define $\text{drop}_S$ that replaces $x^{(m)}$ by a fixed null anchor $x_{\varnothing}^{(m)}$ for $m \in S$ and leaves other modalities unchanged. Then,

$$\|f(\text{drop}_S(x)) - f(x)\| \leq g_2(K) \frac{|S|}{M}, \tag{38}$$

with the fraction $\frac{|S|}{M}$ written w.r.t. the modal deformation distance (i.e., each dropped modality contributes unit path length from $x^{(m)}$ to $x_{\varnothing}^{(m)}$). If one prefers a different path length for each modality, the factor scales accordingly.

*Proof.* From $C_{KT-CW}\left(\mathcal{T}_{\ell}^{(m)}\right) \leq K$ we have (per modality, per layer) the standard convex occupancy bound: no convex set $W \subset \mathbb{R}^{d_{\ell}^{(m)}}$ can contain more than $O_K\left(\Phi_{\ell}^{(m)}(W, \delta)\right)$ tubes (here, $\Phi$ is the model scale factor comparable to $|W|\delta^{1-d_{\ell}^{(m)}}$). Moreover, for each intermediate scale $\rho$ with $\delta \ll \rho \leq 1$, one obtains a two-scale grains decomposition: a near-disjoint cover by rectangular grains $G$ of transverse diameter $\asymp \rho$ and longitudinal size $\asymp 1$, meeting the usual properties-almost full coverage of $\bigcup \mathcal{T}$, bounded tube-grain incidences, and directional coherence inside grains. These are the workhorses of the modern Kakeya machinery (Wolff axioms $\Rightarrow$ grains $\Rightarrow$ volume/occupancy control). The cross-modal alignment bound $C_{\text{align}}\left(\mathcal{T}_{\ell, \text{joint}}\right) \leq K$ adds: for every convex $W \subset Z_{\ell, \text{joint}}$, if we denote by $N_m(W)$ the number or soft count of modality-$m$ tubes whose $\rho$-length core lies in $W$, then the normalized variance across modalities satisfies

$$\text{Var}_m[N_m(W)] \leq K \cdot \Psi_{\ell}(W, \delta), \tag{39}$$

for an appropriate scale factor $\Psi_{\ell}$ (one may equivalently impose a bounded modality imbalance $\max_m N_m(W) - \min_m N_m(W) \lesssim K\Psi_{\ell}$). Intuitively, a single modality can dominate no convex joint-space cell, nor can the modalities be highly uneven there, a quantitative "well-mixed" condition.

As in the single-modal case, by Rademacher's theorem and the chain rule, each (piecewise $C^1$) layer acts on a short axial segment as an affine map with a Jacobian given by the local differential. Thus, a $\delta$-tube is mapped into a $\delta'$-tube whose axis direction is the push-forward of the original axis by the Jacobian, with $\delta' \lesssim \|J\|_{\infty}\delta$ and axial length scaled by $\|Ju\|$ for the local direction $u$. This "ube push-forward lemma" is standard in wave-packet/tube arguments underpinning Kakeya-restriction theory. For a fusion layer $h_{\ell}$ acting on the concatenated state $(z^{(1)}, \ldots, z^{(M)})$, write its Jacobian as blocks

$$J_{h_{\ell}} = \begin{bmatrix} J_{\ell}^{(1)} & \cdots & J_{\ell}^{(M)} \end{bmatrix}, \quad J_{\ell}^{(m)} = \frac{\partial h_{\ell}}{\partial z^{(m)}}. \tag{40}$$

Under push-forward, a collection of per-modality micro-tubes with directions $u^{(m)}$ maps to joint-space micro-tubes with axial direction proportional to $\sum_m J_{\ell}^{(m)} u^{(m)}$.

Fix a modality $m$ and layer $\ell$. Suppose, toward contradiction, that there is a positive-density set of points/directions $(x^{(m)}, u^{(m)})$ with large directional gain

$$\left\|J_{\ell}^{(m)}(x)u^{(m)}\right\| \geq A \quad (A \gg 1). \tag{41}$$

Select a Vitali family of disjoint input micro-tubes in modality $m$ realizing Equation (41), and push them through to the $\ell$-th joint space. As in the single-modal proof, choose a convex capturing prism $W$ aligned with the common output direction so that all images of those micro-tubes lie in $W$. For each fixed stack of parallel micro-tubes, the number of output tubes contained in $W$ is $\asymp$ the number of input micro-tubes in the stack; by taking enough stacks (keeping the total domain bounded), the count inside $W$ becomes $\gg_K, A$ times the scale allowance.

But the modality-$m$ occupancy in any convex set in its own space is bounded by $C_{KT-CW}\left(\mathcal{T}_{\ell}^{(m)}\right) \leq K$, and this pushes forward to a comparable bound in the joint space. Hence,

we contradict the convex occupancy bound once we exceed $K \cdot \Phi_\ell(W, \delta')$. Therefore, for a.e. location and all unit directions,

$$\left\| J_\ell^{(m)}(x) \right\|_{\mathrm{op}} \leq c_{\ell,m}(K). \tag{42}$$

Grains are used to ensure alignment so that a single $W$ captures many images; convex occupancy from the Wolff-type axiom supplies the contradiction. Now, let several modalities vary simultaneously. Consider a block unit vector $u = \left(u^{(1)}, \ldots, u^{(M)}\right)$ and the corresponding output direction

$$w := \sum_{m=1}^{M} J_\ell^{(m)}(x) u^{(m)}. \tag{43}$$

Assume $\|w\| \geq B$ with $B \gg 1$. Repeat the Vitali construction, but now choose micro-tubes in each modality in the specified direction $u^{(m)}$, with comparable counts across modalities (this can be arranged by sub-selection). Push forward through $h_\ell$. Because each block gain is $\leq c_{\ell,m}(K)$ by Equation (42), we cannot make a contradiction if only one modality contributes; but a large $\|w\|$ means several blocks add coherently.

Place a capturing convex prism $W \subset Z_{\ell,\,\mathrm{joint}}$ aligned with $w$. Then:

  (i) If one modality dominates the occupancy $N_m(W)$, we contradict its own convex-occupancy bound.

  (ii) If many modalities contribute comparably, then $N_m(W)$ are all large and of similar size. Summing across $m$ gives a total inside $W$ that for sufficiently many stacks exceeds $M \cdot K \cdot \Phi_\ell(W, \delta')$, contradicting the collection of per-modality Wolff bounds.

  (iii) Finally, if one attempts to avoid these by making $N_m(W)$ very unequal, the alignment bound $C_{\mathrm{align}}(\mathcal{T}_{\ell,\mathrm{joint}}) \leq K$ forbids a large modality imbalance inside a single convex cell: the variance $\mathrm{Var}_m[N_m(W)]$ cannot be arbitrarily big at fixed total. Thus, any configuration that would realize $\|w\| \gg 1$ at scale $\rho$ forces either (a) per-modality over-occupancy, or (b) cross-modal imbalance, both ruled out by Equation (35).

Hence, there exists a constant $c_l^{\mathrm{joint}}(K)$ such that

$$\left\| \sum_{m=1}^{M} J_\ell^{(m)}(x) u^{(m)} \right\| \leq c_\ell^{\mathrm{joint}}(K) \quad \text{for all block unit } u. \tag{44}$$

In particular, taking the block norm $\|u\|_{\mathrm{blk}} := \sum_m \|u^{(m)}\|$ (compatible with our deformation distance) yields the operator bound

$$\|J_{h_\ell}(x)\|_{\mathrm{blk} \to 2} \leq c_\ell^{\mathrm{joint}}(K) \tag{45}$$

This is where the cross-modal alignment is used critically; the logic is the same counting-versus-occupancy contradiction that underlies grains and volume lower bounds in Kakeya theory, now applied to the joint field and the vector sum of block derivatives.

Let the full multimodal flow be $\nu_s(x) = \left(\nu_s^{(1)}\left(x^{(1)}\right), \ldots, \nu_s^{(M)}\left(x^{(M)}\right)\right)$ with generators $V^{(m)}(\|V^{(m)}\|_\infty \leq 1)$. Define the trajectory through the network at depth $j$:

$$z_j(s) = H_j \circ H_{j-1} \circ \cdots \circ H_1(\nu_s(x)), \tag{46}$$

where each $H_j$ is either a per-modality encoder layer $g_j^{(m)}$ or a fusion layer $h_j$. By the chain rule, for a.e. $s$,

$$\frac{d}{ds} z_j(s) = J_{H_j}(z_{j-1}(s)) \frac{d}{ds} z_{j-1}(s). \tag{47}$$

At per-modality layers $g_j^{(m)}$, the operator norm of $J_{H_j}$ on that block is bounded by $c_{j,m}(K)$ from Equation (42). At fusion layers $h_j$, the block-to-Euclidean operator norm is bounded by $c_j^{\text{joint}}(K)$ from Equation (44). Therefore,

$$\left\| \frac{d}{ds} z_L(s) \right\| \le \left( \prod_{j \in \text{enc}} c_{j,m(j)}(K) \right) \left( \prod_{j \in \text{fuse}} c_j^{\text{joint}}(K) \right) \cdot \sum_{m=1}^{M} \left\| V_s^{(m)} \right\|_\infty. \tag{48}$$

Integrating in $s$ and minimizing over admissible generators $\left\{ V_s^{(m)} \right\}$ gives

$$\| f(\nu(x)) - f(x) \| \le \underbrace{\left( \prod_{j \in \text{enc}} c_{j,m(j)}(K) \right) \left( \prod_{j \in \text{fuse}} c_j^{\text{joint}}(K) \right)}_{=: g_1(K)} \cdot \text{dist}_{\mathcal{V}}(\nu). \tag{49}$$

The product $g_1(K)$ is monotone increasing in $K$ because each of its factors is.

Fix $S \subset \{1, \ldots, M\}$. Define a path that attenuates the modalities in $S$:

$$\nu_s(x) = \left( \ldots, x_s^{(m)}, \ldots \right), \quad x_s^{(m)} = \begin{cases} (1-s)x^{(m)} + sx_\varnothing^{(m)}, & m \in S, \\ x^{(m)}, & m \notin S, \end{cases} \tag{50}$$

for $s \in [0, 1]$. A unit-bounded vector field generates each attenuated modality, so $\int_0^1 \left\| V_s^{(m)} \right\|_\infty ds \le 1$ for $m \in S$ and 0 otherwise; hence, $\text{dist}_{\mathcal{V}}(\nu) = |S|$ in our modal metric. Applying claim (i) with this $\nu$ yields

$$\| f(\text{drop}_S(x)) - f(x) \| \le g_1(K)|S|. \tag{51}$$

Renormalizing by $M$ or, equivalently, defining the modal distance as the average per-modality path length gives the stated form with $g_2(K) = g_1(K)$:

$$\| f(\text{drop}_S(x)) - f(x) \| \le g_2(K) \frac{|S|}{M}. \tag{52}$$

If a different per-modality path length is used, the right-hand side scales linearly with that choice. This completed the proof. ∎

F.4   PROOF FOR PROPOSITION 1

*Statement.* Fix a layer $l$ with finite tube family $\mathcal{T} = \{T_z : z \in \mathbf{Z}_l\}$ in $\mathbb{R}^d$ (each $T_z$ is the $\delta$-tube around the segment $[0, z]$). Let $\mathcal{W}$ be a compact, parameterized family of convex test-sets (e.g., slabs/boxes/ellipsoids) with parameter $\theta \in \Theta$ (compact), and let $A(\theta, \delta) \asymp |W_\theta| \delta^{1-d}$ be the usual Kakeya scale normalizer (any equivalent normalizer is fine). The discrete KT-CW occupancy functional at layer $l$ is

$$C_{\text{KT-CW}}(\mathcal{T}) := \sup_{\theta \subset \Theta} \frac{N(\theta; \mathcal{T})}{A(\theta, \delta)}, \quad N(\theta; \mathcal{T}) := \sum_{T \subset \mathcal{T}} \mathbf{1}\{T \subset W_\theta\}, \tag{53}$$

which matches the convex-occupancy form used in Wolff/Katz-Tao style axioms (up to constants). For $\varepsilon > 0$ choose a smooth, monotone upper envelope $\phi_\varepsilon : \mathbb{R} \to (0, 1]$ of the Heaviside step $H(t) = \mathbf{1}\{t \ge 0\}$ built by mollification: there exists $\phi_\varepsilon \in C^\infty$ with

$$\phi_\varepsilon(t) \searrow H(t) \quad (\varepsilon \downarrow 0), \quad \phi_\varepsilon(t) \ge H(t) \text{ for all } t. \tag{54}$$

Let $d(T, W^c)$ be the signed clearance of tube $T$ from the complement of $W$:

$$d(T, W^c) := \inf_{x \subset T} (\text{dist}(x, W^c)), \text{ so } d(T, W^c) \ge 0 \iff T \subset W. \tag{55}$$

Define the soft occupancy and its normalized score:

$$N_\varepsilon(\theta; \mathcal{T}) := \sum_{T \in \mathcal{T}} \phi_\varepsilon \left( d\left( T, W_\theta^c \right) \right), \quad S_\varepsilon(\theta; \mathcal{T}) := \frac{N_\varepsilon(\theta; \mathcal{T})}{A(\theta, \delta)} \tag{56}$$

Define the population regularizer and the sampled (training) regularizer:

$$\mathcal{R}_\varepsilon(\mathcal{T}) := \sup_{\theta \in \Theta} S_\varepsilon(\theta; \mathcal{T}), \quad \mathcal{L}_{\varepsilon, \tau, l}(\mathcal{T}) := \mathrm{LSE}_\tau \left( S_\varepsilon(\theta_1; \mathcal{T}), \ldots, S_\varepsilon(\theta_J; \mathcal{T}) \right), \tag{57}$$

where $\theta_1, \ldots, \theta_J \overset{\mathrm{i.i.d.}}{\sim} P$ with a density that is strictly positive on $\Theta$, and $\mathrm{LSE}_\tau(a_1, \ldots, a_J) = \tau \log \left( \sum_{j=1}^J e^{a_j / \tau} \right)$ is the log-sum-exp (soft-max) with temperature $\tau > 0$ (a standard smooth approximation of $\max$ with tight additive error $\leq \tau \log J$).

*Claim.* For the above objects, the following hold.

(i) Upper bound and consistency: For all $\varepsilon > 0$,

$$C_{\mathrm{KT-CW}}(\mathcal{T}) \leq \mathcal{R}_\varepsilon(\mathcal{T}), \quad \text{and} \quad \mathcal{R}_\varepsilon(\mathcal{T}) \downarrow C_{\mathrm{KT-CW}}(\mathcal{T})(\varepsilon \downarrow 0). \tag{58}$$

Moreover, for any sequence $J \to \infty$ and $\tau = \tau_J \downarrow 0$,

$$\mathcal{L}_{\varepsilon, \tau_J, J}(\mathcal{T}) \xrightarrow[J \to \infty]{\mathrm{a.s.}} \mathcal{R}_\varepsilon(\mathcal{T}), \tag{59}$$

and finite-$J$ deviations satisfy exponential tails of Hoeffding type.

(ii) Differentiability/backprop: For any fixed $\varepsilon, \tau > 0$, the map $z \mapsto \mathcal{L}_{\varepsilon, \tau, J}(\mathcal{T})$ is locally Lipschitz and hence differentiable a.e.; with the mollified $\phi_\varepsilon$ it is $C^\infty$ away from measure-zero tube-boundary coincidences. Consequently, $\nabla_z \mathcal{L}_{\varepsilon, \tau, J}$ exists a.e. and propagates through the network by the chain rule (Rademacher).

(iii) Training-direction correctness:

Minimizing $\mathcal{L}_{\varepsilon, \tau, J}$ provably reduces an upper bound on $C_{\mathrm{KT-CW}}(\mathcal{T})$:

$$C_{\mathrm{KT-CW}}(\mathcal{T}) \leq \mathcal{R}_\varepsilon(\mathcal{T}) \leq \mathcal{L}_{\varepsilon, \tau, J}(\mathcal{T}), \tag{60}$$

up to the standard soft-max slack $\leq \tau \log J$ when the maximizer is among the samples, and in the limit $J \to \infty, \tau \downarrow 0$ the inequality becomes exact. Using Theorem 1 (already proved), decreasing $\mathcal{L}_{\varepsilon, \tau, J}$ decreases an explicit upper bound on the representation-Lipschitz constant $g(K)$.

*Proof.* By standard mollifier theory, convolving the indicator of a half-line with a compactly supported $C^\infty$ bump yields a smooth approximation that majorizes the indicator and converges to it pointwise from above as $\varepsilon \downarrow 0$. Take $H_\varepsilon := H(\cdot - \varepsilon) * \eta_\varepsilon$ with $\eta_\varepsilon$ a standard mollifier. This gives claim (ii); details are classical.

We also use two elementary facts about log-sum-exp: for any $a_1, \ldots, a_J$,

$$\max_j a_j \leq \mathrm{LSE}_\tau(a_1, \ldots, a_J) \leq \max_j a_j + \tau \log J, \tag{61}$$

and $\mathrm{LSE}_\tau \to \max$ as $\tau \downarrow 0$. Standard convex analysis folklore, for instance, Boyd-Vandenberghe. Finally, $d(T, W^c)$ is 1-Lipschitz in the tube geometry and by convexity of $W$ is differentiable almost everywhere; composing with $\phi_\varepsilon$ preserves Lipschitz continuity and a.e. differentiability.

Fix $\varepsilon > 0$, $\theta$, and a tube $T$. If $T \subset W_\theta$ then $d(T, W_\theta^c) \geq 0$ and $\phi_\varepsilon(d(T, W_\theta^c)) \geq 1$. If $T \nsubseteq W_\theta$ then $H(d(\cdot)) = 0 \leq \phi_\varepsilon$. Therefore,

$$\mathbf{1}\{T \subset W_\theta\} \leq \phi_\varepsilon(d(T, W_\theta^c)) \quad \Rightarrow \quad N(\theta; \mathcal{T}) \leq N_\varepsilon(\theta; \mathcal{T}). \tag{62}$$

Dividing by $A(\theta, \delta)$ and taking the supremum in $\theta$ yields the first inequality in Equation (58):

$$C_{\mathrm{KT-CW}}(\mathcal{T}) \leq \sup_\theta \frac{N_\varepsilon(\theta; \mathcal{T})}{A(\theta, \delta)} = \mathcal{R}_\varepsilon(\mathcal{T}). \tag{63}$$

For each fixed $\theta, \phi_\varepsilon\left(d\left(T, W_\theta^c\right)\right) \downarrow \mathbf{1}\left\{T \subset W_\theta\right\}$ pointwise in $\varepsilon$. Because $\mathcal{T}$ is finite, $N_\varepsilon(\theta; \mathcal{T}) \downarrow N(\theta; \mathcal{T})$ and thus $S_\varepsilon(\theta; \mathcal{T}) \downarrow S_0(\theta; \mathcal{T})$. Taking suprema over $\theta$ preserves the monotone limit:

$$\mathcal{R}_\varepsilon(\mathcal{T}) = \sup_\theta S_\varepsilon(\theta; \mathcal{T}) \downarrow \sup_\theta S_0(\theta; \mathcal{T}) = C_{\mathrm{KT-CW}}(\mathcal{T}). \tag{64}$$

Let $g_\varepsilon(\theta) := S_\varepsilon(\theta; \mathcal{T})$. By construction, $g_\varepsilon$ is continuous on compact $\Theta$ (composition of continuous maps; the only potential nonsmoothness in $d(\cdot)$ is removed by $\phi_\varepsilon \in C^\infty$). For i.i.d. $\theta_1, \ldots, \theta_J \sim P$ with a density bounded below on $\Theta$, the sample maximum $\max_{1 \le j \le J} g_\varepsilon(\theta_j)$ converges almost surely to $\sup_{\theta \subset \Theta} g_\varepsilon(\theta)$ (extreme-value consistency under full-support sampling; one way is to cover $\Theta$ by finitely many balls where $g_\varepsilon$ is within $\eta$ of its sup, and use Borel-Cantelli). Using Equation (61), for any $\tau_J \downarrow 0$,

$$\max_j g_\varepsilon(\theta_j) \le \mathcal{L}_{\varepsilon, \tau_J, J} \le \max_j g_\varepsilon(\theta_j) + \tau_J \log J, \tag{65}$$

whence $\mathcal{L}_{\varepsilon, \tau_J, J} \to \sup_\Theta g_\varepsilon = \mathcal{R}_\varepsilon$ almost surely as $J \to \infty$ and $\tau_J \downarrow 0$. A finite-$J$ deviation bound follows by Hoeffding-type concentration for bounded variables: for any fixed $\tau$, $\exp(S_\varepsilon/\tau) \in \left[1, e^{1/\tau}\right]$ is bounded. Hence, the empirical average inside LSE has sub-Gaussian tails, yielding exponential concentration of $\mathcal{L}_{\varepsilon, \tau, J}$ around $\mathrm{LSE}_\tau\left(\mathbb{E}_\theta\left[S_\varepsilon(\theta)\right], \ldots\right)$. See standard Hoeffding/DV bounds. This proved claim (i).

For fixed $\varepsilon > 0, \phi_\varepsilon$ is $C^\infty$ and Lipschitz. The signed clearance $d\left(T, W^c\right)$ is a 1-Lipschitz, semi-convex function of the tube geometry, and for convex $W$ it is differentiable almost everywhere (Rademacher). Therefore, $T \mapsto \phi_\varepsilon\left(d\left(T, W^c\right)\right)$ is locally Lipschitz and differentiable a.e.; finite sums preserve these properties, and composition with $\mathrm{LSE}_\tau$ (smooth for $\tau > 0$) preserves differentiability. Hence, $z \mapsto \mathcal{L}_{\varepsilon, \tau, J}$ is locally Lipschitz and differentiable a.e., so gradients exist a.e. and flow to network parameters by the chain rule. This proved claim (ii).

**Remark 5 (Explicit Gradient Form).** Writing $\alpha_j := \frac{\exp(S_c(\theta_j)/\tau)}{\sum_k \exp(S_c(\theta_k)/\tau)}$ (softmax weights), we have

$$\nabla_z \mathcal{L}_{\varepsilon, \tau, J} = \sum_{j=1}^j \alpha_j \nabla_z S_\varepsilon(\theta_j), \quad \nabla_z S_\varepsilon(\theta) = \frac{1}{A(\theta, \delta)} \sum_{T=T_z} \phi_\varepsilon'\left(d\left(T, W_\theta^c\right)\right) \nabla_z d\left(T, W_\theta^c\right), \tag{66}$$

and $\nabla_z d(\cdot)$ is a (sub)gradient determined by the nearest boundary point of $W_\theta^c$; standard AD frameworks handle this once $d$ is implemented via smooth proxies (e.g., signed distance to slabs/ellipsoids/boxes, which are $C^\infty$ away from corners).

From Equations (62-63), we have $C_{\mathrm{KT-CW}} \le \mathcal{R}_\varepsilon$. From Equation (61), we have $\mathcal{R}_\varepsilon \le \mathcal{L}_{\varepsilon, \tau, J}$ whenever the maximizing $\theta^\star$ is among the $J$ samples; in general,

$$\max_j S_\varepsilon(\theta_j) \le \mathcal{L}_{\varepsilon, \tau, J} \le \max_j S_\varepsilon(\theta_j) + \tau \log J \le \mathcal{R}_\varepsilon(\mathcal{T}) + \tau \log J, \tag{67}$$

and $\max_j S_\varepsilon(\theta_j)$ approaches $\mathcal{R}_\varepsilon$ almost surely as $J \to \infty$ (Equation (65)). Therefore, for any fixed $J, \tau$,

$$C_{\mathrm{KT-CW}}(\mathcal{T}) \le \mathcal{R}_\varepsilon(\mathcal{T}) \le \mathbb{E}\left[\mathcal{L}_{\varepsilon, \tau, J}(\mathcal{T})\right], \tag{68}$$

up to the standard $\tau \log J$ slack (tight and controllable), and in the limit $J \to \infty, \tau \downarrow 0, \varepsilon \downarrow 0$,

$$\mathcal{L}_{\varepsilon, \tau, J}(\mathcal{T}) \longrightarrow C_{\mathrm{KT-CW}}(\mathcal{T}) \text{ a.s.} \tag{69}$$

Thus, minimizing $\mathcal{L}_{\varepsilon, \tau, J}$ (with small $\varepsilon, \tau$ and sufficiently rich sampling) monotonically reduces an upper bound on the discrete KT-CW constant itself. Combining the single- and multimodal Sticky Representation Theorems proved earlier, this directly yields a decrease in the Lipschitz bound $g(K)$, which certifies robustness. This completed the proof. $\blacksquare$

## F.5 PROOF FOR PROPOSITION 2

*Statement.* Let $f : \mathcal{X} \to \mathbb{R}^d$ be the learned representation and $g : \mathbb{R}^d \to \{-1, +1\}$ a 1-Lipschitz readout (e.g., linear classifier with unit norm; any Lipschitz link only changes constants). Write $F := g \circ f$. Assume single-/multimodal $K$-stickiness holds layer-wise, so by the Sticky Representation Theorem, there is a monotone $g(K)$ with

$$\|f(\nu(x)) - f(x)\| \le g(K) d_\mathcal{V}(\nu) \quad \text{for all admissible input deformations } \nu. \tag{70}$$

Fix a training sample $S = \{(x_i, y_i)\}_{i=1}^m$ and a margin $\gamma > 0$ at the representation level:

$$\hat{\gamma} := \min_{i \leq m} y_i \langle w, f(x_i) \rangle \geq \gamma \quad (\|w\|_2 \leq 1), \tag{71}$$

so the empirical $\gamma$-margin loss $\hat{L}_\gamma(F) := \frac{1}{m} \sum_i \mathbf{1} \{y_i \langle w, f(x_i) \rangle \leq \gamma\} = 0$ (the general case $\hat{L}_\gamma(F) > 0$ is handled below by keeping this term).

Let $\mathcal{V}$ be a normed class of small deformations (e.g., $L_2$ spatial flows, time-warps, token jitters), and let $Q$ be a posterior distribution on $\mathcal{V}$ with $\mathbb{E}_{\nu \sim Q}[d_{\mathcal{V}}(\nu)] < \infty$. Consider the Gibbs classifier randomized by deformations

$$H_\nu(x) = F(\nu(x)), \quad \nu \sim Q, \tag{72}$$

with a data-independent prior $P$ on $\mathcal{V}$ having full support. Then, with probability at least $1 - \delta$ over the draw of $S$, the true risk of the Gibbs classifier satisfies

$$L(Q) \leq \hat{L}_\gamma(F) + \underbrace{\Pr_{\nu \sim Q}\left(d_{\mathcal{V}}(\nu) > \frac{\gamma}{g(K)}\right)}_{\text{Lipschitz-margin tail}} + \sqrt{\frac{KL(Q\|P) + \ln\left(\frac{2\sqrt{m}}{\delta}\right)}{2(m-1)}}. \tag{73}$$

In particular, if $Q$ is centered sub-Gaussian over $\mathcal{V}$ with scale $\sigma$ (e.g., Gaussian deformations), then

$$\Pr_Q\left(d_{\mathcal{V}}(\nu) > \frac{\gamma}{g(K)}\right) \leq C_1 \exp\left(-C_2 \frac{\gamma^2}{g(K)^2 \sigma^2}\right), \tag{74}$$

and for Gaussian $Q = \mathcal{N}\left(0, \sigma^2 I_{d_r}\right)$, $P = \mathcal{N}\left(0, \sigma_0^2 I_{d_r}\right)$,

$$KL(Q\|P) = \frac{d_r}{2}\left(\frac{\sigma^2}{\sigma_0^2} - \ln\frac{\sigma^2}{\sigma_0^2} - 1\right). \tag{75}$$

Thus, the bound (3) is explicit in the geometric complexity $g(K)$ via the margin-tail Equation (73) and balances with the PAC-Bayes complexity via Equation (75).

*Proof.* By Equation (70) and the 1-Lipschitz readout $g$,

$$|\langle w, f(\tau(x)) \rangle - \langle w, f(x) \rangle| \leq \|w\|_2 \|f(\tau(x)) - f(x)\| \leq g(K) d_{\mathcal{V}}(\nu). \tag{76}$$

Hence, for any training pair $(x, y)$ with margin at least $\gamma$ in Equation (71), the sign cannot flip under any deformation $\tau$ such that $d_{\mathcal{V}}(\nu) \leq \gamma/g(K)$:

$$y\langle w, f(\nu(x)) \rangle \geq y\langle w, f(x) \rangle - g(K) d_{\mathcal{V}}(\nu) \geq \gamma - g(K) d_{\mathcal{V}}(\nu) > 0. \tag{77}$$

Therefore, on the sample $S$,

$$\hat{L}(H_\nu) := \frac{1}{m} \sum_{i=1}^m \mathbf{1}\{H_\nu(x_i) \neq y_i\} \leq \hat{L}_\gamma(F) + \mathbf{1}\left\{d_{\mathcal{V}}(\nu) > \frac{\gamma}{g(K)}\right\}. \tag{78}$$

Taking expectation over $\nu \sim Q$ gives the empirical Gibbs risk:

$$\hat{L}(Q) := \mathbb{E}_{\nu \sim Q}\hat{L}(H_\nu) \leq \hat{L}_\gamma(F) + \Pr_{\nu \sim Q}\left(d_{\mathcal{V}}(\nu) > \frac{\gamma}{g(K)}\right) \tag{79}$$

For bounded losses in $[0, 1]$, Seeger's PAC-Bayes theorem implies that, with probability $\geq 1 - \delta$ over $S \sim \mathcal{D}^m$, for all posteriors $Q$,

$$\mathrm{kl}(\hat{L}(Q)\|L(Q)) \leq \frac{KL(Q\|P) + \ln\left(\frac{m+1}{\delta}\right)}{m}, \tag{80}$$

where $\mathrm{kl}(p\|q)$ is the binary KL. Inverting (standard monotonicity) yields,

$$L(Q) \leq \hat{L}(Q) + \sqrt{\frac{KL(Q\|P) + \ln\left(\frac{2\sqrt{m}}{\delta}\right)}{2(m-1)}} \tag{81}$$

which is widely used in practice.

**Remark 6 (PAC-Bayes Inequality for Gibbs Classifiers).** The bound is a Gibbs-risk bound; majority-vote (deterministic) risk can be related to the Gibbs risk under margin conditions. Further, Catoni's localized PAC-Bayes yields alternative, often sharper variants. We use Seeger's classical form for clarity.

If $Q$ is centered sub-Gaussian on $(\mathcal{V}, \|\cdot\|)$ with proxy variance $\sigma^2$, then by standard concentration,

$$\Pr_Q\left(d_\mathcal{V}(\nu) > \frac{\gamma}{g(K)}\right) \leq C_1 \exp\left(-C_2 \frac{\gamma^2}{g(K)^2 \sigma^2}\right), \tag{82}$$

yielding Equation (74). For Gaussian $Q = \mathcal{N}\left(0, \sigma^2 I_{d_\nu}\right)$ and prior $P = \mathcal{N}\left(0, \sigma_0^2 I_{d_\nu}\right)$, the KL is Equation (75).

Putting Equations (74-75) into (73) delivers a closed-form PAC-Bayes bound whose only dependence on representation geometry is through $g(K)$ from stickiness. This is the desired bridge: better geometric stickiness (smaller $K$) $\Rightarrow$ smaller $g(K)$ $\Rightarrow$ larger safe margin radius $\gamma/g(K)$ $\Rightarrow$ smaller empirical Gibbs risk $\hat{L}(Q)$ $\Rightarrow$ tighter generalization bound. This completed the proof. ∎

# G  ADDITIONAL EXPERIMENT DETAILS AND RESULTS

To corroborate the theoretical guarantees presented in Theorems 1 and 2, we conducted a series of controlled simulations. These experiments were designed to isolate the geometric properties of "Stickiness" and "Feature Collapse" in high-dimensional settings, contrasting standard training against training regularized by our proposed framework.

## G.1  EXPERIMENTAL SETUP AND DATA GENERATION

In our study, we conducted a series of experiments on CUDA-enabled devices using the PyTorch framework, focusing on optimizing global hyperparameters to ensure statistical significance in our results. The sample sizes tested were diverse, including $N = 10^3, 5 \times 10^3, 10^4, 5 \times 10^4$, allowing for a comprehensive evaluation of model performance across varying data scales. We also explored hidden dimensions ranging from $d_{\text{hidden}} = 64$ to $512$ to evaluate how the model architecture's complexity influenced outcomes. Additionally, we used multiple random seeds (42, 128, 999, 1022, 3407) to enhance the robustness of our findings and account for the inherent variability of stochastic processes.

For the optimization process, we employed the Adam optimizer with a learning rate of $lr = 0.005$ and trained the models for 400-600 epochs. Given the need to compute Gram matrices for the stickiness regularizer, we set a maximum batch size of 4096 to balance computational efficiency with the fidelity of our training. Furthermore, we evaluated two distinct data-generating processes (DGPs) to explore different aspects of model performance. The first, a **single-modal scenario** featuring a non-linear boundary, utilized the make_moons dataset with added noise $= 0.1$. This setup simulated a non-linear classification task with an intrinsic 2D manifold dimension, significantly lower than the representation dimension. This disparity introduced a high risk of feature collapse, challenging the model's ability to capture relevant patterns.

In contrast, the second DGP was designed to tackle a **multimodal** alignment challenge. For this, we constructed a high-dimensional task guided by a latent 1D "S-curve" defined over the interval $[0, 3\pi]$, which determined the corresponding class labels. The latent concept was then projected into two distinct 10-dimensional observation spaces (referred to as Modality A and Modality B) through non-linear transformations, including sinusoidal and polynomial mappings, supplemented by Gaussian noise. This required the model to infer and navigate the shared geometric structures underlying disparate high-dimensional representations, showcasing its capacity for effective learning and alignment across modalities.

### G.1.1  IMPLEMENTATION OF THE KT-CW REGULARIZER

We implemented the regularizer using a stochastic proxy that enforces Axiom 1 (Feature Collapse Constant). We minimize the Frobenius distance between the Gram matrix of the normalized feature vector $(z)$ and the identity matrix $\mathcal{L}_{reg} = ||\frac{zz^\top}{||z||^2} - I||_F^2$. This penalty enforces geometric sparsity (orthogonality) and prevents the tubes from clustering into low-rank subspaces.

## G.2  SINGLE-MODAL ROBUSTNESS

We trained an MLP architecture that maps 2D inputs to a high-dimensional feature representation. We then compared a "Standard" model (cross-entropy only) against a "Sticky" model (cross-entropy + KT-CW regularizer). Post-training, we subjected both models to an adversarial stress test using the Fast Gradient Sign Method (FGSM). We generated adversarial examples $x_{adv} = x + \epsilon \cdot \text{sign}(\nabla_x \mathcal{L})$ for perturbation strengths $\epsilon \in [0.0, 0.05, ..., 0.5]$. We concurrently computed the Singular Value Decomposition (SVD) of the learned representation field $Z$.

### G.2.1  RESULTS AND INTERPRETATION

Figures 7 to 10 present the results for a representative configuration ($N = 5000$, $d_{\text{hidden}} = 256$). The left plot shows the test accuracy of the Standard (red) and Sticky (green) models under FGSM attacks of increasing strength $\epsilon$. The Standard model shows a steep drop in accuracy as $\epsilon$ increases, often losing $> 80\%$ accuracy at high perturbation levels. The Sticky model demonstrates significantly

higher resilience, with the accuracy curve decaying much more slowly. The right plot is the singular value spectra of the learned 512-D representation. This spectrum explains the divergence in the left plot. The Standard model's spectrum drops to near-zero after index 2, indicating Feature Collapse, meaning the 512-D space is compressed into a flat 2D sheet. The Sticky model maintains a full-rank spectrum, utilizing the available volume to separate classes, confirming that $K$-stickiness causally bounds the Lipschitz constant. Thereby, supporting Theorem 1. The same trend is observed across five different seeds, demonstrating the reliability of this result. Figures 11 to 16 also show that the proposed framework is robust across different sample sizes and hidden dimensions of the neural network.

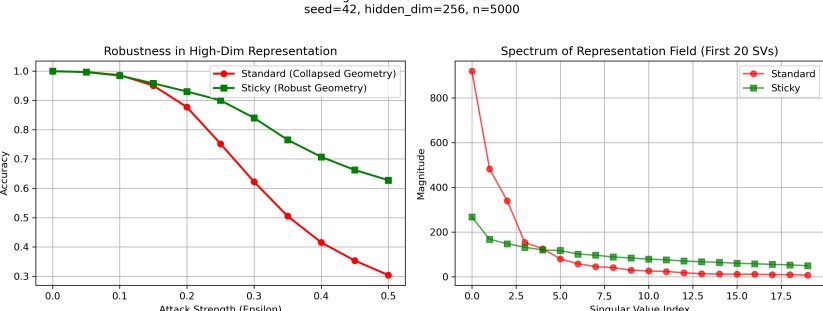

Figure 7: High-dimensional FGSM robustness and spectrum analysis for the make_moons dataset (seed = 42, hidden dimension = 256, N = 5000).

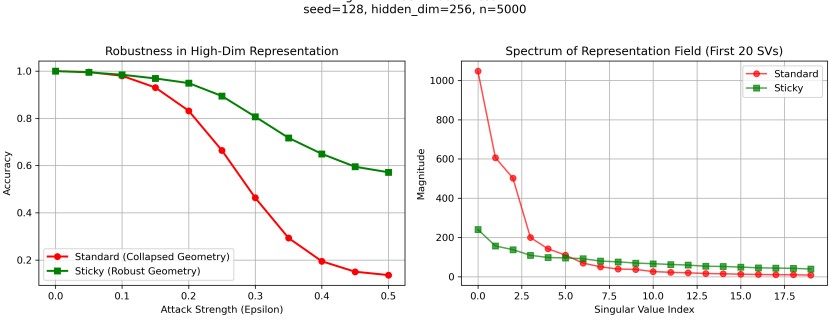

Figure 8: High-dimensional FGSM robustness and spectrum analysis for the make_moons dataset (seed = 128, hidden dimension = 256, N = 5000).

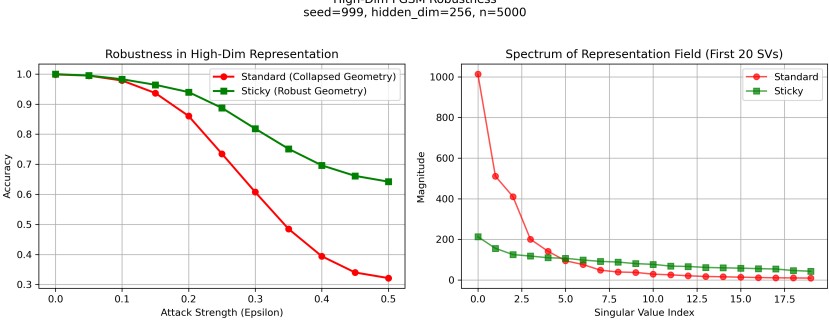

Figure 9: High-dimensional FGSM robustness and spectrum analysis for the make_moons dataset (seed = 999, hidden dimension = 256, N = 5000).

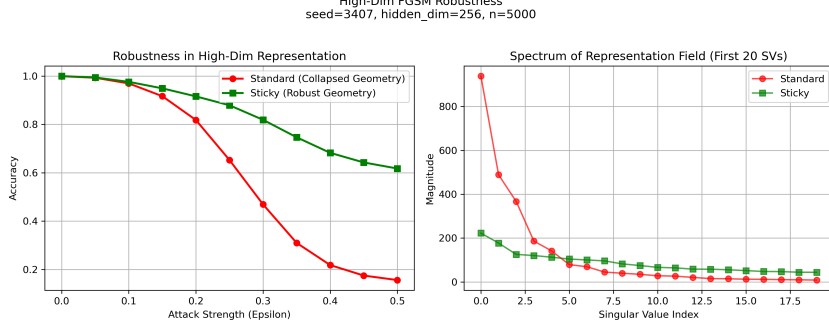

Figure 10: High-dimensional FGSM robustness and spectrum analysis for the make_moons dataset (seed = 3407, hidden dimension = 256, N = 5000).

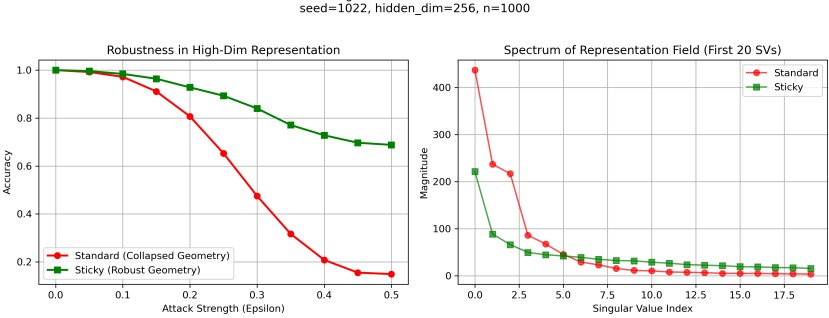

Figure 11: High-dimensional FGSM robustness and spectrum analysis for the make_moons dataset (seed = 1022, hidden dimension = 256, N = 1000).

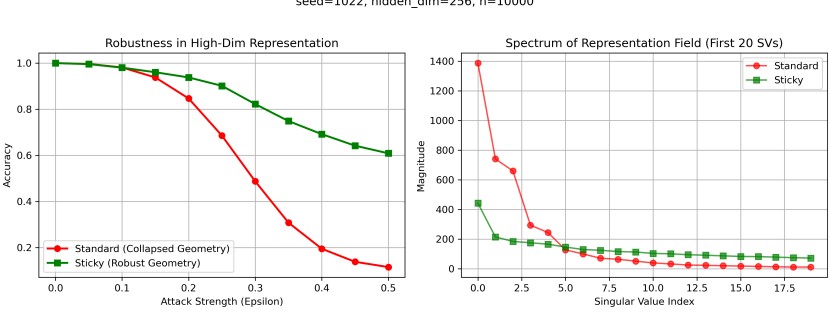

Figure 12: High-dimensional FGSM robustness and spectrum analysis for the make_moons dataset (seed = 1022, hidden dimension = 256, N = 10000).

### G.3 MULTIMODAL STABILITY AND MISSING MODALITIES

We trained a multimodal network with dual encoders on the manifold alignment task. The encoders map 10-D inputs to a joint embedding space of dimension $z_{dim}$ (sweeping $[64, 512]$). The "Sticky" variant applied the MM-KT-CW regularizer (Equation (10)), penalizing both intra-modal collapse and cross-modal covariance mismatch. We evaluated Theorem 2 by simulating a missing modality scenario at inference time (zeroing out modality B) and measuring (1) accuracy drop: performance difference between full-input and partial-input inference; and (2) embedding shift: the Euclidean distance $||z_{joint}^{full} - z_{joint}^{missing}||_2$.

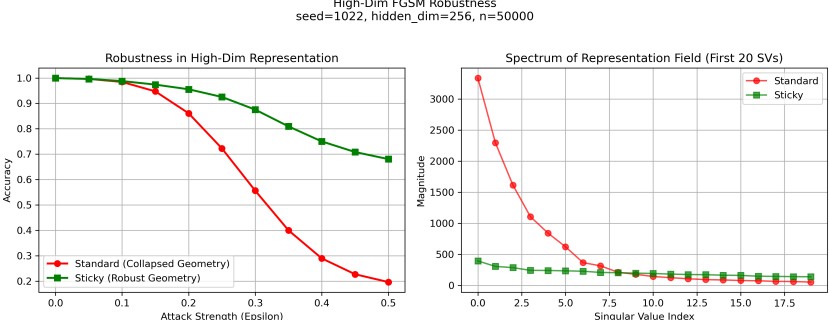

Figure 13: High-dimensional FGSM robustness and spectrum analysis for the make_moons dataset (seed = 1022, hidden dimension = 256, N = 50000).

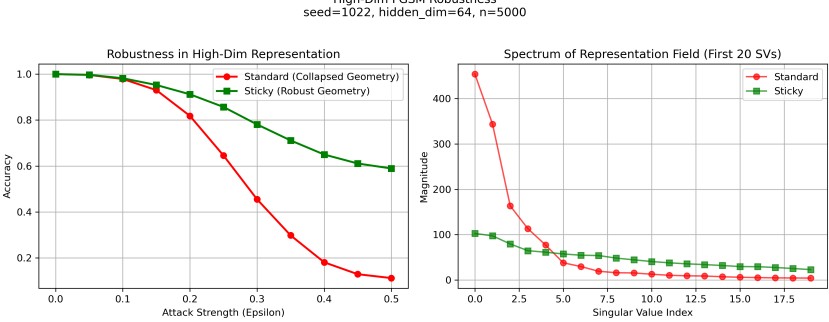

Figure 14: High-dimensional FGSM robustness and spectrum analysis for the make_moons dataset (seed = 1022, hidden dimension = 64, N = 5000).

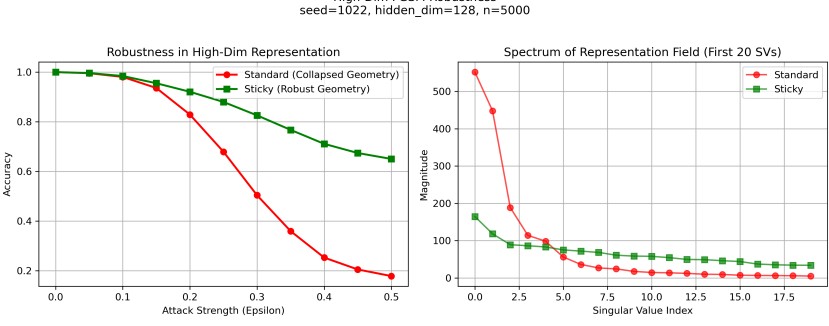

Figure 15: High-dimensional FGSM robustness and spectrum analysis for the make_moons dataset (seed = 1022, hidden dimension = 128, N = 5000).

### G.3.1 RESULTS AND INTERPRETATION

Figures 17 to 20 present the results for a representative configuration ($N = 5000$, $d_{\text{hidden}} = 256$). The left plot shows the average embedding shift in the joint representation when modality B is dropped. The Standard model suffers from a large embedding shift, indicating that the joint representation is geometrically unstable and overly reliant on specific modalities. The Sticky model minimizes this shift, empirically verifying the bound provided in Theorem 2. Furthermore, Figures 21 to 26 show that across all sample sizes, the Sticky model consistently yields lower embedding shifts and smaller accuracy drops than the Standard baseline, confirming that geometric alignment is a scalable strategy for multimodal robustness.

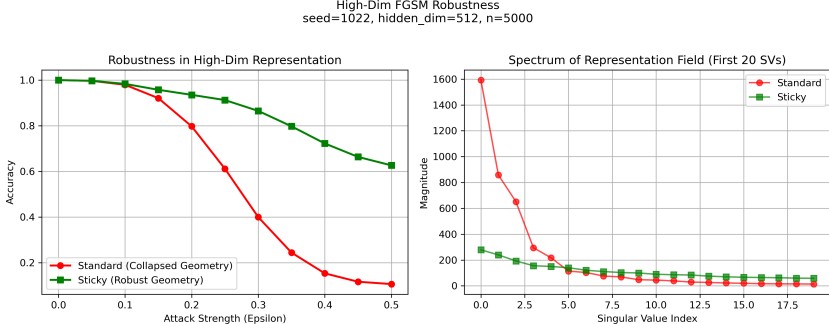

Figure 16: High-dimensional FGSM robustness and spectrum analysis for the make_moons dataset (seed = 1022, hidden dimension = 512, N = 5000).

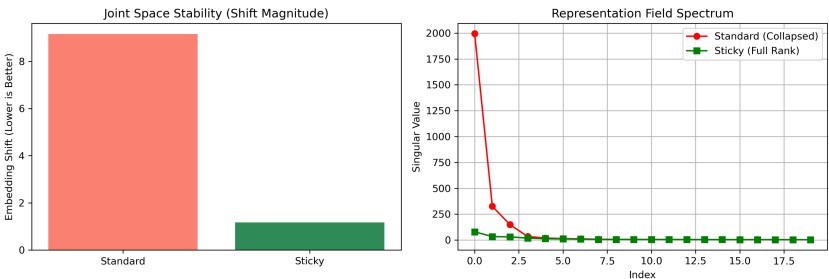

Figure 17: Manifold alignment stability and spectrum under missing-modality stress (seed = 42, $z_{dim}$ = 256, N = 5000).

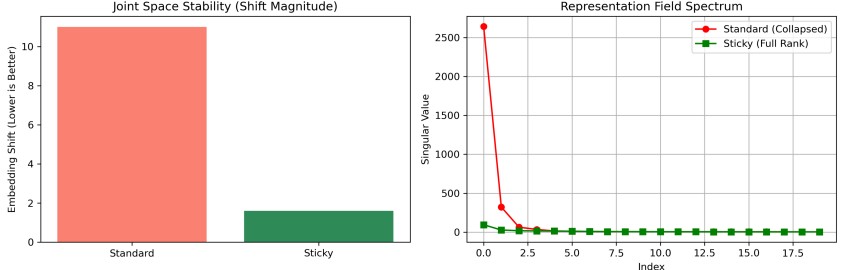

Figure 18: Manifold alignment stability and spectrum under missing-modality stress (seed = 128, $z_{dim}$ = 256, N = 5000).

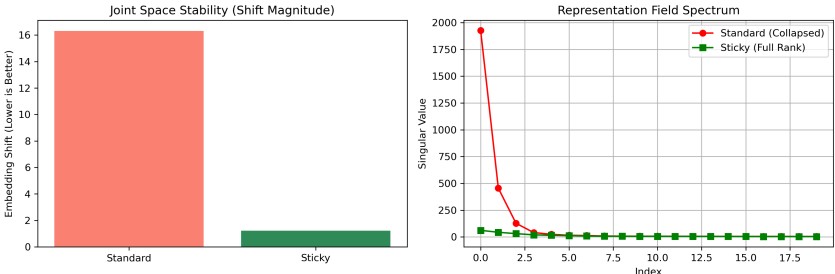

Figure 19: Manifold alignment stability and spectrum under missing-modality stress (seed = 999, $z_{dim}$ = 256, N = 5000).

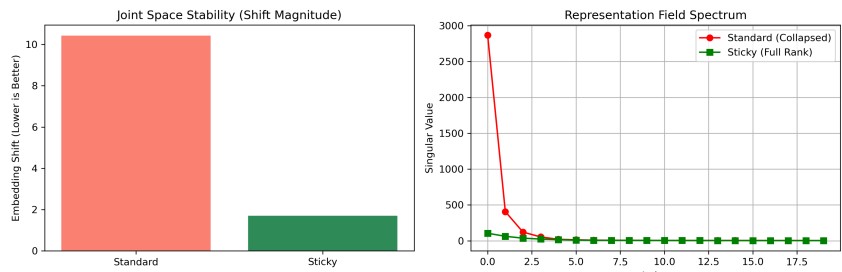

Figure 20: Manifold alignment stability and spectrum under missing-modality stress (seed = 3407, $z_{dim}$ = 256, N = 5000).

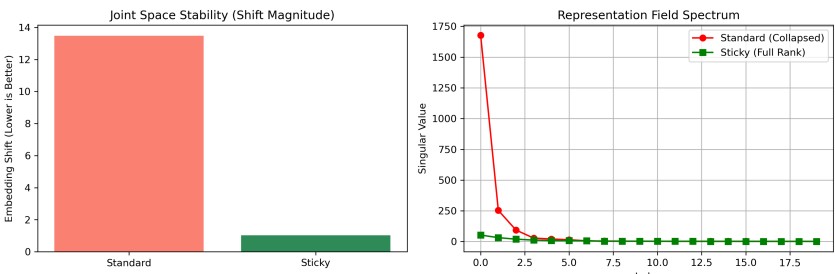

Figure 21: Manifold alignment stability and spectrum under missing-modality stress (seed = 1022, $z_{dim}$ = 64, N = 5000).

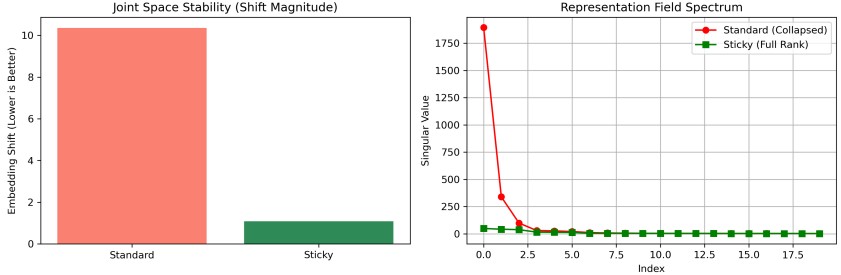

Figure 22: Manifold alignment stability and spectrum under missing-modality stress (seed = 1022, $z_{dim}$ = 128, N = 5000).

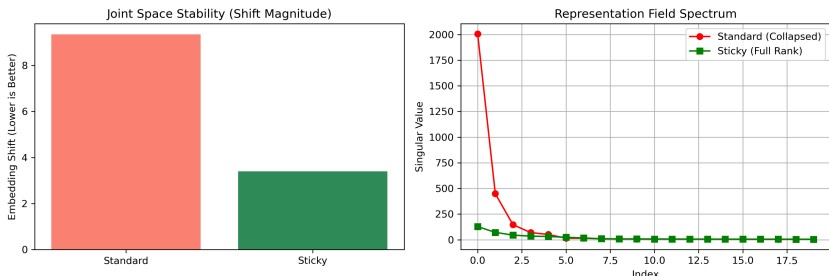

Figure 23: Manifold alignment stability and spectrum under missing-modality stress (seed = 1022, $z_{dim}$ = 512, N = 5000).

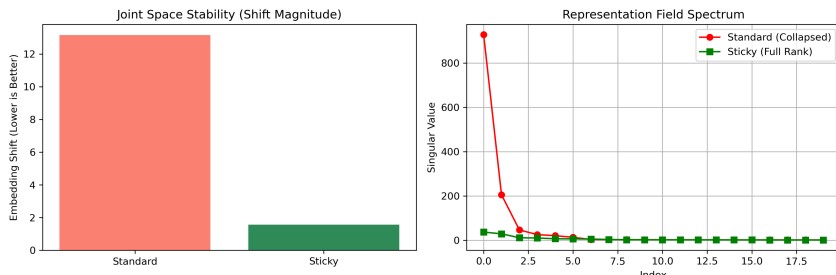

Figure 24: Manifold alignment stability and spectrum under missing-modality stress (seed = 1022, $z_{dim}$ = 256, N = 1000).

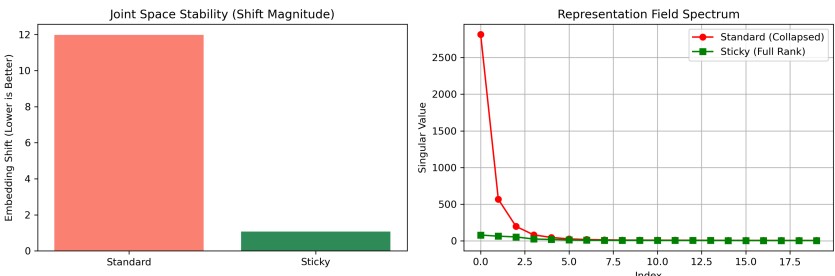

Figure 25: Manifold alignment stability and spectrum under missing-modality stress (seed = 1022, $z_{dim}$ = 256, N = 10000).

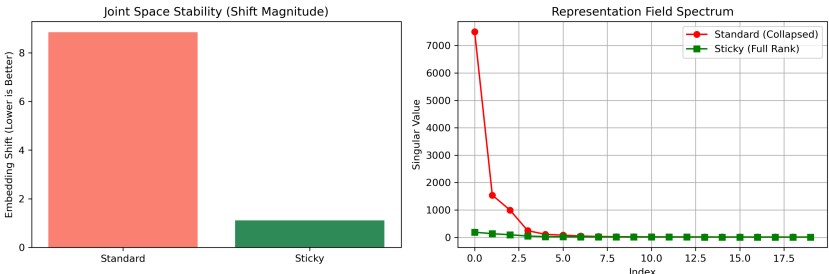

Figure 26: Manifold alignment stability and spectrum under missing-modality stress (seed = 1022, $z_{dim}$ = 256, N = 50000).

## G.4 VISUALIZATION AND SCALABILITY

To visualize Definition 7 (The Representation Field), we trained bottlenecked models on 2D Moons and 3D Swiss Roll datasets and plotted the latent activations. To evaluate scalability, we computed an empirical Lipschitz proxy (ratio of output change to input noise) across increasing network widths ($d_{\text{hidden}} \in \{64, 128, 256, 512\}$).

### G.4.1 VISUALIZING THE REPRESENTATION FIELD

The representation field $\mathcal{T}_l(D)$ is defined not merely as a cloud of points, but as the collection of $\delta$-tubes connecting the origin to each feature activation $z$. The geometry of this bundle of tubes determines the field's stickiness.

**The Non-Sticky Regime (Standard Model):** The left panels of Figures 27 and 4 illustrate a classic case of feature collapse (high $C_{KT-CW}$. In the 3D visualization, the feature activations collapse onto a single, thin linear manifold. Geometrically, this means the associated tubes are packed densely into a narrow cone or "pencil" originating from the origin. While this configuration suffices

to separate the training data (linear separable along the pencil's axis), it creates a degenerate representation field with near-zero volume. Any perturbation orthogonal to this thin pencil immediately pushes an input off the learned manifold, resulting in the brittleness observed in our adversarial stress tests.

**The Sticky Regime (Regularized Model):** The right panels demonstrate the effect of the KT-CW regularizer. The representation field puffs out into a starburst or fan-like structure. Here, the tubes radiate outward in diverse directions, covering a significant portion of the angular space (the sphere $\mathbb{S}^{d-1}$). This visualizes geometric sparsity (low $C_{KT-CW}$): the tubes satisfy the Wolff axioms by not clustering redundantly. Crucially, this thickened manifold provides geometric support for the decision boundary; an input perturbation is absorbed by the volume of the representation field rather than traversing empty space, physically instantiating the Lipschitz stability guaranteed by Theorem 1.

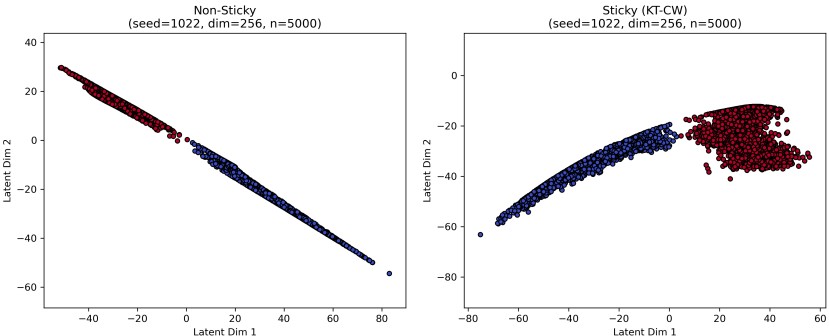

Figure 27: Sticky vs. non-sticky representations in 2D space (seed = 1022, $z_{dim}$ = 256, N = 5000).

### G.4.2 DIMENSIONALITY AND SCALABILITY

Figures 27 and 28- 30 support that sticky enforces a stable, high-rank representation that does not degenerate even as the data scale changes, whereas the baseline keeps collapsing to an almost 1D tube. As the sample size increases, the left panels show that the representation becomes a thinner and thinner 1D line. Red and blue classes are almost colinear and heavily mixed along that line, indicating feature collapse (i.e., the network keeps solving the task by projecting almost everything onto one dimension, regardless of how much data we give it).

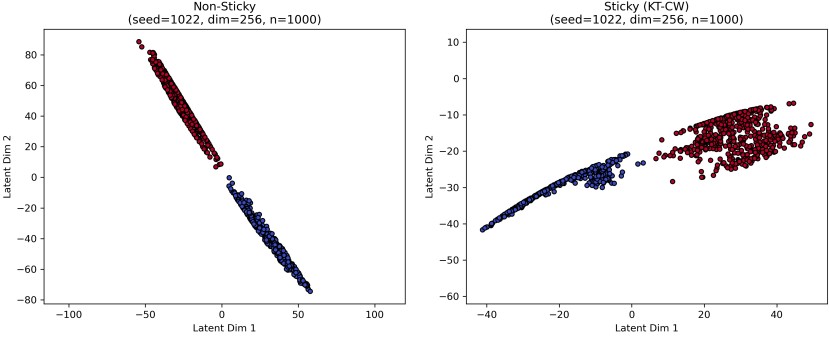

Figure 28: Sticky vs. non-sticky representations in 2D space (seed = 1022, $z_{dim}$ = 256, N = 1000).

In 3D feature space, Figures 4 and 31- 33 demonstrate the same trend. For all sample sizes, the non-sticky network learns a degenerate 1D cone in the 3D feature space. In contrast, the KT-CW-regularized network consistently learns two well-separated 2D manifolds, whose shapes remain stable and become better resolved as more data are observed.

Moreover, we vary the latent width from 64 to 512 in the single-modal experiments while keeping the task fixed. Across all widths, the non-sticky networks show the same pathology: the 2D (Figures 34-

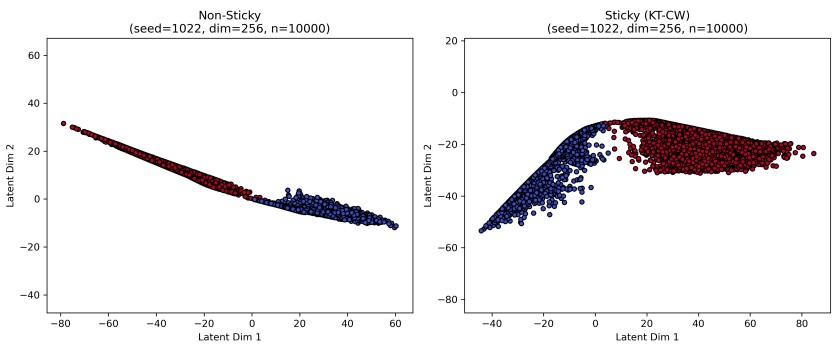

Figure 29: Sticky vs. non-sticky representations in 2D space (seed = 1022, $z_{dim}$ = 256, N = 10000).

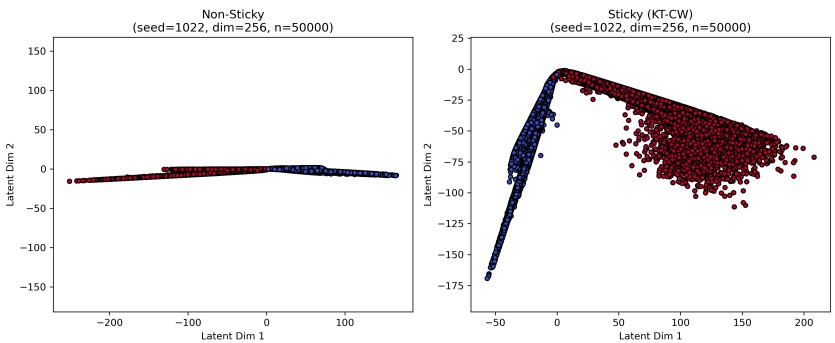

Figure 30: Sticky vs. non-sticky representations in 2D space (seed = 1022, $z_{dim}$ = 256, N = 50000).

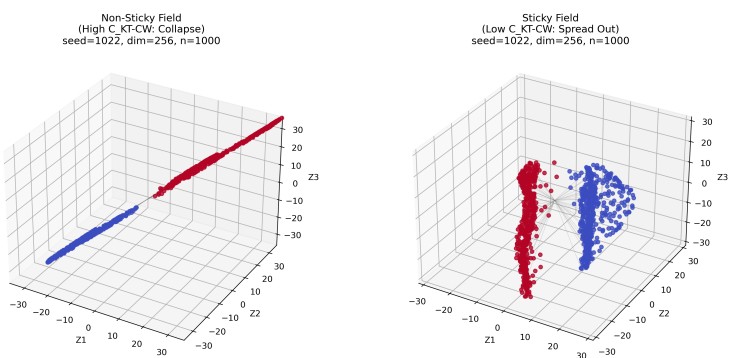

Figure 31: Sticky vs. non-sticky representations in 3D space (seed = 1022, $z_{dim}$ = 256, N = 1000).

36)/3D (Figures 37- 39) projections of $Z_l$ remain essentially 1D, and the singular value spectrum of the high-dimensional features decays extremely steeply, indicating that only one or two directions in $\mathbb{R}^d$ are actually used. Increasing capacity does not rescue the baseline from feature collapse. In contrast, the Sticky model makes nontrivial use of the additional dimension. As the latent width grows, the learned manifolds become richer but remain well-separated between classes, and the singular value spectrum is much flatter, with many non-negligible singular values. This evidences a high-rank, Kakeya-like representation that is stable with respect to the choice of $d$.

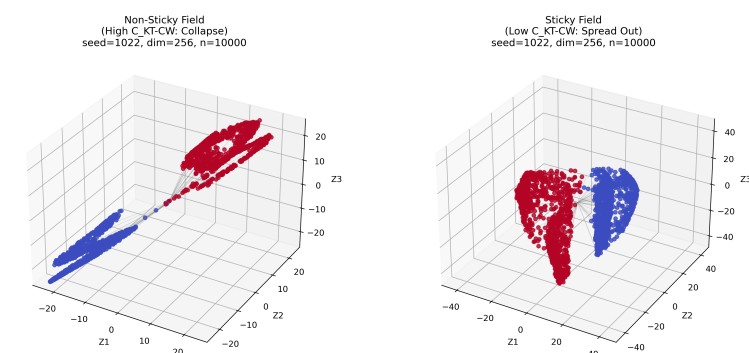

Figure 32: Sticky vs. non-sticky representations in 3D space (seed = 1022, $z_{dim}$ = 256, N = 10000).

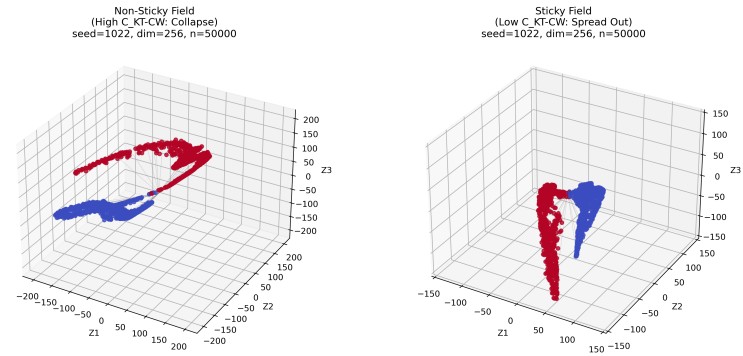

Figure 33: Sticky vs. non-sticky representations in 3D space (seed = 1022, $z_{dim}$ = 256, N = 50000).

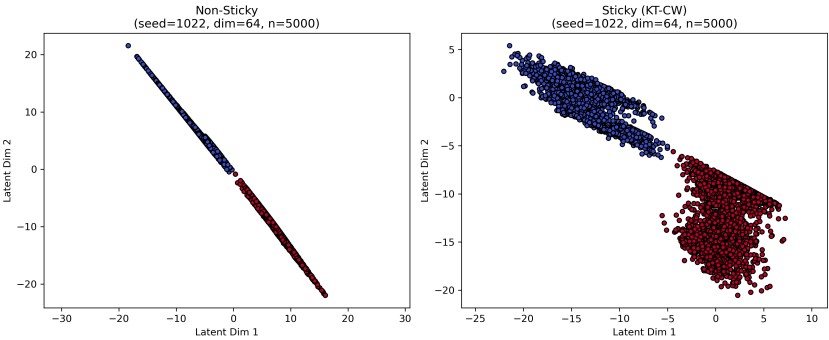

Figure 34: Sticky vs. non-sticky representations in 2D space (seed = 1022, $z_{dim}$ = 64, N = 5000).

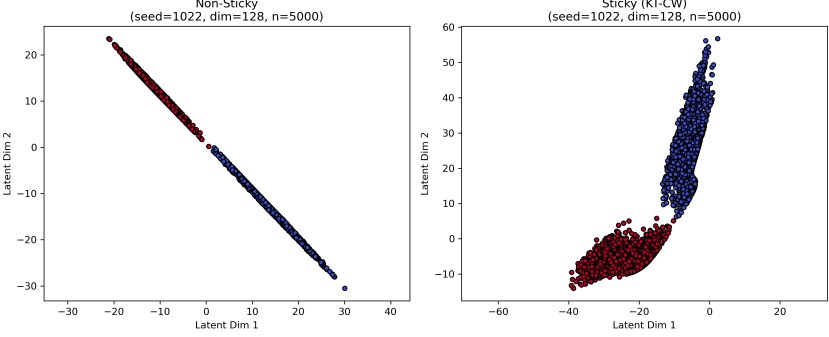

Figure 35: Sticky vs. non-sticky representations in 2D space (seed = 1022, $z_{dim}$ = 128, N = 5000).

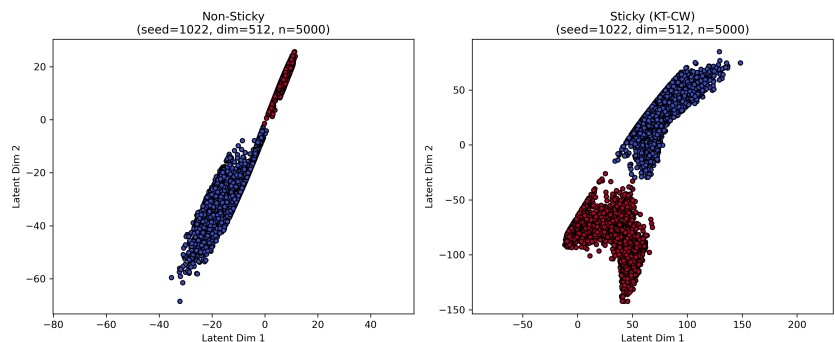

Figure 36: Sticky vs. non-sticky representations in 2D space (seed = 1022, $z_{dim}$ = 512, N = 5000).

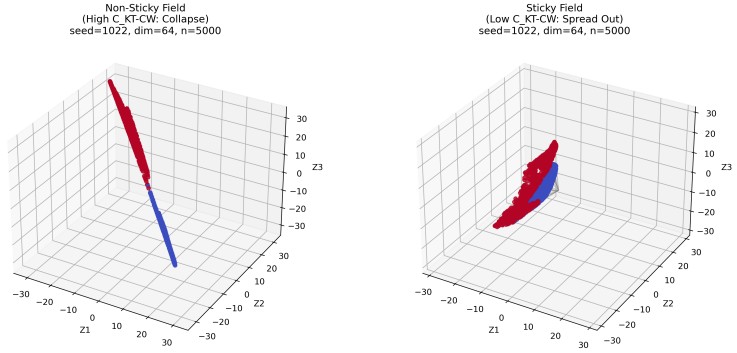

Figure 37: Sticky vs. non-sticky representations in 3D space (seed = 1022, $z_{dim}$ = 64, N = 5000).

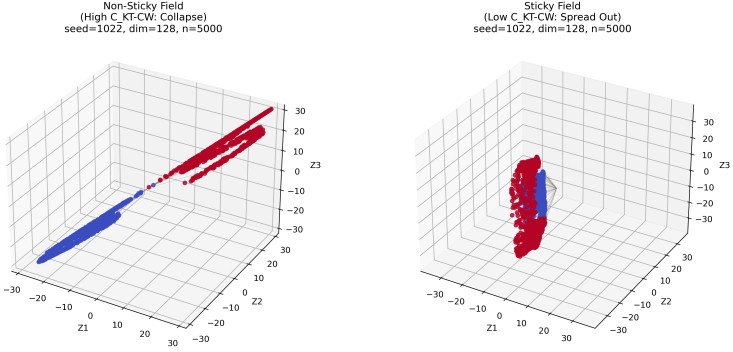

Figure 38: Sticky vs. non-sticky representations in 3D space (seed = 1022, $z_{dim}$ = 128, N = 5000).

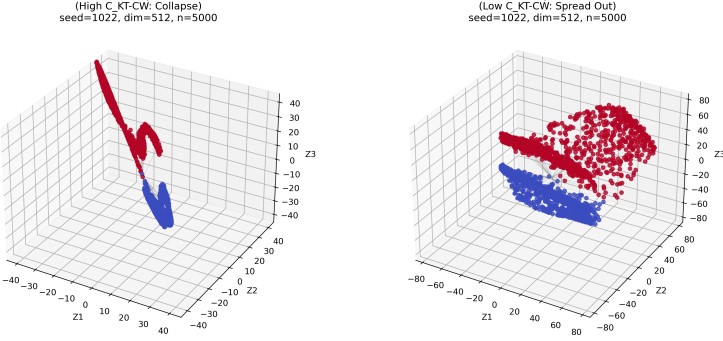

Figure 39: Sticky vs. non-sticky representations in 3D space (seed = 1022, $z_{dim}$ = 512, N = 5000).

# H  LIMITATIONS AND BROADER IMPACTS

## H.1  LIMITATIONS AND FUTURE WORK

While this paper introduces a foundational, theoretically-grounded framework, several limitations in its current form highlight important directions for future research. One limitation is the computational feasibility of the proposed KT-CW regularizer. The process of estimating the Feature Collapse Constant requires sampling random convex sets and counting the contained feature tubes within high-dimensional representation spaces, presenting a considerable computational challenge. Therefore, future research efforts must prioritize the development of highly efficient, scalable, and unbiased estimators for the Wolff axiom constants, which would make the regularizer more applicable for training large-scale models.

Furthermore, the scalability of the geometric intuitions underlying this work needs further exploration. The geometric arguments were inspired by the Kakeya conjecture, recently resolved in three dimensions. A pivotal open question arises regarding how effectively these low-dimensional geometric concepts, such as "tubes" and "slabs," can be adapted to the extremely high-dimensional spaces ($d \gg 3$) typical of neural network representations. While the mathematical definitions are dimension-agnostic, further theoretical inquiries are essential to investigate the behavior and accuracy of the Wolff axioms in these high-dimensional contexts.

Lastly, although the framework is described as "architecture-agnostic," the geometric assumptions may exhibit varying suitability across different architectures. For instance, the local feature processing inherent in CNNs may generate distinct geometric structures in the representation field, in contrast to the global, set-based processing seen in Transformers. Future research should investigate how emergent geometries vary across different architectural families and whether certain architectures are inherently better suited for learning $K$-sticky representations.

## H.2  BROADER IMPACTS

This research aspires to bridge the gap between the empirical successes in deep learning and a deeper theoretical understanding of its core properties, ultimately fostering the development of more reliable and trustworthy AI systems. By offering a theoretical framework and practical tools designed to enhance model robustness, this research directly addresses the fragility that currently limits the deployment of AI systems in safety-critical domains. Success in this domain could lead to the creation of more reliable autonomous vehicles, more accurate medical diagnostic tools that are resilient to noise and domain shift, and more stable financial prediction models. By shifting the focus from empirical heuristics to provable geometric properties, this work contributes foundational science necessary for establishing AI systems that merit public trust.

Moreover, this research aims to catalyze a new direction within the AI research community, situated at the intersection of harmonic analysis, geometric measure theory, and deep learning. It provides a new set of analytical tools, the representation field and multi-scale Wolff axioms, for diagnosing sources of non-robustness in existing models. The geometric reinterpretation of concepts such as the IB and attention mechanisms may also inspire new research initiatives and architectural innovations, including proposed "geometric attention" layers. Ultimately, the framework presented herein offers a new language and a novel perspective for understanding the mechanisms of generalization in deep learning models, which could stimulate further theoretical advancements in the field.

