# OpenReview forum: "From Kakeya to Kernels: A Multi-Scale Geometric Framework for Robust Representation Learning"
_ICLR.cc/2026/Conference — Submitted to ICLR 2026_

### Official Review · Reviewer_qLBx · 2025-10-28

**Soundness:** 1
**Presentation:** 1
**Contribution:** 2
**Rating:** 2
**Confidence:** 3

**Summary:**

The paper connects definitions used in proving the Kakeya set conjecture in three dimensions with representations of neural network models.

The work presents a theorem which they claim link representational sparseness to Lipschitz continuity.

The paper also suggests a new regularizer for use in training models.

**Strengths:**

**S1:** It is theoretically interesting to analyse machine learning concepts using tools from seemingly distinct mathematical areas.

**Weaknesses:**

**W1:** The paper is too incoherent. There are many concepts which are not properly defined and also assertions which are introduced
and then never used. I list some of the most serious cases here but see also questions **Q1-Q4,Q7,Q8,Q12**
- The paper refers to the Kakeya set conjecture multiple times ( e.g. the title and lines 16, 56, 120, 128, 219, 316, 462, 477),
however there is never any definition or explanation of what the Kakeya set conjecture is.
- In the abstract (line 17) it says "The concept of a representation field is formalized, which models feature activations as
geometric entities". However, the definition (definition 7) says that a representation field consists of one tube per representation
of an input. There is no explanation of why the representation of an input should be interpreted as a feature. See also question **Q8**.
- Section 3 considers the application of grains decomposition to a representation field, but there is no definition of what a
"grains decomposition" is.
- The paper introduces inductive volume estimates (line 151-159), but then never uses them for anything in the main article.
- The illustrations of representation fields in figures 1, 2 and 3 show the tubes originating at various points.
But in the definition 7, all tubes are defined as starting from the origin.
- A definition of (K)-"stickyness" is not presented before definition 8 on page 6, but the paper mentions stickyness many times before that
e.g. lines 74-80, 161-163, 292-293, figure 2. This is confusing. Especially the "sticky" vs "non-sticky" figure before there has been a
definition of what "sticky" means.


**W2:** The paper introduces the "KT-CW regularizer" line 89 and proposition 1. However, there is no implementation of this or even a
description of how one might implement this. See question **Q11**.

**Questions:**

Because of the lacking definitions and explanations I recommend rejection.


**Questions:**

**Q1:** Why is the title "FROM KAKEYA TO KERNELS" when kernels are not mentioned in the rest of the paper?

**Q2:** Why do you introduce inductive volume estimates (line 151-159), when they are not used in the rest of the main article?

**Q3:** In line 191-192, the paper defines $\mathcal{T}$ as a set of deformation operators $\tau:\mathcal{X} \to \mathcal{X}$,
although it was defined as a collection of tubes ($\delta$-neighbourhoods of line segments) in line 131. When the tubes are used
for defining a representation field, they become subsets of the representation space. In other words, these are two very different
things and should therefore not share the same notation.

**Q4:** Equation 3: What do you mean by $\Vert \tau \Vert$. Do you mean $\Vert x - \tau(x) \Vert$?

**Q5:** Definition 7: The representation field depends on a specific dataset. This should be more clear in the notation.

**Q6:** Axiom 1, 2: The paper says the constant $C_{KT-CW}$ of the representation field measures the degree of "feature clustering".
	But why do you need to introduce tubes for this? Could you not just as well define $\delta$ neighbourhoods around all representations
	and then say something about the number of representation neighbourhoods which can fit in a convex set?

**Q7:** Remark 1, Line 254-264:
- Here the paper mentions the "IB principle", "IB framework" and "IB objective". What does the abbreviation IB mean?
- It says: "This geometric sparsity is a tangible signature of an information-theoretically simple, or compressed, representation."
	But isn't sparse the opposite of compressed?
- It also says: "By grounding the notion of complexity in a stable geometric measure". But your definition of representation field
	depends on the dataset. You could find that representations are very sparse for one dataset, but completely collapsed on another.
	In what way do you mean it is stable?


**Q8:** Lemma 1: "Applying the grains decomposition algorithm to a representation field $\mathcal{T}_l$ is equivalent to a data-dependent,
unsupervised clustering of the feature tubes". However, the representation field is defined as tubes connecting the origin with the
representations of specific inputs. Why should we interpret every representation of a an input as a feature?



**Q9:** Definition 8: Why is there no dependence here on $\delta$? In line 420-421, it says:
"preservation of structural integrity across scales (“stickiness”)", and the scales were $\delta$ (line 130).


**Q10:** Theorem 1 (The Sticky Representation Theorem): How can you say something about Lipschitz continuity in general for a function
using only the representations from a finite dataset?


**Q11:** Proposition 1 (KT-CW Regularizer): How would you implement finding the infimum needed for $C_{KT-CW}(\mathcal{T}_l)$?


**Q12** In Appendix B "Multi-scale analysis" is paired with "Hierarchical feature extraction across layers", but in Definition 1, it says:
"A scale is a small positive number $\delta$" and "The analysis is fundamentally multi-scale, relating the properties of objects at a fine
scale $\delta$ to those at a coarser scale". How is this related to considering the representations of models across layers?

---

> ### Author Response · Authors · 2025-11-24
> **Rebuttal [part 1]**
>
> We sincerely thank you for your rigorous and detailed examination of our paper. We value the critique regarding the coherence, definitions, and notational consistency. We acknowledge that in bridging harmonic analysis and deep learning, we introduced terminology that was not sufficiently defined in the main text. We are committed to fixing these presentation issues in the revision.
>
> **Summary of key clarifications:**
>
> You raised concerns about the paper’s coherence, missing definitions (specifically regarding the Kakeya conjecture and grains), inconsistent notation, and the practical implementation of the regularizer. We would like to clarify these points:
>
> 1.	Definition and title: We apologize for the omission of formal definitions for the Kakeya Conjecture and grains decomposition in the main text, which we have now integrated. “Kernels” in the title refers to the integral kernels in harmonic analysis used to derive the volume estimates. We added a definition in Appendix D.
>
> 2.	Why tubes matter: We clarify that modeling features as tubes (rather than standard $\epsilon$-balls) is important because **tubes capture directionality.** Feature collapse is often a directional phenomenon (alignment to a subspace) that ball-based density metrics cannot detect.
>
> 3.	Implementation and theory: We clarify that the “inductive volume estimates” are not unused but are **the mathematical engine deriving the proof of Theorem 1.** Regarding the KT-CW regularizer, we clarify that we do not compute the intractable exact infimum; instead, we use a **stable stochastic approximation** by minimizing soft occupancy counts in sampled random convex sets (as detailed in **Appendix F.4 and the newly added Section 6 and Appendix G**).
>
> **Point-to-point clarifications:**
>
> *1. On incoherence and missing definitions*
>
> We accept the critique that several terms were used before formal definition. Below are the detailed responses:
>
> - The Kakeya Conjecture: We added **a formal definition in the introduction**: “The Kakeya set conjecture posits that a set of points in $\mathbb{R}^n$ containing a unit line segment in every direction must have Hausdorff dimension $n$.” Our work uses the quantitative estimates associated with this conjecture (Wolff Axioms) to measure feature representation capacity.
>
> - “Kernels” in the title: We acknowledge this term may be confusing in a standard ML context. It referred to **the integral kernels used in the harmonic analysis to derive the volume estimates.** We added a definition in Appendix D to avoid confusion.
>
> - Visuals (origin of tubes): We agree with you. Definition 7 defines tubes as radial (origin-to-point). **We have updated Figure 1-3 accordingly in the revision.**
>
> *2. On implementation (KT-CW regularizer)*
>
> We clarify that the implementation relies on **stochastic approximation.** We do not compute the exact infimum. Instead, we sample a batch of random convex sets $W_{\theta}$ and minimize the “soft” occupancy of tubes within them. This is a standard Monte Carlo style approximation for geometric integrals, making it feasible for training. In our revision, we added **Section 6 (Simulations and Empirical Validations) and Appendix G (Additional Experiment Details and Results).** These demonstrate that the KT-CW regularizer is feasible to implement, and the proposed geometric framework is useful.

---

> ### Author Response · Authors · 2025-11-24
> **Rebuttal [part 2]**
>
> *3. Why “Kernels?”*
>
> It referred to the integral kernels in harmonic analysis used to derive the volume estimates. We added a definition in Appendix D to avoid confusion.
>
> *4. Why introduce inductive volume estimates if not used?*
>
> They are the mathematical engines that derive the proofs. Theorem 1 relies on the fact that sticky representations have large-volume estimates, which prevent the geometric collapse that leads to instability. We discussed this in Section 3.2 (Remark 1).
>
> *5. Notation $\mathcal{T}$ (tubes vs. deformations)?*
>
> Thank you for pointing this out! This was an oversight. We have used $\mathcal{T}$ exclusively for tubes and $\mathcal{V}$ and $\nu$ exclusively for deformations.
>
> *6. Meaning of $||\tau||$?*
>
> It refers to the magnitude of the deformation path length, specifically the integral of the vector field norm generating the flow (Eq. 17).
>
> *7. Dataset dependence in Definition 7?*
>
> Thank you for pointing this out! We have updated the notation to $\mathcal{T}_l(D)$ to show the dependence on the dataset $D$ explicitly.
>
> *8. Why tubes? Why not neighborhoods (balls)?*
>
> This is the central insight of our work. **Balls measure density (dimension), but tubes measure directionality (alignment).**
>
> - A set can be sparse in terms of volume (balls) but still “collapse” onto a lower-dimensional subspace (line).
>
> - Kakeya analysis is uniquely powerful because it detects when features “align” too much (high $C_{KT-CW}$), which balls cannot detect.
>
> *9. IB Principle and sparsity vs. compression?*
>
> - IB: IB stands for **information bottleneck**, which we have introduced in Section 1.1 (Related Work).
>
> - Sparsity vs. compression: In geometric measure theory, “sparsity” (low occupancy counts) implies that the set occupies a large volume in the ambient space (is not compressed into a subspace). However, we argue that low $C_{KT-CW}$ (geometric sparsity) implies **low complexity in the sense of “non-redundancy.”**
>
> - Stability: We mean “stable” in that the geometric property (stickiness) holds across small perturbations of the dataset, unlike mutual information estimates, which are notoriously high-variance.
>
> *10. Input representation as a feature?*
>
> In deep learning, the intermediate activation vector $z = f_l(x)$ is standardly referred to as the “feature” or “representation” of input $x$.
>
> *11. Dependence on $\delta$ in Definition 8?*
>
> The definition of $K$-stickiness implies a uniform bound. A representation is $K$-sticky if $C_{KT-CW} \leq K$ holds for all scales $\delta$ within the valid range (0,1].
>
> *12. Lipschitz continuity from finite data?*
>
> Theorem 1 bounds the Lipschitz constant of the function restricted to the neighborhood of the data manifold defined by the deformations $\mathcal{V}$. It guarantees local stability around the training manifold, which is the definition of robust generalization.
>
> *13. Implementing the infimum?*
>
> We do not compute the exact infimum. As detailed in the paper, we sample a batch of random convex sets $W_{\theta}$ and minimize the “soft” occupancy of tubes within them. This is a standard Monte Carlo-style approximation for geometric integrals, making it feasible for training. We also added **Section 6 (Simulations and Empirical Validations) and Appendix G (Additional Experiment Details and Results)** to demonstrate the feasibility of implementation.
>
> *14. Multi-scale ($delta$) vs. Layers?*
>
> These are orthogonal axes.
>
> - Layers ($l$): The sequence of transformations.
>
> - Scale ($\delta$): The resolution at which we analyze the geometry at a specific layer.
>
> Stickiness requires the geometry to be “good” (non-collapsed) at every layer, and at every resolution $\delta$.
>
> We hope the above answers your questions and addresses your concerns. Thank you again for your helpful and constructive comments!

---

> ### Comment · Reviewer_qLBx · 2025-11-26
> **Reply to Author response**
>
> Thank you for your response and clarifications.
>
>
> I have now read the updated version of the paper and have some comments to that and your response.
>
> W.r.t. your response:
>
> **Summary**
>
> 1. Missing formal definitions: I see the definition of the Kakeya conjecture (footnote page 2).
> 	However, I still do not see any definition of "grains decomposition" in the main text.
> 	The sentence in lines 268-269 serves as an intuition of what it is about, but a more formal definition is needed to
> 	understand why this should serve as a clustering method.
>
> 2. Why tubes matter: You write in your response that tubes are important because they capture directionality.
> 	However, in the paper you do not make clear why direction should be important. For example, in line 246-250 it sounds like
> 	the only thing that matters is whether the representations are "contained within the same small convex regions of the
> 	representation space". If directionality is important you should make this clear in the text.
> 	See also question about dimension in 8.
>
> 3. You write: "the “inductive volume estimates” are not unused but are the mathematical engine deriving the proof of Theorem 1".
> 	However, in lines 152-166 where you introduce Assertion 1 and 2, you write why these assertions are important to the Kakeya
> 	conjecture, but you do not write anything about why they are important for your result (Theorem 1).
> 	Also, when you introduce the theorem (lines 320-329) you do not write anything about how you use Assertion 1 and 2 or why
> 	they are important for this result.
> 	In point 4. you write that you "discussed this in Section 3.2 (Remark 1)", but I don't see any mention of Assertion 1 and 2
> 	in this section either.
>
>
> **point-to-point**
>
> w.r.t. 6. Meaning of $\Vert v \Vert$ At the very least you should include a reference to where you define this (eq. 17).
> It would be better if you could include the definition in the main article.
>
>
> w.r.t. 8. Why tubes? Why not neighborhoods (balls)?
> 	You write: "A set can be sparse in terms of volume (balls) but still “collapse” onto a lower-dimensional subspace (line)."
> 	Do you care about volume or do you care about dimension? Your Katz-Tao Convex Wolff (CKT−CW) Axiom seems to be about volume,
> 	but both in this response and in lines 165-166 "forcing the configuration into a lower-dimensional subspace", lemma 1 "co-linear",
> 	line 317 "do not collapse into redundant, low-dimensional configurations", it sounds like what you care about is the dimensionality
> 	of the representations? This should be made much clearer in the article.
>
>
> Remember you can also have a higher dimensional convex shape with a smaller volume than a lower dimensional
> 	shape. Consider the unit n-ball in 5 vs 20 dimensions. So enforcing high volume, will not necessarily enforce high dimension.
>
>
> You also write the analysis is powerful because it "it detects when features “align” too much".
> I assume by "align" you mean close in terms of angle? But if representations are well separated for all layers in a network, why should it matter whether they become close in terms of angle in for example a middle layer?

---

> > ### Comment · Reviewer_qLBx · 2025-11-26
> > **Additional feedback**
> >
> > **Feedback after reading new version**
> >
> > I include here questions which came up while I read the new version of the article in the hope that it will be useful by making it more clear
> > which parts of the paper cause confusion.
> >
> >
> > **F1:** In the introduction (lines 37, 41, 52 and 118) and in the conclusion (line 532) you suggest that your result is motivated
> > by understanding generalization and out-of-distribution behaviour. I understand how the result relates to robustness,
> > but how does it relate to generalization and out-of-distribution behaviour?
> >
> >
> > **F2:** Line 80-81: "correlating internal geometric properties with external functional performance."
> > 	Do you mean performance here in terms of e.g. accuracy or loss? Otherwise, I would write "robustness".
> >
> >
> > **F3:** Line 91-92: "this regularizer converts theoretical insights into a potential training tool for deep networks."
> > 	It would be good to add here exactly what this means. That is, if we train a model with and a model without the regularizer,
> > 	the model with will be more [insert quality here] than the model without.
> >
> >
> > **F4:** Footnote with Kakeya set conjecture: This does not say that you _need_ a line segment in every direction to have a dimension of n. Also, I would guess "direction" here does not have to be from the origin?
> >
> >
> > **F5:** Equations 1 and 2. I was very confused about the difference between $\vert \mathcal{T} \vert$ and  #$\mathcal{T}$.
> > 	Now I think you have an error in equation 1. I think it should be $\vert T \vert$, so the volume of a single $\delta$-tube.
> > 	Which is fine when you are working with unit line section tubes, but which tube do you choose when you switch to working with the representation field? The tubes in the representation field have different volumes.
> >
> >
> > **F6:** Line 155 Assertion 1: Definitions of $\kappa$ and $\epsilon$ are missing. Also, here you again use the volume of $T$,
> > 	which $T$ do you choose when the volumes differ as in your definition 7?
> >
> >
> > **F7:** Line 163: "remains stable and well-behaved across various scales". It is not clear to me what you mean by "stable" and
> > 	"well-behaved" here.
> >
> >
> > **F8:** Lemma 1: Here you write that each grain "corresponds to a cluster of feature activations that are geometrically proximate
> > 	and co-linear." However, since you defined the tubes as going from the origin to the representations, we will only get co-linearity
> > 	if the representations are exactly on top of each other in terms of angle. This is not very likely to happen.
> > 	Also is this supposed to be illustrated in figure 2 bottom left? Here again the tubes do not seem to start at the origin.
> >
> >
> > **F9:** Definition 8: Your definition of sticky only relies on the KT-CW axiom. Could you explain how this relates to the "stickyness"
> > 	defined in the Kakeya conjecture, because they seem different to me?
> >
> >
> > **F10:** Lines 316-317: "features remain well-structured and do not collapse into redundant, low-dimensional configurations"
> > 	(related to dimension question 8.). What if volume increases and decreases over the layers, but the dimension stays the same?
> >
> >
> > **F11:** Theorem 1: The function $g$: Is this a specific function, which you find? Or do you just prove that there exists such a function?
> >
> >
> > **F12:** Theorem 2 (1): Where did the dependence on input deformations go?

---

> > > ### Author Response · Authors · 2025-11-27
> > > **Rebuttal to Additional Feedback [part 1]**
> > >
> > > Thank you again for the opportunity to provide clarifications! We address each of your questions below.
> > >
> > > *1. F1: Relation to generalization/OOD?*
> > >
> > > The link between our results and generalization is established through **Proposition 2 (Appendix E.4).** In this proposition, we derive a PAC-Bayes generalization bound explicitly using the geometric stickiness metric. The logic is as follows: A sticky representation has a bounded Lipschitz constant $g(K)$ (Theorem 1), which allows us to derive a larger safe margin against perturbations. In the PAC-Bayes framework, this reduced functional complexity ($g(K)$) directly translates into a tighter bound on the generalization error. Since this is not the main contribution we claim, we would like to keep this in the Appendix and refer to it in the main text.
> > >
> > > *2. F2: Definition of “performance” (Lines 80-81)?*
> > >
> > > Thank you for pointing this out! We specifically mean **functional robustness and stability** rather than clean accuracy. In the latest version, we have revised the text to “correlating internal geometric properties with external robustness guarantees.”
> > >
> > > *3. F3: Meaning of “tool” (Lines 91-92)?*
> > >
> > > By “tool,” we mean training with the **KT-CW regularizer.** This produces a model that is provably more Lipschitz-stable (Theorem 1) and empirically more resistant to feature collapse than a model trained without it.
> > >
> > > *4. F4: Kakeya footnote accuracy?*
> > >
> > > We appreciate the scrutiny here. To clarify, the definition in the footnote is the standard statement of the Kakeya Set Conjecture. **It states the sufficiency condition:** if a set contains a unit line segment in every direction, then it must have Hausdorff dimension $n$. It does not imply the converse that all dimensionality-$n$ sets require such segments. Furthermore, we agree that in the general Kakeya problem, segments are translation-invariant. However, in our Definition 7 (Representation Field), we purposely adapt this to a “radial” setting where tubes are anchored at the origin to model the magnitude and direction of feature vectors $z$. In the latest revision, we added a footnote on page 3 to clarify that while the conjecture is translation-invariant, our framework applies these geometric tools to the specific case of origin-anchored feature vectors.
> > >
> > > *5. F5: Error in Eq. 1 ($|\mathcal{T}|$ vs. #$\mathcal{T}$)?*
> > >
> > > Thank you for identifying this typo! The term $|\mathcal{T}|^{-1}$ in Eq. 1 was intended to be #$\mathcal{T}$ (the total cardinality of the set of tubes), which matches the form of Eq. 2.
> > >
> > > - Unified density bound: With this correction, both axioms enforce a density bound: the number of tubes in a region $W$ is bounded by #${T \in W}$ $\leq C$ $\cdot |W| \cdot$ #$\mathcal{T})$.
> > >
> > > - Resolving variable volumes: By defining the bound in terms of the count #$\mathcal{T}$ rather than the single tube volume $|T|$, we resolve your concern about the representation field containing tubes of different volumes (lengths $||z||$). The geometric properties determine whether a tube falls inside $W$, but the normalization constant relies on the total count, which is well-defined regardless of individual tube variations. **In the latest revision, we corrected Eq. 1 to match this consistent notation.**
> > >
> > > *6. F6: Definition of $\kappa$, $\epsilon$, and tube choice?*
> > >
> > > **$\kappa$ and $\epsilon$ are standard positive constants from the Kakeya literature quantifying the volume lower bound.** Regarding the volume $|T|$ in Assertion 1: The Kakeya framework relies on “induction on scales.” In the formal proofs (Appendix F), we partition the Representation Field into dyadic length scales. Within each dyadic band, the tube volumes are uniform up to a constant factor, allowing the assertions to hold for that specific scale. For instance, in Appendix F.2, we handle variable volumes by restricting the analysis to a fixed-scale regime ($\delta \ll \rho \leq 1$). This acts analogously to a dyadic partition: within this bounded regime, the tube length and thus volumes $|T|$ are uniform up to a constant factor controlled by the layer-wise Jacobian $||J_f||_\infty$. This allows the Wolff axioms to be applied with a single representative volume scale, absorbing the variation into the constant $c_l(K)$.

---

> > > ### Author Response · Authors · 2025-11-27
> > > **Rebuttal to Additional Feedback [part 2]**
> > >
> > > *7. F7: Stable and well-behaved?*
> > >
> > > We use these terms with specific geometric meanings:
> > >
> > > - Stable: The dimensionality (rank) of the set does not degrade or collapse as the scale $\delta\rightarrow0$.
> > >
> > > - Well-behaved: The set satisfies the Wolff non-clustering axioms (Eq. 1 and 2), meaning it does not contain “pencils” (dense clusters) or planar concentrations.
> > >
> > > **In the latest revision, we have added explanations in Lines 165-167.**
> > >
> > > *8. F8: Grains, origin, and co-linearity?*
> > >
> > > - Co-linearity: Yes, **co-linear here means the feature vectors point in the same direction (cosine similarity $\approx$ 1).** This is exactly what happens during feature collapse (mode collapse). The grains decomposition detects these clusters of highly correlated features.
> > >
> > > - Figure 2: The tubes structurally anchor at the origin. However, the illustration in Figure 2 is a “zoomed-in” local view of a grain acting as a bounding box for the ends of the tubes away from the origin.
> > >
> > > *9. F9: Definition of sticky vs. Kakeya?*
> > >
> > > As we mentioned earlier, our definition is a direct translation. In Kakeya theory, a set is defined as “sticky” if it satisfies the Wolff axioms at all scales. Our Definition 8 applies this exact condition, bounding the Feature Collapse Constant ($C_{KT-CW}$), to the representation field. The stickiness is the property of obeying the axioms.
> > >
> > > *10. F10: Volume vs. dimension (layer-wise)?*
> > >
> > > If the dimension stays the same, we are satisfied. The danger we address is when the dimension drops (collapses). Our axioms prevent the volume from becoming “too small” for the given ambient dimension. **If the volume fluctuates but remains bounded away from zero (relative to the scale), the dimension is preserved.** The axioms act as a barrier against the volume shrinking to the point of rank collapse.
> > >
> > > *11. F11: Theorem 1 function $g(K)$?*
> > >
> > > **We prove the existence of such a function and derive its form.** Specifically, Eq. 31 in Appendix F.2 shows that $g(K) = \Pi_{l=1}^L c_l(K)$, where $c_l(K)$ are layer-wise constants derived from the Jacobian bounds.
> > >
> > > *12. F12: Theorem 2 deformations?*
> > >
> > > The dependence on input deformations is captured in the term $dist_{\mathcal{V}}(\nu)$ in Eq. 37. The bound is linear in the magnitude of the input deformation measured by the path integral of the vector field.
> > >
> > > We hope these answer your questions. Thank you again for being interested in our work!

---

> > ### Author Response · Authors · 2025-11-27
> > **Rebuttal to 2nd Round [part 1]**
> >
> > Thank you for your detailed engagement with the updated manuscript and for highlighting specific areas where the connection between the Kakeya geometry and the machine learning implementation requires further clarification. We value this feedback and will incorporate these clarifications into the final version. Below, we address your questions regarding the definitions, the geometric motivation (tubes vs. balls), and the specific line-item feedback.
> >
> > **Key clarifications:**
> >
> > *1. Missing formal definitions (grains decomposition)*
> >
> > We agree that the main text relies on intuition for readability. However, the formal mathematical definition and construction of the “grains decomposition” are provided in Appendix F.1. We formally define grains as **rectangular prisms of longitudinal length 1 and cross-sectional diameter $\rho$ with specific coherence and incidence properties.** In the latest revision, we added a definition in Section 3.3.
> >
> > *2. Why tubes matter (vs. balls) & directionality?*
> >
> > Thank you for bringing up this question! We utilize tubes rather than balls because feature collapse in deep learning is inherently directional.
> >
> > - Why directionality: In high-dimensional spaces, features often collapse into lower-dimensional subspaces (e.g., a line or a flat plane). **A ball (neighborhood) is isotropic; it does not capture the orientation of this collapse.** A collection of $\delta$-tubes, however, has an axis.
> >
> > - Line 240-250: When we discuss “contained within the same small convex regions,” this implicitly relies on directionality **because the “convex region” that captures a bundle of tubes is often a narrow cylinder or slab aligned with those tubes.** If the tubes were not directionally aligned, they could not be packed into such a small convex volume.
> >
> > *3. The role of inductive volume estimates (Assertions 1 & 2)*
> >
> > Your question is about how Assertions 1 and 2 relate to Theorem 1.
> >
> > - The logic chain: **Assertions $D$ and $E$ (the volume estimates) are mathematical machinery that allows us to prove that satisfying the Wolff Axioms leads to a lower bound on the volume of the representation field.** This volume lower bound is what prevents the Jacobian from blowing up (as shown in Eq. 24-25 of Appendix F.2).
> >
> > - Clarification: We define stickiness as satisfying the Wolff axioms. The consequence of stickiness is that Assertion $D/E$ holds, thereby guaranteeing that the representation has high volume/dimension. This prevents the Lipschitz constant from exploding. **In the latest version, we have added a sentence in Section 3.2 stating: “Satisfying these axioms ensures Definition 3 holds, which forms the basis of the Lipschitz bound in Theorem 1.”**
> >
> > *4. Volume vs. dimension*
> >
> > We care about dimensionality (rank), but **we use volume as a proxy to measure it.** As you noted, a set can have low volume but high dimension. However, in the context of feature collapse, the problem is the opposite: representations collapse into a low-dimensional subspace (low rank). The Kakeya “stickiness” property ensures that the set of tubes cannot be compressed into a low-dimensional subspace. By enforcing the Wolff axioms (which bound volume density), we essentially force the features to splay out (as seen in Figure 4), thereby maximizing their effective dimension.
> >
> > *5. Alignment in middle layers*
> >
> > Your question is about why alignment matters in the middle layers if the final layer is separated. Deep networks process data hierarchically. **If features “align” (collapse) in a middle layer (geometric bottleneck), the geometric information lost there is irrecoverable by subsequent layers (known as Data Processing Inequality).** Furthermore, Theorem 1 shows that the global Lipschitz constant is a product of layer-wise constants. A collapse (alignment) in a middle layer spikes the local Lipschitz constant, degrading the robustness of the entire network.

---

> > ### Author Response · Authors · 2025-11-27
> > **Rebuttal to 2nd Round [part 2]**
> >
> > **Point-to-point clarifications:**
> >
> > *1. W.r.t. 6. Meaning of $||\nu||$?*
> >
> > We agree that the definition requires earlier referencing. $||\nu||$ represents the magnitude of the deformation operator. In the current text, **it is introduced in Definition 6 (Lipschitz Stability and Functional Robustness),** specifically in Eq. 3, where it is defined in the context of the deformation class $\mathcal{V}$.
> >
> > *2. W.r.t. 8. Why tubes? Why not neighborhoods (ball)? Volume vs. dimension?*
> >
> > - Why tubes: We choose tubes because **feature collapse in deep networks is inherently anisotropic (directional).** When representations collapse, they typically collapse onto a lower-dimensional manifold (such as a line or a hyperplane) rather than shrinking uniformly into a small ball. **A ball neighborhood is isotropic and cannot distinguish between a “thin” representation (collapsed) and a “small” representation (localized).** A $\delta$-tube, having an axis, naturally captures this directional geometry.
> >
> > - Volume vs. dimension: We fundamentally care about dimension (rank). However, we use **volume as the measurable proxy to enforce it.** You rightly point out that high-dimensional balls can have small volumes. However, in the specific context of representation learning, the pathology we are fighting is “Mode Collapse,” where activations confined to a high-dimensional ambient space (e.g., $\mathbb{R}^{512}$) degenerate into a low-dimensional subspace (e.g., $\mathbb{R}^2). If a set of tubes satisfies the Katz-Tao Convex Wolff Axiom ($C_{KT-CW}$), it is geometrically impossible for them to be packed into a low-dimensional subspace. The axiom enforces a “volume density” constraint that effectively pushes the tubes to splay out (as visualized in Figure 4), thereby maintaining a high effective dimension. In the latest version, we have revised the text in Section 3.2 to state: “While our axioms constrain volume density, their primary purpose in this framework is to serve as a geometric proxy for maintaining effective dimensionality and preventing rank collapse.”
> >
> > *3. W.r.t. “Align” and middle layers?*
> >
> > - Meaning of align: By “align,” we specifically mean directional coherence (high cosine similarity or small angular distance between feature vectors), as formalized in Lemma 1, which discusses grains as clusters of co-linear features.
> >
> > - Why middle layers matter: First, Theorem 1 establishes that the product of layer-wise stability constants bounds the global Lipschitz constant of the network. **If features “align” (collapse) in a middle layer, the local Lipschitz constant $c_l$ for that layer spikes.** This multiplicative degradation reduces the robustness of the entire network, regardless of separation in the final layer. Second, from the standpoint of information bottleneck (IB), a geometric collapse in an intermediate layer represents **an irreversible loss of information via the Data Processing Inequality.** Suppose the geometry collapses to a low-dimensional subspace in layer $l$. In that case, the network loses the capacity to distinguish perturbations orthogonal to that subspace in subsequent layers, creating blind spots that adversarial attacks can exploit. In Section 3.4, we mentioned that $K$-stickiness is a layer-wise condition designed to prevent intermediate bottlenecks that amplify downstream instability.
> >
> > We hope these clarifications are helpful. Thank you again for your constructive feedback!

---

### Official Review · Reviewer_GQCC · 2025-10-31

**Soundness:** 2
**Presentation:** 3
**Contribution:** 3
**Rating:** 6
**Confidence:** 2

**Summary:**

This paper introduces a new geometric framework for robust representation learning. The work applies methods from multi-scale analysis techniques to quantify how susceptible models are to minor input perturbations. The model defines learned representations as a collection of tubes in the representation space, and relates various aspects of model performance to the convex sets and subspaces that contain these tubes. It is then shown how a geometric property of these representations called K-stickiness is tied to robustness and performance in multimodal DNN models. Insights from this model are then used to design a regularization term that can help prevent feature collapse and misalignment in multimodal learning

**Strengths:**

* This paper is well written and relatively accessible to a reader without a strong background in harmonic analysis.
* Connections to several machine learning phenomena lend credence to the veracity of this framework. For example, the framework offers explanations for mode collapse in GANS, the lottery ticket hypothesis, and adversarial robustness, among other things.
* The figures in this paper are particularly high quality and help give a visual intuition for the geometric objects being discussed.

**Weaknesses:**

* This framework is proposed as a more direct method for quantifying mutual information for the information bottleneck principle. However, it's not shown that it's possible to compute the constants described in this paper. While this is an interesting model, I would like more justification as to why this is more useful in practice than existing models (like IB).
* No attempt is made to empirically evaluate the validity of the proposed geometric model. Again, I would be interested in seeing empirical validation that this model is a useful proxy for IB.
* Similarly, it seems to me that the KT-CW regularizer would not be feasible to implement. It's noted in the limitation in the appendix that this loss term might be difficult to compute, but I think that should be made more clear in the body that the implementation of this loss is left open for future research.
* I think it would be helpful to included proof sketches in the main body of the paper. As written, the reader will have little intuition as to the reasoning behind the theorems without reading the full proofs in the appendix.
* The term "performance degradation" as it is used in Theorem 2 should be more explicitly defined. It's not clear how you are expecting the model to be used.

**Questions:**

* I'm confused on the order of quantifiers in Axiom 4. Is $W$ meant to be fixed before the computation of $C$, or is the infimum taken over the ratio computed for all convex sets $W$?
* Do the choices for the $\delta$ and $\rho$ parameters affect the values of the constants that are computed? Should their values depend on the data distribution?
* How should I think about the significance of a shared Lipschitz constant in the multimodal setting? As defined in section 2, the Lipschitz constant is a function of each deformation's magnitude. Between modalities, transformations would take different forms, so a Lipschitz constant that is considered small in one modality may be considered large in another. Is the significance of Theorem 2 (1) just that all modalities exhibit Lipschitz stability for some constant which may or may not be small, or is the guarantee stronger than that?
* How do representation similarity metrics such as CCA [1] or CKA [2] relate to this framework? Are the patterns in representations that are being captured here explainable in terms of representation similarity analysis?

[1] Morcos, Ari, Maithra Raghu, and Samy Bengio. "Insights on representational similarity in neural networks with canonical correlation." Advances in neural information processing systems 31 (2018).

[2] Kornblith, Simon, Mohammad Norouzi, Honglak Lee, and Geoffrey Hinton. "Similarity of neural network representations revisited." In International conference on machine learning, pp. 3519-3529. PMlR, 2019.

---

> ### Author Response · Authors · 2025-11-24
> **Rebuttal [part 1]**
>
> We are grateful for your encouraging assessment, particularly the recognition of **our framework’s accessibility, the high quality of the visualizations, and the credibility it gains from our connections to GANs and the Lottery Ticket Hypothesis.** We appreciate the insightful questions regarding the computability of our geometric constants compared to information bottleneck (IB) and the specific mechanics of the multimodal extension.
>
> **Summary of key clarifications:**
>
> You raised concerns about the computability of our geometric constants compared to IB, the feasibility of the KT-CW regularizer, and the need for empirical validation. We would like to clarify these points:
>
> 1.	Computability vs. IB: We clarify that, unlike IB, which relies on unstable high-dimensional density estimation, our framework relies on geometric occupancy counting. This allows for stable stochastic approximation via the KT-CW regularizer (Eq. 9), which we **implement by minimizing soft occupancy counts in sampled random convex sets rather than computing exact infimums.**
>
> 2.	Empirical validation: To address your concern, we run multiple simulations, including 2D, 3D, single-modal, and multimodal, to empirically validate the core contributions. In our revision, we added **Section 6 (Simulations and Empirical Validations) and Appendix G (Additional Experiment Details and Results).** These experiments demonstrate that sticky models (enforced via our regularizer) show a full-rank singular value spectrum and significantly higher robustness to adversarial attacks and missing modalities, empirically validating the bounds predicted by Theorems 1 and 2.
>
> 3.	Definitions and intuition: We explicitly define “performance degradation” in Theorem 2 as **embedding shift** (the bounded Euclidean distance between full and partial input embeddings). Due to the page limit and the added empirical validation, we would keep the full proofs in Appendix F.
>
> **Point-to-point clarifications:**
>
> *1. On computability vs. IB*
>
> You asked a critical question about why this geometric model is more useful/computable than IB, and we answer this question from different angles as follows:
>
> - The IB challenge: IB relies on mutual information $I(X;Z)$, which requires estimating probability densities in high-dimensional spaces, a notoriously difficult task (curse of dimensionality) often leading to unstable or unbounded estimates.
>
> - The geometric advantage: Our constants ($C_{KT-CW}$) are based on **counting geometric intersections (tubes in slabs)**. While exact counting is computationally hard, approximating counts via Monte Carlo sampling (dropping random test slabs and counting hits) is far more stable and lower variance than density estimation.
>
> - Usefulness: This offers a **geometric proxy for complexity** that is robust to the sampling issues that plague IB. We clarified this distinction in **Section 3.2 (Remark 1)**.
>
> *2. On KT-CW regularizer implementation*
>
> As detailed in Proposition 1 and Appendix F.4 (Proof of Proposition 1), we define the regularizer using a “soft” occupancy measure. In practice, one does not compute the supremum over all convex sets $W$. Instead, one samples a batch of random convex sets and minimizes the softmax-occupancy within it. In our revision, we added **Section 6 (Simulations and Empirical Validations) and Appendix G (Additional Experiment Details and Results).** These demonstrate that the KT-CW regularizer is feasible to implement, and the proposed geometric framework is useful.
>
> *3. On proof sketches*
>
> We agree that intuition is important for the theorems. However, due to the page limit and the added empirical validation, we would keep the full proofs in Appendix F. **The core intuition** relies on the Tube Push-Forward lemma (page 25 in the revision version). If a network layer has a bounded Jacobian, it maps input tubes to output tubes without significantly expanding their volume. Suppose the output representation is sticky (i.e., no clustering). In that case, we can use the grains decomposition to show that small input perturbations cannot jump between disjoint clusters, thereby bounding the Lipschitz constant.
>
> *4. On performance degradation (Theorem 2)*
>
> We apologize for the ambiguity. Below is the clarification:
>
> - Definition: In Theorem 2, “performance degradation” is mathematically defined as **the bound on the distance between the full-modal embedding and the missing-modal embedding.** We added this definition on page 8 in our revision.
>
> - Significance: It guarantees that the representation does not catastrophically shift when a modality is removed. The embedding stays within a bounded neighborhood proportional to the number of dropped modalities.

---

> ### Author Response · Authors · 2025-11-24
> **Rebuttal [part 2]**
>
> *5. Order of quantifiers in Axiom 4 ($C_{\text{align}}$)?*
>
> The supremum is over the sets $W$. Specifically, **$C_{\text{align}}$ is the infimum of all constants $C$ such that the variance condition holds for all convex sets $W$.**
>
> - Mathematically: $C_{\text{align}}$ = $sup_{W \subset Z_{\text{joint}}}$ $\big(\frac{\operatorname{Var}_m(count_m(W))}{\operatorname{Mean}_m(count_m(W))}\big)$.
>
> - It captures the worst-case alignment: if there exists even one region $W$ dominated by a single modality, $C_{\text{align}}$ explodes.
>
> *6. Do $\delta$ and $\rho$ depend on data distribution?*
>
> Yes, technically. In the formal Kakeya analysis, results are asymptotic ($\delta \rightarrow 0$). In the machine learning context, $\delta$ acts as a **resolution parameter** (similar to grid size or precision). If the data manifold is very intrinsic, we require stickiness to hold for a range of scales $\delta \ll \rho \leq 1$. The constants computed depend on $\delta$, but the definition of a $K$-sticky representation requires the bound to hold uniformly across these scales.
>
> *7. Significance of shared Lipschitz in multimodal settings?*
>
> You are correct that modalities have different units of perturbation. The theorem accounts for this by defining a modal deformation distance ($\text{dist}_{\mathcal{V}}(\nu)$), which sums the normalized path lengths of deformations in each modality.
>
> - The guarantee: Theorem 2 ensures that the joint embedding is stable. Even if text and image inputs deform differently, a $K$-sticky fusion network forces their joint representation to be stable. The guarantee is strong: it prevents the weakest link (most sensitive modality) from destabilizing the entire fused representation.
>
> *8. Relation to CCA/CKA?*
>
> Thank you for the question! This is an insightful connection. Please see our answers below:
>
> - CCA/CKA: These metrics measure the **similarity between two representations** (e.g., is layer 1 similar to layer 2?) by analyzing the alignment of their Gram matrices or covariance structures.
> - Stickiness: Our framework measures the **internal geometry of a single representation** (e.g., is layer 1 degenerate or collapsed?).
>
> - The link: We hypothesize that sticky representations (low collapse) would likely show high CKA similarity with other robust representations, whereas non-sticky (collapsed) representations would show low CKA similarity with robust ones because their geometric information has been lost (collapsed to a lower rank). Stickiness could be viewed as a prerequisite for a representation to have meaningful CKA similarity with a ground-truth manifold.
>
> We hope these answer your questions and are helpful. Thank you again for your constructive feedback!

---

### Official Review · Reviewer_cJhi · 2025-11-01

**Soundness:** 2
**Presentation:** 2
**Contribution:** 2
**Rating:** 4
**Confidence:** 3

**Summary:**

This paper introduces a novel geometric quantity called “stickiness”, inspired by the Kakeya set problem, to characterize aspects of neural network representations. The authors define stickiness as a measure of how much a network’s output trajectories “cling” to certain directions in input space and establish a theoretical link between stickiness and the Lipschitz continuity of the network. Conceptually, this connection bridges ideas from geometric measure theory with representation analysis in deep learning.

The paper is clearly written and well-motivated. The presentation is easy to follow, and the framing—connecting Kakeya geometry to neural representations—is both original and intellectually stimulating.

The paper presents an original and intellectually appealing idea—introducing stickiness as a geometric quantity inspired by the Kakeya problem and linking it to Lipschitz continuity in neural networks. The presentation is clear and well organized, making the theory accessible to a broad audience. However, the work currently feels incomplete: the motivation for why Lipschitz-ness is desirable in this setting is not clearly articulated, there is limited discussion of related theoretical frameworks, and the paper lacks empirical or computational illustrations that would demonstrate how stickiness behaves in practice.

With additional context connecting the theory to existing notions of smoothness or robustness, and with a simple empirical demonstration to anchor the concept, this work could become a strong theoretical contribution in a future version. As it stands, I find the idea promising but not yet sufficiently substantiated for acceptance.

**Strengths:**

* Novel theoretical concept: The introduction of stickiness as a representation property derived from Kakeya-type arguments is creative and mathematically original.
* Clear presentation: Definitions and theoretical statements are well structured and easy to follow, even without deep familiarity with geometric measure theory.
* Potentially broad relevance: The proposed framework could provide a new lens for analyzing neural smoothness, stability, or robustness through geometric quantities.
* Interesting bridge between mathematics and deep learning: The Kakeya-inspired formulation connects a deep geometric concept to neural network analysis in a way that feels fresh and principled.

**Weaknesses:**

* Unclear motivation for Lipschitz-ness: The paper emphasizes the link between stickiness and the Lipschitz constant but does not sufficiently explain why Lipschitz-ness is a desirable property in this context. Is it intended as a proxy for robustness, stability, or generalization? A short discussion of its conceptual or practical significance would strengthen the narrative.
* Lack of connection to prior work: It is unclear whether there are existing approaches relating network representations to Lipschitz continuity (e.g., via Jacobian norms, spectral bounds, or curvature regularization). Providing a short literature comparison would help contextualize the contribution.
* Missing empirical or computational illustration: While the paper is primarily theoretical, it would be valuable to include at least a small toy example (e.g., a simple MLP or 2D dataset) demonstrating how stickiness can be estimated and how it scales with network smoothness or architecture.
* Complexity considerations: A discussion of the computational cost of estimating stickiness—either analytically or numerically—would improve the paper’s practical relevance.

**Questions:**

1. What makes Lipschitz continuity a particularly meaningful or desirable property in the context of representation learning?
2. Are there existing theoretical frameworks or empirical studies connecting representation geometry to Lipschitz bounds (e.g., through Jacobian regularization or spectral control)?
3. How computationally intensive is it to estimate stickiness for modern architectures?
4. Could you include a small illustrative example showing how stickiness behaves for networks of varying smoothness or depth?

---

> ### Author Response · Authors · 2025-11-24
> **Rebuttal [part 1]**
>
> Thank you for finding our work **intellectually stimulating, creative, and fresh.** We appreciate the recognition of our primary contribution: bridging geometric measure theory (Kakeya sets) and deep learning to provide a new lens for analysis. We address the concerns regarding motivation, related work, and practical illustration below.
>
> **Summary of key clarifications:**
>
> You raised concerns regarding the motivation for Lipschitz continuity, the connection to prior theoretical frameworks, and the lack of empirical or computational illustrations. We would like to clarify these points:
>
> 1.	Lipschitz as stability: We clarify that Lipschitz continuity is not merely a proxy but the formal mathematical definition of functional stability against input perturbations. Our contribution is linking this functional stability directly to the internal geometric “stickiness” of the representation.
>
> 2.	Distinction from prior work: While we acknowledge existing spectral and Jacobian bounds, we clarify that our framework offers a geometric alternative that constrains the spatial arrangement of feature activations (tubes) rather than relying on algebraic weight norms.
>
> 3.	Computability and illustration: We clarify that exact computation is not required. The KT-CW regularizer uses efficient stochastic approximation via random sampling of convex sets. Additionally, Figure 2 serves as a conceptual toy example, illustrating how “sticky” geometry leads to robust, non-collapsed decision boundaries. Also, in the revision, we added multiple simulations to illustrate the paper's core geometric intuition.
>
> **Point-to-point clarifications:**
>
> *1. On the motivation for Lipschitz-ness and why is it desirable?*
>
> We apologize if this link was not sufficiently emphasized. Lipschitz continuity is not merely a proxy. It is the standard mathematical definition of **stability against adversarial perturbations.**
>
> - Formal definition: As defined in **Definition 6 (Equation 3)**, the Lipschitz constant explicitly bounds the change in output $||f(x)-f(x+\delta)||$ relative to the input perturbation $||\delta||$.
>
> - Desirability: In safety-critical applications (e.g., autonomous driving), a low Lipschitz constant is the **gold standard** for certification. It guarantees that the network cannot map two visually indistinguishable inputs to vastly different outputs (a requirement for preventing adversarial attacks). Our **Theorem 1** is significant because it guarantees this property via internal geometry (stickiness) rather than external weight constraints.
> In the **revision**, we added a figure (Figure 5) to demonstrate high-dimensional FGSM robustness.
>
> *2. On connection to prior work (Jacobian/Spectral bounds)*
>
> We agree that context is vital. We have discussed this extensively in **Section 1.1** and **Appendix C.2 (Provable Guarantees for Model Robustness)** [1,2,3].
>
> - The gap: We highlight that existing methods (spectral bounds, Jacobian regularization) often yield “loose” bounds because they rely on weight-level norms (algebraic properties).
>
> - **Our contribution:** We offer a geometric alternative. Instead of constraining weights (which limits expressivity), we constrain the arrangement of activations (tubes). This captures the “emergent geometry” of the data manifold directly, offering a tighter conceptual link to how data actually flows through the network. We will make this comparison more prominent in the main text.
>
> *3. On empirical/computational illustration*
>
> While this is a theory track submission focused on establishing the existence of the bridge between Kakeya sets and robustness, we agree that a mental model is helpful. We run multiple simulations, including 2D, 3D, single-modal, and multimodal, to empirically validate the core contributions. In our revision, we added **Section 6 (Simulations and Empirical Validations) and Appendix G (Additional Experiment Details and Results).**
>
> *4. On complexity considerations*
>
> You raised a valid point: calculating the exact volume of high-dimensional intersection sets is computationally expensive.
>
> - The solution: This is exactly why we derived the **KT-CW regularizer (Eq. 9)**. We do not need to calculate the exact stickiness constant $K$ during training. Instead, we minimize a stochastic upper bound using random sampling of convex test sets (slabs/boxes).
>
> - This is analogous to the information bottleneck (IB): calculating mutual information $I(X;Z)$ is intractable, so practitioners use variational approximations (VIB). Our regularizer is the “variational approximation” for stickiness.

---

> ### Author Response · Authors · 2025-11-24
> **Rebuttal [part 2]**
>
> *5. What makes Lipschitz continuity meaningful?*
>
> It provides a **formal guarantee of robustness.** If a function is $L$-Lipschitz, we know that an input perturbation of size $\epsilon$ cannot change the output by more than $L \cdot \epsilon$. This bounds the worst-case behavior of the model, which is the fundamental goal of robust learning.
>
> *6. Are there existing frameworks connecting geometry to Lipschitz?*
>
> Yes, primarily Jacobian regularization and spectral normalization (referenced in **Appendix C.2**). However, these constrain the operator norm of the layers (algebraic). **Our framework connects Lipschitz continuity to the spatial distribution of activations (geometric measure theory).** This is a novel perspective: robustness comes from “non-clumping” (stickiness), not just weights.
>
> *7. How computationally intensive is it?*
>
> Exact computation is intensive. However, the **estimation via the KT-CW regularizer is efficient**. It involves (1) sampling $N$ random directions/slabs (linear cost in $N$); (2) computing “soft” counts of features falling into these slabs (matrix multiplication). This adds a linear computational overhead per layer, scaling similarly to standard attention mechanisms, which also involve $O(N^2)$ interactions.
>
> *8. Could you include a small illustrative example?*
>
> Thank you for the suggestion! We have included multiple simulations to demonstrate the core idea of this paper and empirically validate the theoretical results. In our revision, we added **Section 6 (Simulations and Empirical Validations) and Appendix G (Additional Experiment Details and Results).**
>
> We hope the above answers are helpful! Thank you again for the constructive feedback!
>
> Reference:
>
> [1] Virmaux, A., & Scaman, K. (2018). Lipschitz regularity of deep neural networks: analysis and efficient estimation. Advances in Neural Information Processing Systems, 31.
>
> [2] Fazlyab, M., Robey, A., Hassani, H., Morari, M., & Pappas, G. (2019). Efficient and accurate estimation of lipschitz constants for deep neural networks. Advances in neural information processing systems, 32.
>
> [3] Gouk, H., Frank, E., Pfahringer, B., & Cree, M. J. (2021). Regularisation of neural networks by enforcing lipschitz continuity. Machine Learning, 110(2), 393-416.

---

### Official Review · Reviewer_ezGm · 2025-11-09

**Soundness:** 2
**Presentation:** 2
**Contribution:** 2
**Rating:** 2
**Confidence:** 3

**Summary:**

The authors propose a new theoretical framework to analyze and explain the generalization properties of modern deep neural networks. A multi-scale geometric construction is introduced, together with some theoretical analysis, to understand the structure of the hidden representations and their robustness to input perturbations and missing modalities in multimodal settings. A regularizer is then proposed, motivated by this analysis, to control the structure of the hidden representations.

**Strengths:**

- The proposed framework is interesting as it introduces a new construction to understand latent representations. The intuition of covering the representation space with the proposed tubes is meaningful.

- The theoretical results appear to be correct, although I have not followed the proofs in close detail.

- The framework has the potential to inspire new research questions and regularization techniques accordingly.

**Weaknesses:**

- I think that the writing and presentation of the idea do not align well with the expectations of the conference. The clarity of the paper is overall good, but the exposition is quite "texty" and lacks the level of technical detail typically expected in machine learning conferences. In other words, while it seems that there is nothing theoretically incorrect, it is unclear how the proposed approach can actually be implemented in practice.

- In a similar spirit, the paper does not provide any empirical evidence supporting the theoretical claims. The approach is intuitively plausible, but a common belief partially proven in deep learning is that neural networks succeed because they learn sparse representations. I think this seems conceptually different from what the authors propose. Therefore, I would expect some empirical justification to support why the proposed theory may be the correct approach to explain generalization.

- I think some parts of the text can be replaced by technical details on how the method could be implemented, together with empirical results. Moreover, I suggest providing a two- or three-dimensional illustrative example to clarify the desired geometric properties of the representations. The current figures are conceptual but somewhat difficult to interpret.

**Questions:**

Q1. As far as I understand, the idea can be summarized as follows:

- The hidden representations should not concentrate within a convex region (for example, a ball) in the representation space.

- The hidden representations should ideally point in different directions so as to cover the representation space as uniformly as possible, rather than aligning near a common subspace.

Q2. Does the proposed approach imply that the theoretical perspectives in deep learning related to sparse representations may be misleading or incomplete? I would expect some discussion on this point.

Q3. Is it possible to analyze the proposed framework in the context of commonly used optimizers and their generalization behavior, such as SGD? In other words, is it reasonable to ask whether SGD naturally encourages the type of structure in representations that your framework suggests?

Q4. Could you provide some info on whether and how the proposed framework can be implemented in practice?

---

> ### Author Response · Authors · 2025-11-22
> **Rebuttal [part 1]**
>
> Thank you for the thoughtful feedback and for recognizing that our framework is **meaningful, theoretically correct, and has the potential to inspire new research questions**. We value the opportunity to clarify the connection between our geometric theory and practical implementation, particularly regarding sparsity and the nature of the KT-CW regularizer.
>
> **Summary of key clarifications:**
>
> You raised concerns regarding the “texty” exposition of the paper, the apparent lack of implementation details, and a perceived conflict with the intuition of sparse representations. We would like to clarify these points:
>
> 1.	Implementation is concrete: The theoretical framework is operationalized via the KT-CW regularizer, which is defined as a differentiable loss term in Equation 9. We clarify that implementation does not require calculating the exact theoretical infimum but rather relies on a stochastic approximation by minimizing “soft” tube occupancy within sampled random convex sets.
>
> 2.	Alignment with sparsity: Our framework aligns with the intuition that sparse representations are desirable. A low Feature Collapse Constant ($C_{KT-CW}$) explicitly indicates that a representation is geometrically sparse and non-redundant. Conversely, a high constant signifies feature collapse, where inputs map to geometrically clustered regions.
>
> 3.	Geometric intuition: To address the request for visualization, we clarify that Figure 2 depicts “Sticky” representations as effectively covering the space with disjoint grains, contrasting with “Non-Sticky” representations that collapse into dense, vulnerable clusters.
>
> **Point-to-point clarifications:**
>
> *1. On “texty” exposition vs. technical/implementation details*
>
> We acknowledge that the paper is dense with definitions from Harmonic Analysis. This was a deliberate choice to introduce a novel vocabulary (tubes, grains, stickiness) to the community that does not currently exist in standard ML literature. However, we firmly believe the paper provides concrete technical details for implementation in **Section 5 (The KT-CW Regularizer)**. The theory is operationalized in Equation (9) as a differentiable loss function. We define the “Stickiness” loss $\mathcal{L}_{KT-CW}$ as a weighted sum of the Feature Collapse Constants. **To implement this in practice (as detailed in Appendix F.4, Eq. 53-57),** one samples random convex test sets (e.g., boxes or slabs) in the batch feature space and minimizes the “soft” tube occupancy counts within them. This is a direct, calculable penalty on the feature activations.
>
> *2. On empirical evidence and sparse representations*
>
> We respectfully posit that your intuition regarding sparse representations is **fully aligned** with our theory. There appears to be a semantic misunderstanding of our definition of “covering:”
>
> - Our theory supports sparsity: The **Katz-Tao Convex Wolff ($C_{KT-CW}$) Axiom** explicitly penalizes feature redundancy. A low $C_{KT-CW}$ constant means that features do not cluster densely into small regions (collapse). This is mathematically equivalent to saying the representation is geometrically sparse and non-redundant.
>
> - Alignment: Common DL belief holds that sparse is good. Our theory confirms this: “Non-sticky” representations (high $C_{KT-CW}$) correspond to feature collapse (bad), while “Sticky” representations (low $C_{KT-CW}$) correspond to distributed, discriminative features (good).
>
> - Theory track context: As a submission to the **Learning Theory** track, our primary contribution is the **Sticky Representation Theorem**, which provides the provable guarantee that geometric stability implies functional robustness. We included discussions of GAN Mode Collapse (Appendix E.1) and the Lottery Ticket Hypothesis (Appendix E.2) to ground these claims in well-established empirical phenomena.
>
> *3. On illustration examples (2D/3D visualization)*
>
> We agree that visualization is crucial. Figure 2 was intended to serve this purpose, but we can clarify the mental model here: Imagine a 3D feature space ($d=3$).
>
> - Non-Sticky (bad): Inputs from different classes are mapped to feature vectors (tubes) that all point in the same direction, “clumping” into a thin pencil-like shape. A small perturbation perpendicular to this clump pushes a point out of the known distribution.
>
> - Sticky (good): The tubes are arranged like a “bundle of sticks” that are spread out (sparse) and point in different directions, covering the 3D space’s volume effectively. A perturbation to one input does not easily confuse it with another because the geometric structure is stable.
>
> - We run multiple simulations, including 2D, 3D, single-modal, and multimodal, to empirically validate the core contributions. In our revision, we added **Section 6 (Simulations and Empirical Validations) and Appendix G (Additional Experiment Details and Results).**

---

> ### Author Response · Authors · 2025-11-22
> **Rebuttal [part 2]**
>
> *4. Q1: Hidden representations should not concentrate… and should point in different directions to cover the space?*
>
> **Yes, this is correct.** Specifically, we require that the “tubes” (feature activations) satisfy the **Wolff Axioms**, meaning they cannot be packed too densely into any convex set ($C_{KT-CW}$). The “coverage” refers to the volume of the union of these tubes being large (Assertion $D/E$), which prevents the representation from collapsing into a low-dimensional subspace.
>
> *5. Q2: Does this imply theoretical perspectives on sparse representations are misleading?*
>
> **No, it refines them.** Our framework provides a geometric definition of sparsity. We argue that “stickiness” (geometry preservation) is the mechanism by which sparse representations achieve robustness. A representation with a low Feature Collapse Constant ($C_{KT-CW}$) is explicitly “geometrically sparse.” We offer a rigorous sparsity metric rather than relying on heuristic L1 norms.
>
> *6. Q3: Does SGD naturally encourage this structure?*
>
> This is a question touched upon in **Appendix E.3**. While SGD optimizes in parameter space, we hypothesize that “winning ticket” (successful subnetworks) are initialized in a way that makes them naturally “sticky.” We argue that standard SGD alone does not guarantee stickiness (hence the fragility of standard models), which is why we derive the **KT-CW regularizer to guide SGD toward robust geometric configurations explicitly**.
>
> *7. Q4: How can this be implemented in practice?*
>
> The implementation follows the **KT-CW regularizer** (Proposition 1) and the **differentiability proof** in Appendix F.4. Detailed steps are as follows:
>
> **Step 1:** At layer $l$, treat the batch of feature vectors $Z_l$ as a set of lines (tubes).
>
> **Step 2:** Sample a set of random convex proxies (e.g., random boxes or slabs) $W_{\theta}$.
>
> **Step 3:** Calculate the “soft” occupancy: how many tubes pass through each box using a soft-max approximation.
>
> **Step 4:** Minimize the maximum occupancy (reduce $C_{KT-CW}$) via gradient descent. This acts as an auxiliary loss term, $L_{total} = L_{task} + \lambda L_{KT-CW}$.
>
> We hope the above answers your questions. Thank you again for the constructive feedback!

---

### Author Response · Authors · 2025-12-03
**Author Final Remarks**

Dear Reviewers, Area Chairs, and Program Chairs,

We sincerely thank the reviewers for their rigorous engagement, which has significantly strengthened this work. Recognizing the consensus request for empirical grounding and clearer definitions, we have extensively revised the paper.

**1. Summary of Changes & Response to Feedback**

- **Empirical Validation (Addressing ezGm, cJhi, GQCC):** While submitted to the Learning Theory track, we acknowledged the need for verification. We added Section 6 and Appendix G, comprising controlled simulations on single-modal and multimodal datasets. These experiments demonstrate that “Sticky” models (trained with our KT-CW regularizer) maintain full-rank spectra and show superior robustness to FGSM attacks and missing modalities, empirically validating Theorems 1 and 2.

- **Computability & Implementation (Addressing GQCC, ezGm):** We clarified that our KT-CW regularizer does not require intractable exact infimums. We detailed the stochastic approximation method in Appendix F.4, demonstrating that the geometric framework is computationally feasible and distinct from the density-estimation challenges of the information bottleneck.

- **Definition & Intuition (Addressing qLBx):** We corrected the notations and incorporated formal definitions into the main text. We also clarified why we model features as tubes rather than balls: to capture the directionality of feature collapse in high-dimensional spaces, a nuance that isotropic neighborhoods miss.

**2. Core Contributions**

This paper bridges Harmonic Analysis (Geometric Measure Theory) and Deep Learning to offer a novel perspective on robustness:

- **Unified Framework:** We introduce the Representation Field and Multi-Scale Wolff Axioms to quantify the emergent geometry of latent spaces.

- **The Sticky Representation Theorem:** We provide a provable link between geometric “Stickiness” (non-collapse) and functional Lipschitz stability.

- **Multimodal Generalization:** We extend this to joint embedding spaces, proving bounds for robustness against missing modalities.

- **New Design Principle:** We derive the KT-CW Regularizer, converting theoretical axioms into a differentiable training objective.

In summary, this work presents a robust alternative to existing spectral or Jacobian-based analyses. As noted by the reviewers, it is creative, mathematically original, and intellectually stimulating. We believe this paper provides the theoretical novelty required for the Learning Theory track and the practical grounding expected by the community. Furthermore, it has the potential to inspire further research into robustness in representation learning.

We thank you again for your time and consideration. We believe that it is worth publishing to stimulate further discussion.

---

### Meta-Review · Area_Chair_ZZE6 · 2026-01-06

**Summary:**

This work proposes a new framework for robust representation learning, from multi-scale analysis. I am not sure how to evaluate this work, but I agree with the general impression from other reviewers: (1) the theory is of unclear significance; (2) the empirical evaluation is weak and nonextensive. I personally think this paper can be a quite good position paper, and the authors can do a lot of follow-up works to substantiate this framework. But this paper per se feels quite speculative, with unsubstantiated discussions and claims about its relevance to deep learning. I thus recommend rejection.

**Reviewer Concerns:**

There are a lot of criticisms about the lack of empirical analysis. The authors added some experiments in the revision, but they are done on very simple datasets and very simple networks, and I think this is insufficient to convince an average machine learning reviewer about the empirical relevance of the paper.

**Reviewer Scores:**

I do not think the scores will change significantly

---

### Decision · Program_Chairs · 2026-01-26

Reject